# Marine ecosystem shifts with deglacial sea-ice loss inferred from ancient DNA shotgun sequencing

Heike H. Zimmermann [1,9], Kathleen R. Stoof-Leichsenring [1], Viktor Dinkel [1,2], Lars Harms [3], Luise Schulte [1], Marc-Thorsten Hütt [2], Dirk Nürnberg [4], Ralf Tiedemann[5,6] & Ulrike Herzschuh [1,7,8] ✉

Sea ice is a key factor for the functioning and services provided by polar marine ecosystems. However, ecosystem responses to sea-ice loss are largely unknown because time-series data are lacking. Here, we use shotgun metagenomics of marine sedimentary ancient DNA off Kamchatka (Western Bering Sea) covering the last ~20,000 years. We traced shifts from a sea ice-adapted late-glacial ecosystem, characterized by diatoms, copepods, and codfish to an ice-free Holocene characterized by cyanobacteria, salmon, and herring. By providing information about marine ecosystem dynamics across a broad taxonomic spectrum, our data show that ancient DNA will be an important new tool in identifying long-term ecosystem responses to climate transitions for improvements of ocean and cryosphere risk assessments. We conclude that continuing sea-ice decline on the northern Bering Sea shelf might impact on carbon export and disrupt benthic food supply and could allow for a northward expansion of salmon and Pacific herring.

The organismal composition in high-latitudinal oceans is highly vulnerable to anthropogenic global warming, which may alter ecosystem services (e.g., food supply, biological carbon pump) due to sea-ice loss and related effects on rising ocean temperature, increasing light transmission, stronger water column stratification and decreasing nutrient availability[1]. This will likely change the seasonal timing, biomass, and composition of algal blooms, which play a central role in trophic interactions by supporting food webs, including benthic communities and primary producers[2].

Long-term trends in the Bering Sea show an ongoing decline of sea-ice duration which is projected to shorten further due to later freeze-up (by 20 to 30 days) and earlier break-up (by 10–20 days) until the middle of this century[3]. Extreme events, such as the lowest sea-ice extent in the Bering Sea over the last 5500 years[4] in 2018 with persistent southerly warm winds, allow for assessments of immediate ecosystem responses to sea-ice loss. Cascading effects through the food web have been recorded, possibly resulting from reduced energy transfer from lower to upper trophic levels[5]: A late spring phytoplankton bloom and a scarcity of large, lipid-rich copepods led to decreasing abundance of young pollock and other forage fish that were later linked with seabird reproductive failures as well as seabird and seal mortality events in 2018 and 2019[6].

A better understanding of ecosystem changes during the transition from seasonal sea-ice to ice-free conditions due to ongoing

[1]Alfred Wegener Institute Helmholtz Centre for Polar and Marine Research, Polar Terrestrial Environmental Systems, D-14473 Potsdam, Germany. [2]Constructor University Bremen, Computational Systems Biology, Bremen D-28759, Germany. [3]Alfred Wegener Institute Helmholtz Centre for Polar and Marine Research, Data Science Support, D-27568 Bremerhaven, Germany. [4]GEOMAR Helmholtz Centre for Ocean Research Kiel, Ocean circulation and climate dynamics, D-24148 Kiel, Germany. [5]Alfred Wegener Institute Helmholtz Centre for Polar and Marine Research, Marine Geology, D-27568 Bremerhaven, Germany. [6]MARUM, Center for Marine Environmental Sciences, University of Bremen, D-28334 Bremen, Germany. [7]University of Potsdam, Institute of Biochemistry and Biology, D-14476 Potsdam, Germany. [8]University of Potsdam, Institute of Environmental Sciences and Geography, D-14476 Potsdam, Germany. [9]Present address: Department of Glaciology and Climate, Geological Survey of Denmark and Greenland (GEUS), DK-1350 Copenhagen, Denmark. ✉e-mail: Ulrike.Herzschuh@awi.de

climate warming is needed. Despite extensive monitoring efforts, we still have scarce knowledge on long-term ecosystem responses to climate transitions for many taxonomic groups, particularly zooplankton, fish, non-fossilizing algae, and benthic organisms such as tunicates, starfish, or macrophytes. Furthermore, the ability and constraints of ecosystems to adapt are uncertain, which limits ocean and cryosphere risk assessments, as emphasized in the IPCC report[7]. The short-term responses of the Bering Sea ecosystem to warmer and colder regimes have been well-documented[2,5,8], yet long-term rearrangements of the organismal composition in pelagic and benthic communities are not clear and more data are needed to constrain the future ecosystem development. While pelagic communities are strongly linked to hydrographic factors and climate, benthic communities are strongly dependent on primary production and sinking particles from the water column above[9]. Past ecosystem responses to sea-ice loss at the Pleistocene–Holocene boundary in the southern part of the Bering Sea could possibly reveal such long-term rearrangements and serve as an analog for future changes in the Pacific Arctic.

Marine sediments are a natural archive of climate history from which sea-ice can be reconstructed via proxy records, such as from biomarkers[10] or microfossil remains[11]. In the Bering Sea, a palaeoceanographic framework for past sea-ice coverage has been based on reconstructions using the highly branched isoprenoid alkane biomarker IP$_{25}$[12] (produced by diatoms bound to a life in seasonal sea-ice[10]) and microfossils (from diatoms)[11]. Alkenones, which are produced by haptophytes can add complementary information about sea-surface temperatures of the late summer/early fall season (SST$_{UK'37}$)[13]. Physical remains, such as diatom frustules[11,14–16], provide evidence for past organismal responses to sea-ice variability. For many non-fossilizing organismal groups (e.g., many protists and zooplankton) or such whose fossil records remain underexplored (e.g., fish), analyses of sedimentary ancient DNA (sedaDNA) have the potential to fill the gap.

Within marine settings, sedaDNA metabarcoding (enrichment of a genetic marker sequence via polymerase chain reaction (PCR)) has revealed past responses of plankton to changes, for example, in water mass characteristics[17] or sea-ice[18–20]. Recently, sedaDNA applications were expanded by metagenomic shotgun sequencing[21–23], which increases the spectrum of taxonomic groups that can be analyzed in a single approach, facilitates authentication by profiling damage patterns, and circumvents typical PCR biases, arising, for example, from the mismatch between the ancient DNA fragment size distribution and the size of genetic markers[22].

Correlation networks have been used in the interpretation of multidimensional data by means of co-occurrence networks and contributed to our understanding of ecosystems by identifying, for example, habitat preferences in aquatic bacterial communities[24] or geographic co-occurrence patterns in European amphibians, reptiles, breeding birds, and mammals[25]. The sparsity of metagenomic datasets resulting from either true absences or insufficient sequencing depth can lead to spurious correlations[26]. Gaussian copula graphical models (GCGMs) are a novel statistical framework developed to separate environmental effects from intrinsic interactions among taxa[27]. They are suited to address the compositional structure of sequencing data, non-linear correlations, and to remove spurious correlations via mediator taxa (if a correlation between two taxa arises only because both are correlated to a third taxon) or via similar responses to environmental effects (covariance). Correlation networks have not yet been applied to sedaDNA data before. As sedaDNA samples contain averaged information on decadal to centennial time scales depending on sedimentation rates, correlation in the network should not be interpreted as ecological interactions between linked taxa or actual co-occurrence on a short-timescale. We here assume that positively correlated families show similar responses to environmental changes (i.e., sea-ice cover and SSTs).

Here, we explore the potential of sedaDNA metagenomic shotgun sequencing to reveal shifts in ecosystem composition in response to postglacial sea-ice loss using sediment core SO201-2-12KL, which was taken from the western continental slope of Kamchatka about 70 km from the coast of Kamchatka (Fig. 1). To identify taxa that show a similar response to sea-ice loss, we used co-occurrence networks and compared Spearman networks with GCGMs. The paleoenvironmental backbone is based on previous multi-proxy reconstructions from the same core, including diatom microfossils and the biomarker IP$_{25}$. The proxies suggest that the coring site was covered by seasonal sea-ice during the Last Glacial Maximum (LGM), Heinrich Stadial 1, and the Younger Dryas, while the phases of the Bølling-Allerød and the Holocene were predominantly sea-ice free[12,16]. Our approach includes a broad spectrum of taxonomic and functional groups, particularly among eukaryotes, which we consider a major step towards deciphering the consequences of climate transitions for marine ecosystem structure.

## Results and discussion
### Shotgun sequencing results

Sequencing (25 samples, five negative controls) resulted in 918,186,452 paired-end reads of which, on average, 70.76% of the reads passed the quality check within samples (Supplementary data file 1). Of those, we were able to classify 13,119,146 read pairs (merged and paired-read pairs together). About 98.5% of our sequences remained unclassified (based on the full NCBI nt database), which is in a similar range as reported for metagenomic shotgun sequencing of ancient cave sediments (79.1–96.1%)[28] and ancient lake sediments (99.6[29], 84[30,31], 97%[32]). Our results are not surprising since estimates from 2011 suggest that

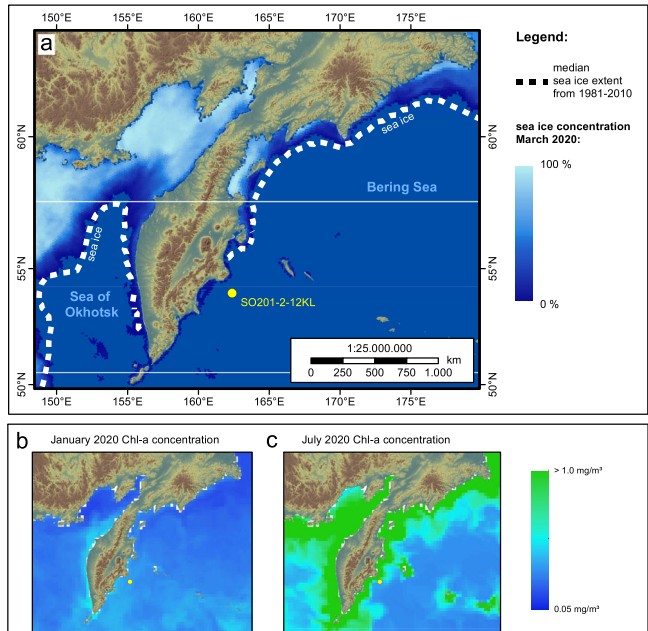

**Fig. 1 | Map showing the coring site SO201-2-12KL (yellow dot) in the context of modern sea-ice and primary productivity. a** The coring site (53.993°N, 162.37°E; water depth 2173 m) is located south of the modern sea-ice extent as shown by the median for March between 1981–2010[103] (dotted line) and the mean modern sea-ice concentration for March in 2020[104, 105] (color scale from blue to white). **b** Modern seasonal patterns of primary productivity are indicated by a low mean chlorophyll-a concentration in January compared to **c** higher mean chlorophyll-a concentrations in July from 2020 (E.U. Copernicus Marine Service Information: https://resources. marine.copernicus.eu/?option=com_csw&view=details&product_id=GLOBAL_ ANALYSIS_FORECAST_BIO_001_028). The map was created with ArcGIS using the Global 30 Arc-Second Elevation (https://www.usgs.gov/centers/eros/science/usgs-eros-archive-digital-elevation-global-30-arc-second-elevation-gtopo30#overview).

91% of marine species have not yet been described[33]. Especially among protists, the sequence reference database is largely incomplete[34], resulting in low confidence at genus and species level assignments[35]. Therefore, we limited our analyses to assignments on the family level. Most classified read pairs overlapped and were merged (72.5%). Among samples, this resulted in mean fragment lengths between 83 and 105 bp per sample, which matches the expectation that *sed*aDNA is highly fragmented[36].

### Pelagic taxonomic assignments are dominated by primary producers and fish

The novel *sed*aDNA metagenomics approach allowed for the successful identification of 167 families with 456,058 read pairs after filtering and resampling, representing functional and taxonomic groups (Supplementary data file 1) which are known from sea-ice or the pelagic or demersal zones (hereafter grouped as pelagic) of the subarctic North Pacific and the Bering Sea. Of all samples combined, high shares of sequences are assigned to fish (66.8%), phototrophic protists (10.2%), and phototrophic bacteria (12.8%) after resampling. Among phototrophic families (including mixotrophic taxa), the highest numbers of reads are assigned to pico-sized Cyanobacteria (11.5%), Chlorophyta (3.6%), and Chlorobi (0.8%), as well as to typically nano- to micro-sized diatoms (4.7%) and haptophytes (1.0%) (Fig. 2).

The proportions of pelagic resampled reads assigned to phototrophic primary producers exceed those of zooplankton, which comprises single-celled, heterotrophic protists (2.1%) and copepods (0.3%). The group of heterotrophic protists is phylogenetically diverse, and the highest read counts are found for Ciliophora (0.8%), colloidarian Polycystinea (0.3%), Discoba (0.3%), Choanoflagellata (0.2%), and Dinoflagellata (0.1%).

The low proportion of assigned reads likely reflects a strong grazing intensity on zooplankton in the upper water column, which leads to a low zooplankton DNA export towards the seafloor. Alternatively, this discrepancy between the proportions of assigned reads to zooplankton (multicellular organisms) and phytoplankton (unicellular organisms) could be explained by differential preservation or degradation due to different skeletal properties or large gaps in the sequence reference database regarding marine macrofauna for both barcoding genes and genomes[37].

Among the analyzed metazoans, most reads were assigned to fish of the families Salmonidae (12.1%), Serranidae (8%), and Gadidae (6%), which are common in the subarctic North Pacific and its adjacent seas[2,38]. Furthermore, we found few, but notable read counts assigned to marine mammals (4.5%) such as whales (Delphinidae 0.3% and Balaenopteridae 0.3%), seals (Otariidae 0.1% and Phocidae 0.4%), and sea otter (Mustelidae 2.9%), which are also common in this region.

### Co-occurrence network-derived pelagic environment associations

We investigated the influence of environmental variables such as temperature and seasonal sea-ice cover using a co-occurrence network based on Spearman rank correlations (Fig. 3). The Spearman network (167 pelagic families kept after filtering for at least ten counts and in at least three samples and resampling) is composed of two modules containing 148 nodes (families) and 446 edges (positive correlations Spearman's $\rho > 0.4$, adjusted $p$ value <0.1) (Fig. 3a). The upper module contains 42 nodes, of which eight positively correlated (adjusted $p$ value <0.1; Supplementary Table 4) with IP$_{25}$. We, therefore, define it as the sea-ice module. The lower module contains 120 nodes, of which 13 are showing a trend or are positively correlated (adjusted $p$ value <0.1; Supplementary Table 4) with higher SST. Therefore, we define it as the sea-ice-free module, hereafter.

As Spearman correlations cannot separate environmental effects from intrinsic interactions among taxa, we used ecoCopula to test how mediator taxa and environmental conditions (IP$_{25}$ and SSTs used as covariates) affect associations between taxa.

The ecoCopula network is composed of 167 nodes and 474 edges, which is in a similar range compared to the edge density of the Spearman network. Two modules are dominated by a specific functional group: the green and red modules by diatoms and the violet module by fish, while the other modules are more mixed (Supplementary Fig. 13a). Despite removing the environmental effects of IP$_{25}$ and SSTs for network generation, the green module is mainly composed of families that are positively correlated with IP$_{25}$ in the Spearman network suggesting a strong co-occurrence pattern of these families based on biogeographic distribution patterns or other environmental factors not included here, such as nutrient availability, salinity, or light conditions. After accounting for the effects of IP$_{25}$, SSTs, and mediator taxa, a high overlap of edges (34% of positive associations) can be found between the two network approaches, which is significantly different from the randomized null-model (Supplementary Fig. 13a, b).

### Environmental preference of pelagic taxa inferred by network analysis

Nodes of the sea-ice module contain families that show higher proportions during the LGM, Heinrich Stadial 1, and Younger Dryas compared to warmer phases (Fig. 3a). Among algae, this pattern can be observed for Bacillariaceae (0.8% of pelagic counts), Stephanodiscaceae (0.6%), Thalassiosiraceae (1.1%), Triparmaceae (0.3%), and Phaeocystaceae (0.6%), which can be found in sea-ice or as a part of cryopelagic communities[39]. They co-occur with the chlorophytes Mamiellaceae (0.5%) and Bathycoccaceae (1.7%), which contain cold-adapted lineages that prevail in the marginal ice zone or in landfast sea-ice[40,41]. Closely linked to these primary producers are copepods (Calanidae 0.03%, Metridinidae 0.03%), which rely on sea-ice algae as a food source due to their high quantity of polyunsaturated fatty acids[42]. Among fish, Gadidae, for example, are linked with the above-mentioned algae and copepods. Reads assigned to this family can most likely be attributed to Pacific cod (*Gadus macrocephalus*), walleye pollock (*Gadus chalcogrammus*), or Polar cod (*Boreogadus saida*). While pollock and Pacific cod are generalist predators in the Bering Sea, they are strongly influenced by bottom-up effects through prey composition and availability[38,43] compared to the Polar cod, which is a keystone species of the high Arctic with a strong linkage to sea-ice[44].

The 8 families that are positively correlated with IP$_{25}$ (adjusted $p$ value <0.1) contain 109 links to 28 families. Of those links, 42.2% are shared with diatoms (15.6% centric and 26.6% pennate), which comprise overall some of the most connected families in the network. This emphasizes that diatoms are the central functional group among primary producers, particularly during phases characterized by seasonal sea-ice cover, and that sea-ice might act as a strong environmental filter favoring organisms with adapted functional traits.

The ice-free ecosystem composition, which prevailed for most of the Holocene (Fig. 2), suggests a response of past biota to warmer SSTs with reads assigned to families known to occur in warmer surface waters from their modern spatial distribution (Hemiaulaceae[45], Rhizosoleniaceae[46], and Chloropicaceae[47]).

The 28 families in the network that are positively correlated with SSTs (adjusted $p$ value <0.1) encompass 114 links to 51 families within 13 functional groups (Fig. 3). Hence, connections of SST-correlated families are more diverse compared to those of IP$_{25}$-correlated families. Families, which are positively correlated with SSTs, have on average fewer neighbors (i.e., links to adjacent nodes) and, therefore, are less densely linked in comparison to IP$_{25}$-correlated families ($p$ value <0.001, Supplementary Fig. 11). Neighbors of SST-correlated families belong mostly to the functional groups of fish (43% of links), phototrophic bacteria (25.4%), Chlorophyta (11.4.3%), and centric

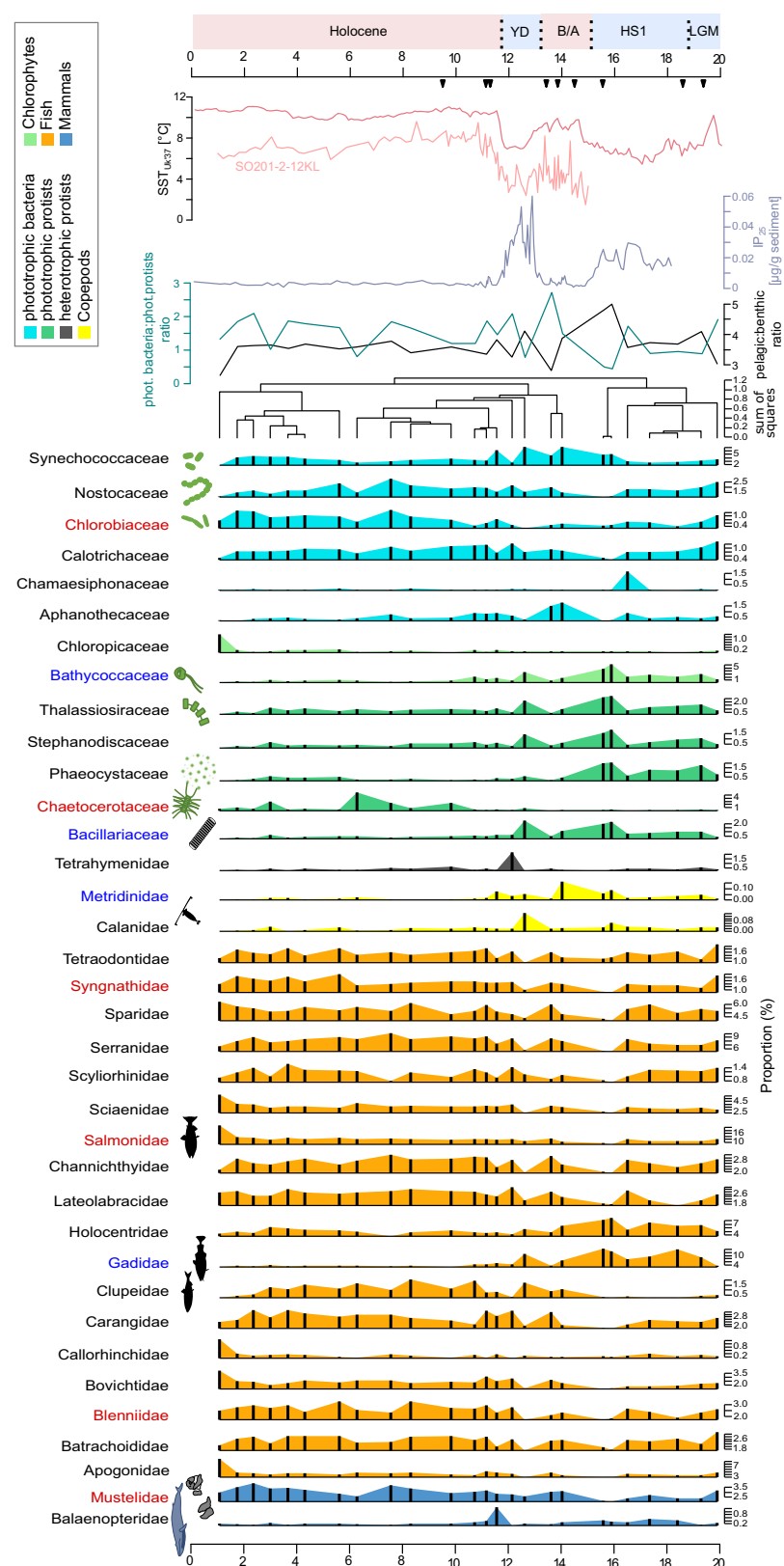

**Fig. 2 | Pelagic families in the *seda*DNA record of core SO201-2-12.** Only families represented in at least three samples and by at least 1.5% proportion of read counts after resampling are shown. The dendrogram (based on constrained hierarchical clustering) and IP$_{25}$ concentrations from the same core[98] and sea-surface temperature (SST$_{Uk'37}$) reconstructions of the Northwest Pacific[52] (light red) and Northeast Pacific[97] (dark red) are added for comparison. Families which are significant, and positively correlated with IP$_{25}$ or SST$_{Uk'37}$ are highlighted in blue or red, respectively. The pelagic:benthic ratio shows the proportion of reads assigned to pelagic families in relation to reads assigned to benthic families. The ratio between reads assigned to phototrophic bacteria and reads assigned to phototrophic protists is given as a blue-green line. Black triangles next to the IP$_{25}$ record mark radiocarbon-dated calibrated ages[52]. LGM last glacial maximum, HS1 Heinrich stadial 1, B/A Bølling/Allerød, YD Younger Dryas, and colors of the boxes refer to warm (red) and cold (blue) phases. Source data are provided as a Source Data file.

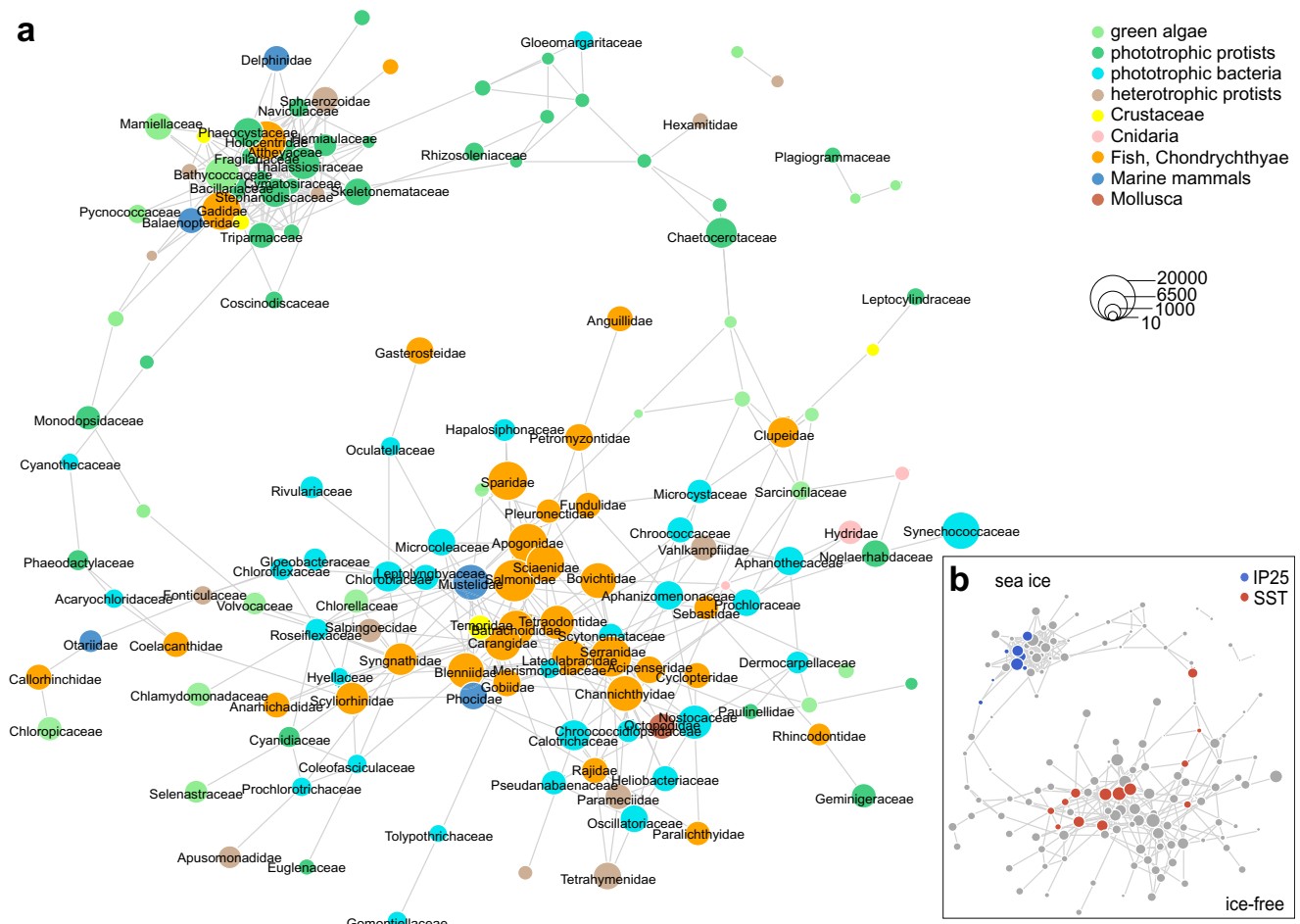

**Fig. 3 | Correlation network of pelagic families.** Families (nodes) that are positively correlated (Spearman rank correlation coefficients >0.4, Benjamini–Hochberg adjusted *p* value < 0.1) with each other are connected by links (edges) after resampling. The node size is log-scaled according to taxa abundance, while node colors represent **a** the functional group to which the organisms belong, and **b** whether the node showed a positive trend (Spearman rank correlation coefficients >0.4, adjusted *p* value < 0.1) with the seasonal sea-ice biomarker IP$_{25}$[98] (blue) or SSTs[97] (red). Node labels are given if a family exceeds the log-scaled threshold of 4. Source data are provided as a Source Data file.

diatoms (1.8%). We assume that predominantly sea-ice-free conditions, and thus a longer growing season, might have favored a broader spectrum of families.

Higher read counts assigned to Clupeidae (Pacific herring) and Salmonidae in warm phases, for example, the early Holocene (Fig. 2), are consistent with previous studies that have shown that warmer temperatures provide good conditions for fish spawning, quick hatching from eggs, and low mortality rates during early life stages[48]. These patterns support modern observations of increased herring distribution on the northern Bering Sea shelf in warmer than in colder years[49] and of increasing population densities in pink salmon since the 1990s[50]. Rivers in Kamchatka and nearby Chukotka are spawning grounds for Salmonidae[51] and are known as a diversity hotspot; for example, all species of Pacific salmon (*Oncorhynchus*) occur in the area. Summer temperature reconstructions from Chukotka show a similar trend to SST reconstructions from our core[52] with lowest values pre-15 ka and an about 5 °C rise until the early-to-mid Holocene temperature maximum[53].

The ice-free sub-network is mainly characterized by phototrophic bacteria and chlorophytes (Fig. 3a). Furthermore, we find indications of increased continental runoff reflected in the presence of different functional groups of freshwater organisms (e.g. Sarcinofilaceae (*Sarcinofilum mucosum*; chlorophyte[54]), Roseiflexceae (*Roseiflexus*; bacteria[55]), Acanthocerotaceae (*Acanthoceras zachariasii*; diatom[56]);

Supplementary Fig. 17). While freshwater discharge is a source of organic carbon and nutrients that can enhance productivity, the lower salinity of surface waters increases vertical water column stratification[57]. Over the course of the productive season, nutrients may become depleted by the spring/early summer phytoplankton blooms, especially when the water column is strongly stratified by warm sea-surface temperatures and freshwater input. Under nutrient-depleted conditions, cyanobacteria are assumed to have several advantages, such as the small surface area-to-volume ratio of their cells and thus lower nutrient requirements[58], the capability of nitrogen fixation (here limited to Nostocaceae), and their preference for inorganic nitrogen in the form of ammonium. Ammonium has an energetic advantage compared to nitrate, the preferred form by diatoms, which has to be transformed to ammonium within the cell at an energetic cost[59].

## Benthic *sed*aDNA composition and temporal patterns

Despite the vastness of deep-sea benthic ecosystems, the responses of its inhabitants to past climatic changes are still underexplored. From benthic foraminifers and ostracods, we know that deep-sea habitats have been influenced by glacial–interglacial cycles in the past[60–62].

The overall lower number of reads attributed to benthic (263,193 originally, 45,975 after resampling) in comparison to pelagic (476,058 originally, 210,800 after resampling) organisms

could suggest that upper-ocean productivity is higher than benthic productivity at the deep continental margin. However, the low assignment rate, particularly among benthic fauna, may, to an unknown extent, also relate to a large number of undescribed species and the sequence database gap of deep-sea biota in general[63]. Some typical deep-sea biota were detected, such as free-living nematodes (Oxystominidae, 0.1%), albeit only with a few reads (Supplementary data file 2).

The majority of resampled reads were assigned to families, which potentially represent the benthic composition of the shelf and upper continental slope[64,65], such as bivalves (Pectinidae, 22.7% and Mytilidae, 1.5%), starfish (Asteriidae, 14.1%), priapulid worms (Priapulidae, 6.9%), corals (Nephtheidae,1%; Pocilloporidae, 2.8%), and sea urchins (Strongylocentrotidae, 4.5%) (Fig. 4). Storms can whirl up and redistribute benthic inhabitants[66,67], and the very steep Kamchatka slope may foster downslope transport, which might also explain the considerable share of reads assigned to near-shore macrophytes such as seagrass (Zosteraceae, 0.8%) and brown (Phaeophyta, 2.9%) and red macroalgae (Rhodophyta, 10.9%) in our record. The ratio between counts assigned to pelagic and benthic families (pelagic:benthic ratio) is highest in samples dated to the late glacial and Younger Dryas compared to the warmer Bølling/Allerød and the Holocene (Figs. 2, 4). During phases with higher SSTs there is a general increase in read counts of pelagic and benthic taxa. This increase is proportionally stronger in the benthic compared to pelagic taxa resulting in a smaller difference between them under warmer SSTs and, thus a smaller pelagic:benthic ratio. Overall, the pelagic:benthic ratio shows a weak, negative correlation with SSTs (Spearman's $\rho = -0.39$, $p$ value = 0.052), indicating that more reads are assigned to benthic families in phases with warmer SSTs.

In our dataset, Zosteraceae have higher read counts during the Holocene and are significantly, positively correlated with summer/early fall SSTs (Spearman's $\rho = 0.73$, adjusted $p$ value = 0.026; Fig. 4b). Laminariaceae (kelp) have higher read counts during the late glacial. Kelp form underwater forests on shallow, rocky, cold-water coasts with complex vertical structures that provide habitat for epiphytic algae, and a food source for sea urchins (Strongylocentrotidae)[68]. In addition, the occurrence of kelp in the North Atlantic is linked to the behavioral traits of cod[69]. The co-occurrence of kelp and Pacific cod in the assemblages of ice-rich periods (Figs. 2, 4) suggests functional links between benthic and pelagic taxa in our study. Seagrass rather grows in temperate waters with soft sediments, such as sheltered, shallow bays[70]. The modern distribution range in the north is restricted to circumboreal coasts, which is in line with our finding of low late-glacial abundance in our study area and that in response to warmer and ice-free conditions in the Holocene, a range expansion occurred. Important benthic-pelagic relationships are established between seagrass and salmon as well as Pacific herring, including foraging opportunities and spawning grounds[71,72], which are reflected in our data by similar temporal co-occurrence patterns.

## Implications for sea-ice decline in the northern Bering Sea

Our study combines metagenomic shotgun sequencing of sedaDNA with a correlation network analysis[24] to explore changes in organismal composition and potential co-occurrence in the course of sea-ice change over the past ~20,000 years. During warmer, ice-free phases, namely the Bølling/Allerød and since the early Holocene, fewer sequences are assigned to diatoms, and sequences assigned to Cyanobacteria dominate among primary producers. Phytoplankton groups with large cell sizes, such as diatoms, are assumed to contribute more to carbon export than other pico-sized plankton, such as phototrophic bacteria[73], which are presumed to be mostly recycled by the microbial loop in the water column[74]. A shift from larger phototrophic protists to smaller bacterioplankton in the course of sea-ice loss could therefore alter the amount and quality of food-

sustaining benthic organisms[75] and the efficiency of the biological carbon pump[58]. Similar observations have been made in the Arctic, where sea-surface freshening due to increased meltwater and runoff have led to a rise in cell abundance of upper-ocean bacterioplankton, while the abundance of nanophytoplankton decreased[58]. The latter study also found that despite the abundance shifts of the two plankton size classes, the chlorophyll-a content, which is often used as a proxy for biomass, remained stable. These results, combined with the data presented in this study, suggest that paleo-productivity reconstructions based on biogenic silica (produced e.g., by diatoms and silicoflagellates) or calcium carbonate (produced e.g., by haptophytes and foraminifers) can strongly underestimate past primary productivity estimations especially during climate transitions, because they do not consider changes in taxonomic composition among primary producers.

Although metagenomic shotgun sequencing is currently limited by the lack of reference databases to assess past diversity changes, our approach detected a variety of fish and marine mammals that are common in coastal areas of the Bering Sea and the subarctic North Pacific. So far, no studies have been published that target marine mammals in sediments using ancient DNA, and the only marine sedaDNA record targeting fish extends back 300 years[76]. By pushing back the detection limit of fish and mammals to almost 20,000 years, our approach paves the way to assess millennial-scale to glacial–interglacial changes in the distribution and natural dynamics of these key groups in future studies. Furthermore, our results suggest that in the future, sedaDNA holds promise in identifying the organismal sources of organic matter, which has been recognized as one of the key fundamental questions in blue-carbon science[77].

Overall, our results indicate that sea-ice is a key factor for the structure and functioning of the near-coastal Western Bering Sea ecosystem on a millennial timescale. By analogy to the past, substantial shifts of pelagic and benthic compositions can be expected for subarctic regions that will likely become ice-free in the next decades, such as the northern Bering Sea shelf, which today is subject to frequent regime shifts between cold and warm conditions and related variation in sea-ice extent[2]. Along with changes in ecosystem composition, alterations in ecosystem services can be expected, particularly with respect to carbon sinks and fishing grounds.

This sedaDNA record implies that ongoing climate warming on the Bering Sea shelf might lead to a shift towards a pico-sized phytoplankton community composed mainly of cyanobacteria (Fig. 5) where previously, primary productivity was dominated by nano- and micro-phytoplankton. This is supported by observations that recent warming and increasing freshwater runoff is paralleled by increasing picoplankton abundance[58]. The dominance of picoplankton-derived DNA in our dataset indicates efficient carbon export and burial of Cyanobacteria, potentially facilitated by aggregate formation or inclusion in detritus. However, the lack of polyunsaturated fatty acids and their potential toxicity might limit the value of Cyanobacteria as a high-quality food source for benthic organisms[78].

Our data also suggest that climate warming and prolonged sea-ice-free conditions in coastal areas allow for the northward expansions of seagrass in protected, soft-sediment bays and kelp along shallow, rocky coastlines of the Russian and North American Arctic[79]. The expansion of seagrass meadows, in particular, could assist the northward expansions of Pacific herring and salmon, which our data suggests could be benefiting from climate warming on the northern Bering Sea shelf. However, high abundances of pink salmon have been connected with negative effects on other species, such as increased seabird mortalities, changes in nesting phenology, and a decrease in breeding success due to their competitive advantage over shared prey[5,50]. In contrast, warmer, ice-free conditions leave pollock and cod at a disadvantage. Their survival depends on the availability of large copepods, which are reduced in warm years, as suggested by our data

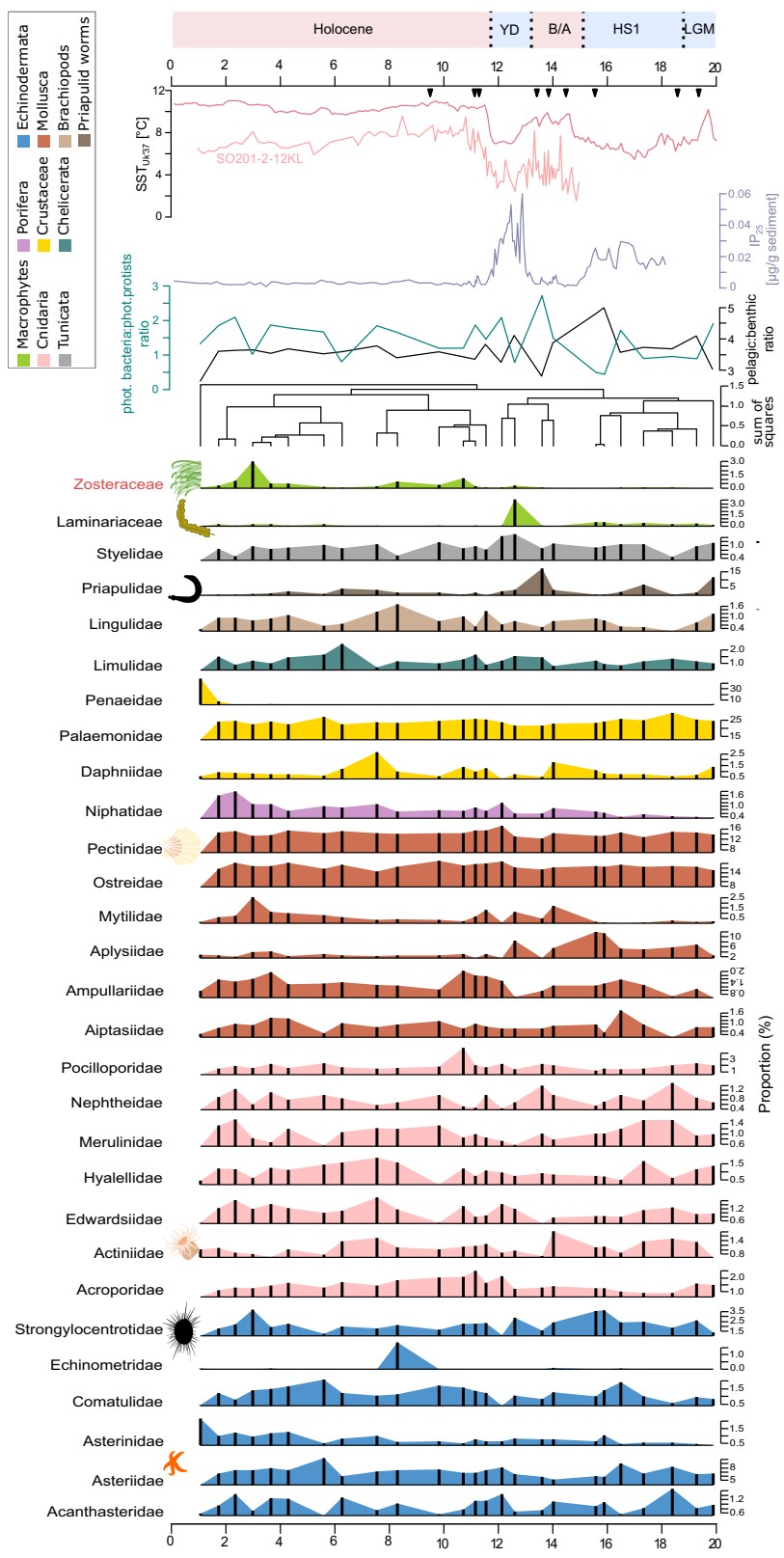

**Fig. 4 | *Sed*aDNA record of benthic families of core SO201-2-12.** Only families represented in at least three samples and by at least 1.5% proportion of read counts after resampling are shown. The dendrogram (based on constrained hierarchical clustering) and IP$_{25}$ concentrations from the same core[16] and sea-surface temperature (SST$_{Uk'37}$) reconstructions of the Northwest Pacific[52] (light red) and Northeast Pacific[97] (dark red) are added for comparison. Zosteraceae, which is significant, and positively correlated with SST$_{Uk'37}$, are highlighted in red. The pelagic:benthic ratio shows the proportion of reads assigned to pelagic families in relation to reads assigned to benthic families. The ratio between reads assigned to phototrophic bacteria and reads assigned to phototrophic protists is given as a blue-green line. Black triangles next to the IP$_{25}$ record mark radiocarbon-dated calibrated ages[52]. LGM last glacial maximum, HS1 Heinrich stadial 1, B/A Bølling/Allerød, YD Younger Dryas, and colors of the boxes refer to warm (red) and cold (blue) phases. Source data are provided as a Source Data file.

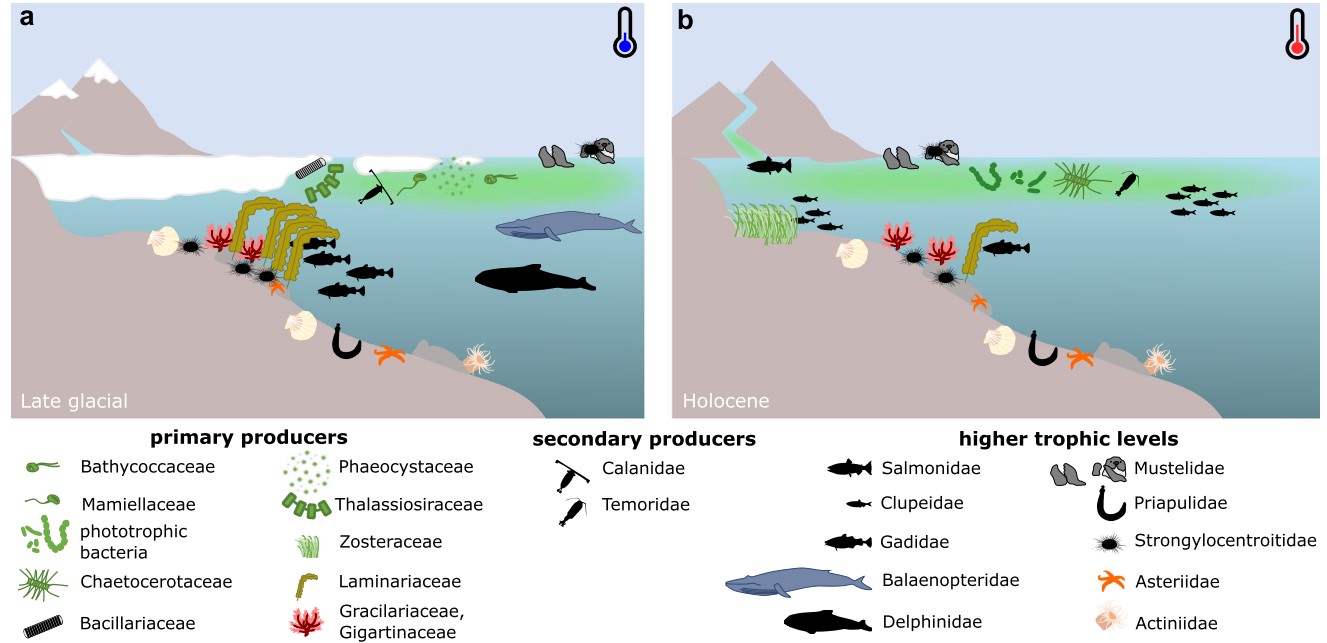

**Fig. 5 | Schematic illustrations of ecosystem change after postglacial sea-ice loss.** Representatives of functional groups of **a** the seasonal sea-ice ecosystem, which prevailed for most of the late glacial and **b** the ice-free ecosystem, which dominated during the Holocene.

and modern observations[2]. Our results thus confirm that ecosystem service changes related to future sea-ice loss is of major relevance not only for global climate but also for regional economies.

## Methods

### Sample material

In 2018 we collected and extracted samples from the archive half of a 9.05 m piston corer (SO201-2-12KL, Fig. 1) which was retrieved during RV Sonne cruise SO201 "KALMAR" in 2009[52] and since then stored at 4 °C. We applied the chronostratigraphy established by ref. [52]. Its high sedimentation rates with 80 cm kyr$^{-1}$ during the last deglaciation and 30 cm kyr$^{-1}$ in the Holocene allows us to detect ecosystem changes with a temporal resolution of, on average, 780 years according to our sampling density. According to our age model, the one-centimeter thick samples contain, on average, a contribution of DNA deposition over 15.5 years during the late glacial and over 28.7 years during the Holocene. The sampling was carried out at GEOMAR, Kiel, in a sedimentology laboratory devoid of any molecular biology work. Details about the subsampling procedure and the opening of the cores can be found in the Supplementary Information.

### DNA extraction

DNA extractions and subsequent library preparation were carried out in a dedicated paleogenetic facility devoid of any post-PCR laboratories, whereas the PCRs and downstream preparations for sequencing were carried out in the genetics laboratories located in a different building.

The DNeasy PowerMax Soil Kit (Qiagen, Germany) was used to extract total DNA from an average of 7.5 g sediment[80]. For a previous DNA metabarcoding analysis[20] we extracted DNA for 63 samples in seven batches, where each batch contained up to nine samples and an extraction blank (in total, seven negative controls). From those, we chose the stock solutions of 25 DNA extracts and the corresponding extraction blanks, which had been kept continuously frozen since the extraction. The sample choice led to a mean temporal resolution of ~780 years, thereby ensuring that each millennium is represented by at least one and each climatic phase by at least two samples.

We measured the DNA concentration on a Qubit 4.0 fluorometer (Invitrogen, Carlsbad, CA, USA) using the Qubit dsDNA BR Assay Kit (Invitrogen, USA) and depending on the initial DNA concentrations, we concentrated a volume of 300 or 600 µL using the GeneJET PCR Purification Kit (Thermo Scientific, USA) and eluted twice with 25 µL elution buffer. Immediately afterward, we checked the DNA yield again with 2 µL concentrated DNA using the Qubit dsDNA BR Assay Kit and the DNA was diluted to 3 ng µL$^{-1}$. The DNA extracts and aliquots were stored at −20 °C.

### Single-stranded library preparation and sequencing

For metagenomic shotgun sequencing, we chose the method of single-stranded library preparation, which was specifically developed for highly degraded ancient DNA[81,82] and used 15 ng DNA as a template. We prepared three batches of libraries for sequencing. The first batch contained a set of samples covering each major climatic phase of the core for a first assessment of the method. Therefore, five of the seven extraction blanks (50 µL per extraction blank were pooled and 5 µL of the pool used for the library preparation) were sequenced together with their corresponding samples in the first run, while the last two blanks were processed later in the second sequencing run. To monitor contamination during library preparation, we included one negative control for each batch of libraries (library blank).

The libraries were quantified using qPCR[82]: First, a pUC19 vector (New England BioLabs, Germany) was amplified by PCR. The PCR products were purified with the MinElute PCR Purification Kit (Qiagen, Germany) according to the manufacturer's instructions and subsequently quantified using the Qubit dsDNA BR Assay Kit. Three independent tenfold dilution series (109 to 102 copies per µL) were prepared to calculate the standard curves. The qPCR setup contained 1x Maxima™ SYBR™ Green (Thermo Scientific, Germany), 0.2 µM IS7 and IS8, and 1 µL of the libraries diluted 1:20 with TET buffer in a final volume of 25 µL. The qPCR was run in the RotorGeneQ cycler (Qiagen, Germany) with the following settings: 95 °C for 10 min, followed by 40 cycles of 30 s at 95 °C, 30 s at 60 °C and 30 s at 72 °C. The fluorescence was acquired after each cycle.

For samples, the subsequent double-indexing was performed by PCR in 10–14 cycles (for samples and blanks) depending on library concentration with indexed P5 (5′–3′: AATGATACGGCGACCACCG AGATCTACACNNNNNNNACACTCTTTCCCTACACGACGCTCTT; IDT, Germany) and P7 (5′–3′: CAAGCAGAAGACGGCATACGAGATNNN NNNNGTGACTGGAGTTCAGACGTGT, IDT, Germany) primers using AccuPrime Pfx polymerase (Life Technologies, Germany) and 24 μL of each library. Blanks were amplified with the same number of cycles, although the initial DNA concentration was 0 ng/μL. The indexing PCR products were purified using MinElute PCR Purification Kit (Qiagen, Germany). Then the library size distribution was checked on the Agilent TapeStation using the D1000 ScreenTape (Agilent Technologies, USA). The samples were pooled equimolarly, but with the negative controls in a 10:1 ratio to achieve a final molarity of 10 mM. Three paired-end sequencing (2 × 125 bp) runs were carried out with a modified sequencing primer CL72[82] (5′–3′: ACACTCTTTCCCTACACG ACGCTCTTCC, IDT, Germany) by the Fasteris SA sequencing service (Switzerland). The first library pool was sequenced on a single lane of an Illumina HiSeq 2500 instrument (high-output V4, total output of 40.6 GB), while the second and the third pools were sequenced on two lanes of a full SP flow cell (total output 198.1 GB) using the Illumina NovaSeq device.

## Bioinformatic processing and sequence classification
We used FastQC v. 0.11.9[83] to check the quality of the raw data, followed by duplicate removal with FastUniq v. 1.1[84], which reduced the number of read pairs from 918,186,452 to 876,349,245 (Supplementary Data 1). We additionally applied clumpify (BBMap package v. 38.87; sourceforge.net/projects/bbmap/) to perform deduplication. Both approaches resulted in comparable duplication rates. The average of sample and blank duplication rate for FastUniq is 5.3 and 10.6% for clumpify. The average duplication rate within samples is FastUniq = 4.2% and clumpify = 7.2%, when considering blanks only, the approaches identified a duplication rate of FastUniq = 9.5% and clumpify = 22.5 %, which indicates a better performance of clumpify to remove duplicates in the blanks.

Afterward we used Fastp v. 0.20.0[85] for trimming and merging of overlapping reads in parallel, which further reduced the total number of read pairs to 742,117,607. The reads were trimmed for quality, length residual adapters, and stretches of low complexity using a sliding window (settings: length_required = 30, cut_front, cut_tail, cut_window_size = 4, cut_mean_quality = 10, n_base_limit = 5, unqualified_percent_limit = 40, complexity_threshold = 30, qualified_quality_phred = 15, low_complexity_filter). Nucleotide stretches of at least 10 poly-x were trimmed at the tail of the reads (-q 10, -x 10). Overlapping reads were merged and the quality information was used for base pair correction (settings: correction, overlap_len_require = 10, overlap_diff_limit = 5, overlap_diff_percent_limit = 20).

For the taxonomic classification of reads, we used the k-mer-based classifier Kraken2 v. 2.0.8-beta[86] in single mode for merged and in the paired mode for unmerged reads (confidence threshold of 0.2). We downloaded the National Center for Biotechnology Information (NCBI) non-redundant nucleotide database (ftp://ftp.ncbi.nlm.nih.gov/blast/db/FASTA/nt.gz; downloaded in June 2021) and the NCBI taxonomy (obtained using kraken2-build in June 2021) for taxonomic classification and built the Kraken2 database.

Using R v. 4.0.3[87], we retrieved all entries assigned on the family level and combined merged and unmerged paired-read classifications. Not all classified reads were kept for further analyses. The sediment core was kept at 4 °C since 2009; hence we expected prokaryotic and fungal composition to be biased and possibly reflect storage conditions. As we are interested in organismal responses to past environmental conditions, we kept only families belonging to phototrophic bacteria, photo- and heterotrophic protists, marine macrophytes, and Metazoa of likely regional and aquatic origin

(among fish, we kept only those which occur in the subarctic North Pacific and the Bering Sea based on FishBase[88]). According to Fish-Base, the region is inhabited by marine Cyprinidae, yet it has previously been debated as contamination[22] and was detected in our blanks, which is why we removed it from our analyses. Among protists, we excluded parasites, mostly belonging to Apicomplexans, Euglenids, Amoebozoa, and Myxozoa, although we acknowledge the important roles of parasites in marine environments. Euglenids contain many free-living bacterivorous flagellates[89], yet the majority of sequences were assigned to the human pathogens *Trypanosoma* and *Leishmania*. Apicomplexans are highly diverse in marine ecosystems, but the NCBI reference database suffers from false annotations with widespread mistakes in human and animal parasites, like *Theileria*, *Babesia*, *Cryptosporidium* and Eimeriidae, which we predominantly recovered in our data[90].

Finally, we kept only families that occurred in at least three samples with at least ten counts and resampled the dataset 500 times to a pelagic sample effort of 6593 counts and a benthic sample effort of 1839 counts per sample to circumvent bias arising from different sequencing depths across the samples[91] to capture the main compositional trend and allow for stable correlation signals. This measure was effective in reducing the zero-inflation of our dataset for the correlation network analysis and the few affected taxa have no significant role within the network (Supplementary Fig. 12).

## Negative controls
After deduplication, quality filtering, and trimming of DNA reads in the blanks (see details in Supplementary Data 1), only 0.37 and 2.14% of the original read pairs were left. In reads classified to family level and lower, the negative controls showed a negligible number of read counts. Between 111 and 540 reads were assigned to blanks, summing up to 1519 reads (0.1%) in total (Supplementary Table 1). The majority of reads in blanks were assigned to Hominidae (28.1% of blanks) and bacteria (Sphingomonadaceae 10.5%, Alteromonadaceae 5.6%, Vibrionaceae 5.4%) (Supplementary Figs. 4, 5). A total of 98 families occurred with 1 read count in blanks and 38 families have up to 2 reads (Supplementary Fig. 5).

## Damage pattern analysis
Damage pattern analysis applied by the automated computational screening pipeline HOPS Version 0.34[92] supports the ancient origin of the analyzed DNA pool (Supplementary Fig. 6). HOPS can either run for merged or unmerged data, but it cannot combine both. For the analyses of ancient patterns, only those reads were considered which had at least one mismatch in the first five base pairs. According to this filter, we first analyzed ancient patterns for key taxonomic groups (eukaryotic and prokaryotic algae, pelagic fish, benthic crustacean) and excluded taxa (freshwater fish and marine bacteria) using the nt (non-redundant nucleotide) database (see Supplementary Information). The frequency of C to T substitutions of the reads in the first ten and the last ten positions are given in Supplementary Fig. 6a–c.

## Statistical analysis
All statistical analyses have been performed in R v. 4.0.3[87]. After resampling, we tested each of the pelagic and benthic taxa for temporal autocorrelation using acf from the stats package and concluded that our trends and correlation networks are not influenced by temporal autocorrelation (Supplementary Figs. 8, 9).

We calculated pairwise Spearman rank correlation coefficients (ρ) using corr.test including Benjamini–Hochberg correction[93] for multiple testing from the psych package v. 2.0.12[94], and those exceeding at least 0.4 (adjusted *p* value <0.1) were used for undirected network generation by the igraph packagev. 1.2.6[95] without self-loops and isolated nodes (Supplementary Data 2). Networks with more and less stringent p-value thresholds are shown in Supplementary Fig. 16.

We included only positive correlations, as negative correlations were rare and did not affect the structure of the network. Furthermore, the samples are temporally not sufficiently resolved to infer direct negative interactions such as exclusions, as usually, samples represent a time average of several years. For the scale-free, force-directed network layout, we used the Fruchtermann–Reingold method (layout_with_fr)[96].

The GCGM network was built using the package ecoCopula v. 1.0.2[27]. The stackedsdm function was used to build the stacked species regression model with interpolated SSTs and $IP_{25}$ values as covariates. Then a Gaussian copula graphical lasso was fitted to the co-occurrence data with the function cgr and a lambda of 0.51, which was determined empirically (Supplementary Fig. 13). To test the similarity between the Spearman network and the ecoCopula network, we computed the Jaccard-index over the edges in the networks and compared the results with a randomized null-model. The null-model was generated via double-edge swaps on the generated ecoCopula networks, which was performed ten times for each lambda value (shrinking parameter: with increasing lambda, the networks become less dense) between 0.1 and 1.0. Co-occurrence networks with negative associations for both Spearman and ecoCopula are presented in Supplementary Figs. 14, 15, respectively.

The $IP_{25}$ record was available for this core, whereas we used an independent stack of northeast Pacific SST records that covers the full interval of this study but could be influenced by regional processes, such as ice-sheet sourced meltwater pulses[97]. SST[97] and $IP_{25}$[98] data were interpolated to our sample ages using the R function approx[87]. As $IP_{25}$ data could not be extrapolated for the last three samples, we calculated the mean of the last three interpolated values and replaced the missing data based on regional sea-ice reconstructions for the LGM[12] (Supplementary Fig. 10). Then the interpolated variables were added to the sample-by-taxa matrix, and pairwise Spearman rank correlations were calculated as described above (Supplementary Table 3). Yet, we advise caution because of our limited sample size.

Furthermore, we highlight that both proxies contain biases. We, therefore, applied Spearman rank correlations following Djurhuus et al.[35] to identify SST- or $IP_{25}$-correlated families as described above (Supplementary Table 4). Then, we filtered out families which are linked with either SST- or $IP_{25}$-correlated families (neighboring nodes) using the adjacent_vertices function of igraph and the table function to build a contingency table with the abundance of each family or functional group as a neighboring node (Supplementary Fig. 11 and Supplementary Data 2). This was followed by a two-sided $t$-test using the t.test function of the stats package v. 4.0.3[87]. The stratigraphic diagrams, including the CONISS dendrogram, were prepared using packages rioja v. 0.9-26[99] and vegan v. 2.7-7[100].

### Reporting summary

Further information on research design is available in the Nature Portfolio Reporting Summary linked to this article.

## Data availability

The sequencing data generated in this study have been deposited at the European Nucleotide Archive (ENA) under Bioproject number PRJEB46821. The metadata and taxonomic count data on the family level used in this study are available at PANGAEA[101]. The cores are stored at the GEOMAR core repository in Kiel. For access, please contact co-author Dirk Nürnberg. Source data are provided with this paper. The sea-ice data of March 2020 were retrieved from https://www.meereisportal.de (Funding: REKLIM-2013-04). The Global 30 Arc-Second Elevation (GTOPO30) was retrieved from https://www.usgs.gov/centers/eros/science/usgs-eros-archive-digital-elevation-global-30-arc-second-elevation-gtopo30#overview[102]. Source data are provided with this paper.

## Code availability

The code used to process, filter, and analyse the sequencing data were available at https://github.com/ZimmermannHH/BeringSea_shotgun_sequencing/.

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

## Acknowledgements
We would like to thank Stefan Niehaus for bioinformatic support, Thomas Böhmer for generating the map, Janine Klimke for guidance with the library preparation, and Cathy Jenks for English proofreading and editing.

## Author contributions
H.H.Z.: conducting lab work, formal analysis, visualizations, and writing of the initial draft, K.R.S.-L.: formal analysis of damage patterns and conceptualization; V.D.: network similarity and ecoCopula networks; L.H.: software and formal analysis of damage patterns, L.S.: Methodology, M.-T.H.: supervision of network analyses, D.N.: Resources, R.T.: Resources, U.H.: conceptualization, study design, supervision, co-writing of the initial draft, and resources. All authors commented on the initial draft.

## Funding

## Competing interests
The authors declare no competing interests.
