## [Peer Review File · Nature Communications]

Marine ecosystem shifts with deglacial sea-ice loss inferred from ancient DNA shotgun sequencingREVIEWER COMMENTS

Reviewer #1 (Remarks to the Author):

Marine ecosystem shifts with deglacial sea-ice loss inferred from ancient DNA shotgun sequencing

Review by Linda Armbrrecht

The manuscript “Marine ecosystem shifts with deglacial sea-ice loss inferred from ancient DNA shotgun sequencing” describes a metagenomic analysis of sediment samples from Kamchatka (Arctic) over the last ~20,000 years. Revealed is a shift from a sea ice-adapted late-glacial ecosystem, characterised by diatoms, copepods, and codfish, to an ice-free Holocene characterised by cyanobacteria, salmon, and herring. The analyses combine shotgun metagenomics with network and correlation analysis. The manuscript has several strengths, including the dataset being unique, the methodology being robust, and the illustrations being of high quality. Weaknesses include the current structure of the results (parts of text that would be better placed in the introduction or discussion are scattered in amongst the results), and discussion (placing the results into the regional and temporal context). I have provided detailed suggestions for improvements below. Once these points are addressed I believe the manuscript will be a valuable addition to the literature and suitable for publication in Nature Communications.

Abstract:

L9: Cyanobacteria: no need to capitalise.

L10: I agree that it is an important new tool, but this sentence sounds like it is the only one, so suggest to change “the key” to “an important new tool/proxy/...” or similar.

L12: Remove “Regarding future ecosystem services”

L12/13: “continuing sea-ice decline on the northern Bering Sea shelf might pose a risk for changes in carbon export and benthic food supply” – simplify this sentence, eg.: might impact on/reduce carbon export and disrupt benthic ...?

Introduction:

The introduction touches on most of the concepts of the study that follows, however, it is missing some background information on the network analysis that follows, which is instead introduced in the results (L123 following), on the characteristics of pelagic and benthic communities (treated separately in the study) and environmental proxies such as IP25 and SSTUK37. All of these aspects are important parts of the study, however, are not introduced/mentioned in the Introduction (currently focussing on food web structure, short-term records and metagenomics).

L17: This sentence should be rephrased, as not all organisms can be considered a carbon sink, there are many organisms that respire, too, and so might be considered a carbon source.

L19: Be more specific on how exactly sea-ice loss effects these following parameters that you list, not just that they are related.

L30: I'm not sure if protists are the best example here, as there are many studies on nano- and microfossil assemblages which would have all been protists at the time they were alive. You could refer to organisms that are not normally part of conventional studies, in your study that may be copepods and fish.

L32: This reference needs correction in the reference list – I assume you are referring to the 2018 Special Report?

L49: replace “on” with “using”

L50: “Close to the coast” – can you add the approximate distance to the coast? Around 2,000 m depth does not sound very close, and this would be useful information for your discussion about downslope sediment transport later in the manuscript.

L55: “Our approach includes a broad spectrum of taxonomic and functional groups” – this sentence is unclear; change to “our metagenomic/shotgun approach enables the investigation of”

Fig. 1: Add yellow site marker also in Panels B and C, and in panel A capitalise “March”.

Results:

This section needs some focussing. In the current form there are several occasions where parts that should be in the Introduction or Discussion are intermingled with the description of the data and figures/tables, which makes this section difficult to follow. For example, there are discussion points in L77-79, L103-L106, L179, L182-187, L205-207, L213-217; and Introduction points in L123-126, L198-200.

Fig. 2 and Fig. 4. Pelagic and Benthic profiles; these are currently part of the sections “Taxonomic assignments to potentially pelagic organisms are dominated by primary producers and fish” and “Benthic sedaDNA sources and network dynamics”. While SST and IP25 measurements are shown as part of the figures, a more detailed description of the changes that occurred with varying SST/IP25 or on a temporal scale, i.e.. in relation to the deglacial/sea-ice loss through time, could be included.

L122: Environmental preference of pelagic taxa inferred by network analysis; this is a lengthy section, consider introducing subheadings to describe the sea-ice and ice-covered networks.

L130: “decadal to centennial”: You might have done this already, but have a good overview about sedimentation rates and that ~780 years lie between each depth sampled, based on this could you narrow down the estimate how many years would be covered within your sedaDNA sub-sample?

L150: Are you able to investigate these reads to see if they might be one of those two suspected species?

L197: I note that the network analysis for the benthic families is provided with the Supplementary Material, but if not limited in figure numbers it might be worth adding this figure to the main text.

Discussion:

Much of the discussion currently focuses on speculation about potential changes in carbon export resulting from changes in differently sized phytoplankton communities. While this is an important point that is worthy of discussion in this manuscript, I am missing some discussion of the findings of the study about changes in the pelagic and benthic communities with time, correlations with SST, IP25, and in relation to the discussion points that are currently part of the results (see above).

Methods:

L334: Add year of extraction.

L341: 2-4mL – the mL using makes me wonder if this was a sediment slush, was it diluted? If not I suggest to use cm³ or g as unit.

L350: change “with twice” to “twice with”

L362: Add after “two”: “blanks”

L486/487: Delete, this is already in the previous sentence.

L487/488: Add a reference.

Reviewer #2 (Remarks to the Author):

Review of MS NCOMMS-22-05048

1. Key results: Your overview of the key messages of the study, in your own words, highlighting what you find significant or notable. Usually, this can be summarized in a short paragraph.

MY COMMENTS:

This study brings the field of ancient environmental DNA (eDNA) forwards by presenting change in functional diversity of a wide range of marine organisms (notably prokaryotic and eukaryotic phytoplankton, macrophytes, fish, some marine mammals, but also potentially less described invertebrates) during the last 20,000 using a novel ancient eDNA approach on marine sediments (shotgun metagenomics). Correlation network analyses and correlations to paleoclimatic variables (SST, IP25) are used to associate taxonomic and functional groups to climate-driven change in sea-ice

conditions at the site. The main 'shifts' in ecosystem response to post-glacial sea-ice loss recorded by this novel approach lie in smaller-sized algal primary producers related to different C storage and export from the oceans and major fish community reorganization at higher trophic levels.

2. Validity: Your evaluation of the validity and robustness of the data interpretation and conclusions. If you feel there are flaws that prohibit the manuscript's publication, please describe them in detail.

MY COMMENTS:

103-106: While the taphonomic-based explanations provided are plausible causes for the discrepancy in relative abundance, I would think this difference is also likely due to database bias, where a number of groups e.g., invertebrates are relatively poorly represented in NCBI. Perhaps the authors have a better idea of genomic coverage of their organisms of interest to assess reference bias, please provide if possible.

I. 147: I could not find the read counts associated to copepods (rather %), please add.

I. 174: Warmer conditions seem to favor early-stage fish that, in the case of salmon at least, would occur in freshwater rivers, possibly in quite distant localities, correct? Do we know where the fish populations at the site breed at present? The evidence here is for SST, strictly speaking, which globally would be correlated to land temperatures of the breeding grounds, but supporting evidence/references would help make this point clearer.

I. 179: Do increased freshwater inputs bring more nutrients overall?

I. 186: Could N-fixation Cyanobacteria also be an advantage? Please elaborate.

Discussion: The evidence presented for a shift in phytoplankton size is quite convincing, but I would like to see some comments on the change in other functional aspects in primary producers that may offer alternative or complementary explanation on the impact of this shift in composition on the ecosystem e.g., N-fixation by cyanobacteria and role in the biogeochemical cycling.

I. 203-207: It is potentially interesting that such fewer reads were found from the benthic groups, but it would be best to discuss number of reads assigned from rarefied reads only, not both. Raw reads might differ in different samples based on library success, which could be confounded with e.g., sediment properties during different time periods (e.g., more abundant classified reads in Suppl. Fig. 1, 13-19 ka

yrs BP). The difference between benthic and pelagic reads would be more effectively presented by plotting a Pelagic:Benthic ratio along Figs 2 and 4. This part is also somewhat speculative, as a lot of questions remain as of the taphonomy of ancient DNA in sediments.

I. 299 How strong is the DNA signal for Copepods (or, as mentioned here, large Copepods)? I could not find the number of reads for the three Families presented in text and Fig. 5. This group is an important part of the foodweb as presented in Fig. 5, but I am not convinced that the strength of the signal for this group supporting the discussion points.

I. 475: Having control over sea-ice from the same sediment profile is a strong asset of this MS, which would be highlighted further by stating clearly early on in the MS and/or detailed in methods that these records are derived from the same core (if Fig. 2, add also intext). However, the lack of it for the first and I.475 'last two for SST and the last four for IP25 data' samples (here there actually seem to be an offset with the data presented in Figs. 2 and 4, where SST and IP25 records appear longer by 1 sample or so) limit the conclusions, particularly for the early phase. The SST record available in the Northeast Pacific appears to have incursions into warm conditions at the very base of the record, which could correspond to 'ice-free conditions', or perhaps not well represented by the averaging of the closest samples (same applies for the top-most sample). Please provide further justification for the extrapolation, or remove samples from statistical analyses with SST and IP25 for these intervals and adjust conclusions accordingly.

3. Significance: Your view on the potential significance of the conclusions for the field and related fields. If you think that other findings in the published literature compromise the manuscript's significance, please provide relevant references.

MY COMMENTS:

I am not familiar with metabarcoding studies of ancient eDNA from marine sediments, however to my knowledge this could be one of the first applications of shotgun metagenomics in such context, at once providing a paleoecological record of the largest breath of lifeforms over the longest period. The strong record for fish, in particular, is a major advance for the marine-based and sedaDNA field, which is also of potential great significance for evaluating long-term changes in fish stocks and fish ecological and evolutionary dynamics in the future, which remain difficult to quantify in oceans today.

Other papers using shotgun sequencing on marine sediments that I am aware of: Gaffney et al 2020 Geosciences, who presented plant sedaDNA from a tsunami sediment deposits, which did not aim to reconstruct paleoecological dynamics. Eukaryote composition from shotgun metagenomics were also compared to metabarcoding data from Santa Barbara Basin deep-sea sediments over the Holocene, mainly focusing on diatoms (Armbrecht et al 2021 ISME).

Furthermore, the authors go beyond listing taxa and provide an elegant and straightforward statistical analysis of their molecular data against environmental variables by using correlation networks (please see comments below). Altogether, their novel approach allows them to measure the effects of (long-term) climate change on important ocean processes like primary production and food-web interactions at an unprecedented level of detail and taxonomic breadth. The relevance of this work as 'analogous' to ongoing sea-ice loss trends in subarctic coastal environments that is discussed by the authors in Discussion could be emphasized in the introduction already by describing how the 'seasonal sea-ice' to 'sea-ice free' conditions over the period studied have changed and relate to documented ongoing and expected e.g., IPCC trends for relevant regions.

4. Data and methodology: Your assessment of the validity of the approach, the quality of the data, and the quality of presentation. We ask reviewers to assess all data, including those provided as supplementary information. If any aspect of the data is outside the scope of your expertise, please note this in your report or in the comments to the editor. We may, on a case-by-case basis, ask reviewers to check code provided by the authors (see this Nature editorial for more information).

Reviewers have the right to view the data and code that underlie the work if it would help in the evaluation, even if these have not been provided with the submission (see this Nature editorial). If essential data are not available, please contact the editor to obtain them before submitting the report.

MY COMMENTS:

Outliers: At least three (1.08, 5.6 and 9.84, cal ka yrs) of the samples presented seem to be outliers in the dataset (own branches on dendrogram Figs 2 and 4, very distant from neighbours), and I suspect the low raw, classified pre- and post- read counts exclusion of taxa and resampling may explain this as an artifact, rather than a true signal. This could be showed by presenting e.g., a PCoA of the samples (with and without controls) with a biplot of the main taxa before and after exclusion of taxa in Suppl. I think removing them would change correlations and interpretations slightly (likely strengthen them), and help present the counts from the Figures 2 and 4 on clearer scales. Please provided the data used to plot Suppl Fig 1 in table form, and please also provide read counts before and after exclusion of taxa (described on l. 409-424).

All figure captions and tables titles: specify whether reads presented are rarefied or not.

Fig. 2: Please consider using the same scale for rel. abundances. At present it is difficult to see groups most recorded and transitions. Also, as they are re-sampled and not actual read-counts, I suggest presenting into percentages for ease of comparisons, also to be more consistent with numbers reported in text.

Figs 2 and 4: Boxes for the ice-free ecosystem are different, which one is correct (Fig. 4)?

I. 83: 72.5% of classified read pairs overlapped and were merged. Is the total read number on I.76 including overlapping read pairs (reads from 'the same DNA strand' counted twice?). Please clarify this elsewhere also, it's important that the merged reads are not counted twice.

I. 139: Consider marking the Families (significantly) positively correlated to SST and IP25 in red and blue in Figs 2 and 4 for ease of interpretation.

I. 142: Add climatic periods mentioned in text to Figs 2 and 4.

I. 190, Fig. 3: Nice figure, well thought out and functional groups easy to distinguish. Please add legend for equivalent % for size of nodes (same applies to other network provided in suppl. materials). Add explanation to caption for labelling of nodes or not (some nodes don't have labels, likely because they have too low %). Some family names are difficult to read and associate to the nodes, consider placing them more off from one another with black lines to the nodes, include fewer names or replace with images of taxa as per Fig 5 where appropriate.

I. 322 The chronology was reported in a 2012 publication; please indicate whether the same, 2009 core was used, and if so, indicate the date or year and procedure for the splitting of the core that was used for the DNA work. Cores stored unopened vs opened for many years prior to DNA sub-sampling may be differently affected by secondary growth of e.g., fungi, yeast and Bacteria that would influence DNA results (as per stated by the authors). If different cores were used, please describe the splitting procedure and how those were correlated for the age-depth model.

I. 327 As the lab facilities used for sub-sampling are not a dedicated clean facilities, please include the precautions that were undertaken to limit contaminants (air, surfaces, tools) from the wide range of organisms studied at GEOMAR, like fish and phytoplankton.

I. 333 The Epp et al 75 is wrong, it is likely the authors meant 'Epp, L. S., Zimmermann, H. H., & Stoof-Leichsenring, K. R. (2019). Sampling and extraction of ancient DNA from sediments. In B. Shapiro, A. Barlow, P. D. Heintzman, M. Hofreiter, J. L. A. Paijmans, & A. E. R. Soares, (Eds.), *Ancient DNA: Methods and protocols* (pp. 31–44). Springer New York.' rather than 'Epp, L. S. A global perspective for biodiversity history with ancient environmental DNA. *Molecular Ecology* 28, 2456–2458 (2019).'

I. 345: Please clarify in text whether the same extracts as ref 15 were used (I assume this is the case, as the same extract controls are analysed), or whether new extracts were made from the samples. If the same extracts were used, please clarify that the protocol is exactly as per ref 15 from the first sentence of this paragraph.

I. 376: Were the controls also provided with 10-14 cycles? This may explain why such few reads could be sequenced.

I.390. A 5% replication rate seems impressively low; is this in a similar range from other sedaDNA studies? Could the authors provide the distribution from Suppl. Fig. 1 for all samples in table form? Apologies if I have missed this.

I. 409-424: I think the approach for inclusion/exclusion of taxa is reasonable, however it would be useful to include the read counts that were removed by this 'exclusion step' from the initial read count at the classification at Family level step. Pls see additional comment on damage patterns.

I. 425 Though I acknowledge that similar threshold approaches have been used on similar datasets and at least partly agree with the reasoning behind it, keeping 'only families that occurred in at least 3 samples with at least 10 counts' prior to re-sampling (or other form of normalization) increases the risk of removing true positives, rare taxa that may by chance (or because of sediment matrix effects on library building success) occur only in shallower sampled intervals of the core. Could the authors comment on this effect in relation to their data?

Negative controls: It is good practice that the authors included a number of negative controls from the extraction step in the sequencing, however the number of reads from them is quite low (total of ~500 classified reads from all controls combined). It is actually common for controls to yield relatively high levels of reads assigned compared to samples, because human-related taxa are overall more commonly sequenced and thus more represented in reference databases (erroneously or not), and it would be also important to know many raw reads were sequenced for controls (maybe I have missed this information). Please provide the read counts after the different bioinformatic steps for the controls as per the samples. Ideally, one would provide similar sequencing effort to controls as for samples, however such a standard has not yet been established in the field and 'best practice' approaches from previous works have varied, none reaching this ideal (to my knowledge). Furthermore, it can be challenging to obtain sufficient amounts of reads from controls for sequencing, that would indeed contain very little DNA (increasing the number of indexing PCR amplification cycles usually helps, but also result in high levels of duplication). That being said, I do not have a reason to dispute the authenticity of the results provided for the main taxa detected supporting the MS conclusions other than the relatively minor verifications I have pointed out elsewhere, particularly if the damage patterns from at least some of the representatives of major groups of organisms can be established.

l. 444: Insufficient number of reads assigned were found for the damage pattern analyses, however pelagic and benthic merged and unmerged counts sum up to potentially higher reads counts than merged and unmerged read counts for some samples. Please clarify.

5. Analytical approach: Your assessment of the strength of the analytical approach, including the validity and comprehensiveness of any statistical tests. If any aspect of the analytical approach is outside the scope of your expertise, please note this in your report or in the comments to the editor.

MY COMMENTS:

I'm not very familiar with correlation networks, but I find the use here to be useful in highlighting differences in community assemblages and function under different climatic regimes. I would like for the authors to address or comment on a couple of questions that would relate to the structure of ancient metagenomic datasets could be addressed in the MS: What is the sensitivity of correlation networks to detect non-linear relationships between taxa, especially over centennial to millennial timescales? Do biogeographical distribution patterns matter on this scale? How is this statistical approach handling the zero-inflated dataset? Trends in the dataset (vs environmental variables) could perhaps be further explored with GAMs (e.g., Simpson, G. L. Modelling palaeoecological time series using generalised additive models. *Frontiers in Ecology and Evolution* 149 (2018). To what extent do the correlations between families may be affected by reference bias within closely related taxa, where for instance, only a few members of a family with differential degree of coverage may be available for mapping in NCBI? How could this affect interpretations (e.g., l. 155)?

Could the author also comment on the potential impact of downcore autocorrelation between samples for correlation networks and overall correlations with environmental variables to strengthen the conclusions?

All relative abundance datasets suffer from closed sum effects. To further support one of the main conclusions of the MS about the change between smaller and larger celled primary producers from the pelagic environment, the authors could test the strength of the correlation between the ratio of read abundances of small:large phytoplankton and their environmental variables.

6. Suggested improvements: Your suggestions for additional experiments or data that could help strengthen the work and make it suitable for publication in the journal. Suggestions should be limited to the present scope of the manuscript; that is, they should only include what can be reasonably addressed

in a revision and exclude what would significantly change the scope of the work. The editor will assess all the suggestions received and provide additional guidance to the authors.

MY COMMENTS:

I have already provided suggestions in previous sections that would, in my opinion, strengthen the manuscript, and added a few more below.

A major point: damage patterns were assessed for diatoms, but overall conclusions would be stronger if the other major groups found were also assessed like fish (codfish, salmon, herring), cyanobacteria and copepods (the latter may have too low counts). Cnidaria, Echinodermata, Mollusca also had potentially sufficiently high number of reads. Where counts are of sufficient abundance, consider also assessing the damage patterns for some of the main groups that were excluded to show the difference in damage. Because it is particularly prone to reference bias because of its high representation in NCBI as a model organism, to further validate the signal of Salmon (vs that of other fish), the authors could consider providing information on whether reads fall within unique parts of its genome (vs other fish).

Presenting rarefaction curves for each sample and for both pelagic and benthic groups in supplementary would help assessing sampling completeness at the Family level.

7. Clarity and context: Your view on the clarity and accessibility of the text, and whether the results have been provided with sufficient context and consideration of previous work. Note that we are not asking for you to comment on language issues such as spelling or grammatical mistakes.

MY COMMENTS:

The manuscript was a nice read, and I have made only a few suggestions that may improve clarity and context for the wider readership.

Intro: Consider providing a bit more context to support the validity of 'climate transition' and the analogous environments compared, by establishing the current, historical and paleo sea-ice conditions, and those expected under future scenarios more clearly (e.g., how many months of sea ice? Extent? spatial patterns?). How do the 'the phases of the Bølling-Allerød and the Holocene' compare with now/expected scenario in relevant regions? This relates to my last comment in the significance section.

I. 26: This sentence: 'This demonstrates the need for a better understanding of ecosystem changes during the transition from seasonal sea-ice to ice-free conditions due to ongoing climate warming.' I'm actually not certain that 'it' does. Perhaps restructure this part of the intro for better flow.

Minor comments:

I. 4: 'service' is phrased wrong, should be 'services delivered by' or something along those lines.

I. 20. 'timing', do you mean seasonal timing?

I. 21: 'play a central role in trophic interactions' could be more specific by adding in what way they play a central role e.g., 'supporting foodwebs as primary producers' or food source etc.

I. 22: Try to be more specific for the wide readership: 'air temperatures' I assume could be written as e.g., 'high air temperatures' or 'record high air temperatures' if that applies?

I. 24-26: What were those shifts? Could be more specific here.

I. 32: Add 'latest' IPCC report

I. 216: Please add the taxa groups names (not only common names), as per above in text.

I. 233: specify whether observations were from paleo or modern datasets.

8. References: Your view on whether the manuscript references previous literature appropriately.

MY COMMENTS:

References seemed appropriate and up to date.

9. Your expertise: Please indicate any particular part of the manuscript, data or analyses that you feel is outside the scope of your expertise, or that you were unable to assess fully.

MY COMMENTS:

My expertise lies in paleoecology of Arctic freshwater ecosystems and foodwebs, paleoclimatology as well as shotgun metagenomics, and is therefore more limited for marine environments and organisms beyond characteristics that would apply to aquatic ecosystems more broadly, like major ecological functional groups and foodweb interactions.

Alexandra Rouillard

Reviewer #3 (Remarks to the Author):

The manuscript entitled: "Marine ecosystem shifts with deglacial sea-ice loss inferred from ancient DNA shotgun sequencing" suggest that the compositions and structure of a marine ecosystem is different between a period with and without sea ice in the Bearing Sea by looking at fossil record data.

I found the study interesting, but I had concerns about the methodology used especially with regards to the statistical analyses performed in the study. I listed my concerns about the statistical analyses below.

1. L474-476: "As for both variables the first sample age as well as the last two for SST and the last four for IP_25 data could not be interpolated, we calculated the mean of the first and last three interpolated values and replaced the missing data."

Why was it not possible to interpolate these samples? Please give detailed explanations.

Also, by using "...the mean of the first and last three interpolated values..." as a way to impute the missing data, the imputed missing values could lead to misinterpretation of the results. This is especially true because there are so few samples in the data. To put it differently, for IP_25, 20% of the data (5 out of 25 samples) are imputed. In this context, using an imputation approach that accounts for temporal autocorrelation (e.g. an Autoregressive integrated moving average [ARIMA] model) would be much

more appropriate. In addition, the model used for the imputation should be validated using SST and IP_25 data gathered elsewhere to check if the trend makes sense.

2. Currently, the analysis carried out suppose that each sample is completely independent from the others. In other words, it assumes that there are no temporal autocorrelations in the data. A test of temporal autocorrelation in the taxon data should be carried out to make sure the interpretation is not affected by temporal autocorrelation. If there is temporal autocorrelation in the data, it will need to be accounted for in the analysis and interpretation. If there is no temporal autocorrelation in the taxon data, it would be an interesting addition to the results that is valuable ecologically because it would refine the understanding gain from this system (e.g. a deeper understand of temporal turnover, or temporal beta diversity, could be assessed). There exist a few options to test for temporal autocorrelation for multivariate data, two that come to mind is to use the approach proposed by Legendre and Gauthier (2013) or Dray et al. (2008) (both proposed techniques have been implemented in the *adespatial* R package). Another approach is to test the temporal autocorrelation for each taxon independently, this could be done, e.g. with a Moran's I (Moran 1948) or an ARIMA model.

3. L463-466: "We calculated pairwise Spearman rank correlation coefficients (ρ) using `corr.test` including Benjamini-Hochberg correction for multiple testing from the `psych` package v. 2.0.1291, and those exceeding at least 0.4 (adjusted p-value < 0.05) were used for undirected network generation by the `igraph` package v. 1.2.692 without self-loops and isolated nodes."

Technically, building the network based on Spearman correlation is problematic because it does not explicitly account for the multivariate nature of the data and as such it is likely to lead to biased results and as such wrong interpretations. In this respect, the approach proposed by Popovic et al. (2019) is much better adapted to the question and data of the present study.

4. L467-468: "We included only positive correlations as negative correlations were rare and did not affect the structure of the network."

Negative correlations should be presented and discussed. They present valuable information that could lead to a better understanding of the studied system.

Reference

Dray, S., D. Bauman, F. G. Blanchet, D. Borcard, S. Clappe, G. Guenard, T. Jombart, G. Larocque, P. Legendre, N. Madi, and H. H. Wagner. 2021. *adespatial: Multivariate Multiscale Spatial Analysis*.

Dray, S., S. Said, and F. Debias. 2008. Spatial ordination of vegetation data using a generalization of Wartenberg's multivariate spatial correlation. *Journal of Vegetation Science* 19:45–56.

Legendre, P., and O. Gauthier. 2014. Statistical methods for temporal and space–time analysis of community composition data. *Proceedings of the Royal Society B: Biological Sciences* 281:20132728.

Moran, P. A. P. 1948. The interpretation of statistical maps. *Journal of the Royal Statistical Society Series B-Statistical Methodology* 10:243–251.

Popovic, G. C., D. I. Warton, F. J. Thomson, F. K. C. Hui, and A. T. Moles. 2019. Untangling direct species associations from indirect mediator species effects with graphical models. *Methods in Ecology and Evolution* 10:1571–1583.

Point-by-point response to the reviewers

REVIEWER COMMENTS

Reviewer #1 (Remarks to the Author):

Marine ecosystem shifts with deglacial sea-ice loss inferred from ancient DNA shotgun sequencing

Review by Linda Armbrrecht

The manuscript “Marine ecosystem shifts with deglacial sea-ice loss inferred from ancient DNA shotgun sequencing” describes a metagenomic analysis of sediment samples from Kamchatka (Arctic) over the last ~20,000 years. Revealed is a shift from a sea ice-adapted late-glacial ecosystem, characterised by diatoms, copepods, and codfish, to an ice-free Holocene characterised by cyanobacteria, salmon, and herring. The analyses combine shotgun metagenomics with network and correlation analysis. The manuscript has several strengths, including the dataset being unique, the methodology being robust, and the illustrations being of high quality. Weaknesses include the current structure of the results (parts of text that would be better placed in the introduction or discussion are scattered in amongst the results), and discussion (placing the results into the regional and temporal context). I have provided detailed suggestions for improvements below. Once these points are addressed I believe the manuscript will be a valuable addition to the literature and suitable for publication in Nature Communications.

Abstract:

Reviewer Comment: L9: *Cyanobacteria: no need to capitalise.*

Our response: Changed accordingly.

Revised text L27-29: We traced shifts from a sea ice-adapted Late-glacial ecosystem, characterized by diatoms, copepods, and codfish to an ice-free Holocene characterized by cyanobacteria, salmon, and herring.

Reviewer Comment: L10: *I agree that it is an important new tool, but this sentence sounds like it is the only one, so suggest to change “the key” to “an important new tool/proxy/...” or similar.*

Our response: Changed accordingly.

Revised text L29-32: “By providing information of marine ecosystem dynamics across a broad taxonomic spectrum, our data show that ancient DNA will be an important new tool in identifying long-term ecosystem responses to climate transitions for improvements of ocean and cryosphere risk assessments.”

Reviewer Comment: L12: *Remove “Regarding future ecosystem services”*

Our response: Removed.

Revised text L32-34: “We conclude that continuing sea-ice decline on the northern Bering Sea shelf might impact on carbon export and disrupt benthic food supply and could allow for a northward expansion of salmon and Pacific herring.”

Reviewer Comment: L12/13: “continuing sea-ice decline on the northern Bering Sea shelf might pose a risk for changes in carbon export and benthic food supply” – simplify this sentence, eg.: might impact on/reduce carbon export and disrupt benthic ...?

Our response: Changed accordingly.

Revised text L32-34: “We conclude that continuing sea-ice decline on the northern Bering Sea shelf might impact on carbon export and disrupt benthic food supply and could allow for a northward expansion of salmon and Pacific herring.”

Introduction:

Reviewer Comment: *The introduction touches on most of the concepts of the study that follows, however, it is missing some background **information on the network analysis** that follows, which is instead introduced in the results (L123 following), on the **characteristics of pelagic and benthic communities** (treated separately in the study) and **environmental proxies** such as IP25 and SSTUK37. All of these aspects are important parts of the study, however, are not introduced/mentioned in the Introduction (currently focussing on food web structure, short-term records and metagenomics).*

Our response: We included background information on (1) network analysis, (2) pelagic and benthic taxa and (3) environmental proxies in the introduction. The introductory parts about the network analysis were moved from the material and methods sections to the introduction as well as the interpretations of the IP₂₅ and SST proxies.

Revised text to include networks in the introduction L87-103: “Correlation networks have been used in the interpretation of multidimensional data by means of co-occurrence networks and contributed to our understanding of ecosystems by identifying, for example, habitat preferences in aquatic bacterial communities²³ or geographic co-occurrence patterns²⁴. The sparsity of metagenomic datasets resulting from either true absences or insufficient sequencing depth can lead to spurious correlations²⁵. Gaussian copula graphical models (GCGMs) are a novel statistical framework developed to separate environmental effects from intrinsic interactions among taxa²⁶. They are suited to address the compositional structure of sequencing data, non-linear correlations, and to remove spurious correlations via mediator taxa (if a correlation between two taxa arises only because both are correlated to a third taxon) or via similar responses to environmental effects (covariance). Co-occurrence networks have not yet been applied to *secaDNA* data before. As *secaDNA* samples contain averaged information on decadal to centennial time-scale (here on average 15.5 years during the late glacial and 28.7 years during the Holocene in a centimeter thick sample), correlation in the network should not be interpreted as ecological interactions between linked taxa or actual co-occurrence on a short-time scale. We here assume that positively correlated families show similar responses to environmental changes (i.e., sea ice coverage and SSTs).”

L201-202: “To identify taxa that show a similar response to sea ice loss, we used co-occurrence networks and compared Spearman networks with GCGMs.”

Included citations:

23. Comte, J., Lovejoy, C., Crevecoeur, S. & Vincent, W. F. Co-occurrence patterns in aquatic bacterial communities across changing permafrost landscapes. *Biogeosciences* **13**, 175–190 (2016).

24. Araújo, M. B., Rozenfeld, A., Rahbek, C. & Marquet, P. A. Using species co-occurrence networks to assess the impacts of climate change. *Ecography* **34**, 897–908 (2011).
25. Machado, M. S. *et al.* Network analysis methods for studying microbial communities: A mini review. *Computational and Structural Biotechnology Journal* **19**, 2687–2698 (2021).
26. Popovic, G. C., Warton, D. I., Thomson, F. J., Hui, F. K. C. & Moles, A. T. Untangling direct species associations from indirect mediator species effects with graphical models. *Methods in Ecology and Evolution* **10**, 1571–1583 (2019).

Revised text pelagic and benthic taxa in the introduction L40-43:

“This will likely change the seasonal timing, biomass, and composition of algal blooms, which play a central role in trophic interactions by supporting food webs, including benthic communities as primary producers².”

L 63-65:

“While pelagic communities are strongly linked to hydrographic factors and climate, benthic communities are strongly dependent on primary production and sinking particles from the water column above⁹.”

L 55-58:

“Despite extensive monitoring efforts, we still have scarce knowledge on long-term ecosystem responses to climate transitions for many taxonomic groups, particularly zooplankton, fish and benthic organisms such as clams, tunicates, starfish or macrophytes.”

L 60-63:

“The short-term responses of the Bering Sea ecosystem to warmer and colder regimes have been documented well^{2,5,8}, yet long-term rearrangements of the organismal composition in pelagic and benthic communities are not clear and more data are needed to constrain the future ecosystem development.”

Revised text environmental proxies in the introduction L 69-75: “Marine sediments are a natural archive of climate history from which sea ice can be reconstructed via proxy records, such as from biomarkers¹⁰ or microfossil remains¹¹. A regional palaeoceanographic framework for past sea ice coverage in the Bering Sea is based on reconstructions using the highly branched isoprenoid alkane biomarker IP₂₅¹², produced by diatoms bound to a life in seasonal sea ice¹⁰, and diatom microfossils¹¹. Alkenone-based sea surface temperatures (SST_{UK37}), which are produced by haptophytes, and can add complementary information about SSTs of the late summer/early fall season¹³.”

L 108-113:

“The paleoenvironmental backbone is based on previous multi-proxy reconstructions of the coring site derived from this core, including diatom microfossils and the biomarker IP₂₅, which show that the coring site was covered by seasonal sea-ice during the Last Glacial Maximum (LGM), Heinrich Stadial 1, and the Younger Dryas, while the phases of the Bølling-Allerød and the Holocene were predominantly sea-ice free^{12,16}. “

Citations:

9. Grebmeier, J. M. & Barry, J. P. The influence of oceanographic processes on pelagic-benthic coupling in polar regions: A benthic perspective. *Journal of Marine Systems* **2**, 495–518 (1991).

10. Belt, S. T. & Müller, J. The Arctic sea ice biomarker IP₂₅: a review of current understanding, recommendations for future research and applications in palaeo sea ice reconstructions. *Quaternary Science Reviews* **79**, 9–25 (2013).
11. Matul, A. G. Probable limits of sea ice extent in the northwestern Subarctic Pacific during the last glacial maximum. *Oceanology* **57**, 700–706 (2017).
12. Méheust, M., Stein, R., Fahl, K. & Gersonde, R. Sea-ice variability in the subarctic North Pacific and adjacent Bering Sea during the past 25 ka: new insights from IP₂₅ and UK'37 proxy records. *Arktos* **4**, 8 (2018).
13. Harada, N., Shin, K. H., Murata, A., Uchida, M. & Nakatani, T. Characteristics of alkenones synthesized by a bloom of *Emiliania huxleyi* in the Bering Sea. *Geochimica et Cosmochimica Acta* **67**, 1507–1519 (2003).
16. Méheust, M., Stein, R., Fahl, K., Max, L. & Riethdorf, J.-R. High-resolution IP₂₅-based reconstruction of sea-ice variability in the western North Pacific and Bering Sea during the past 18,000 years. *Geo-Marine Letters* **36**, 101–111 (2016).

Reviewer Comment: L17: *This sentence should be rephrased, as not all organisms can be considered a carbon sink, there are many organisms that respire, too, and so might be considered a carbon source.*

Our response: We rephrased the sentence.

Revised text L 37-40: “The organismal composition in high-latitude oceans is highly vulnerable to anthropogenic global warming, which may alter ecosystem services (e.g., food supply, biological carbon pump) due to sea-ice loss and related effects on rising ocean temperature, increasing light transmission, stronger water column stratification and decreasing nutrient availability¹.”

Reviewer Comment: L19: *Be more specific on how exactly sea-ice loss affects these following parameters that you list, not just that they are related.*

Our response: We added the information.

Revised text L 37-40: “The organismal composition in high-latitude oceans is highly vulnerable to anthropogenic global warming, which may alter ecosystem services (e.g., food supply, biological carbon pump) due to sea-ice loss and related effects on rising ocean temperature, increasing light transmission, stronger water column stratification and decreasing nutrient availability¹”

Reviewer Comment: L30: *I'm not sure if protists are the best example here, as there are many studies on nano- and microfossil assemblages which would have all been protists at the time they were alive. You could refer to organisms that are not normally part of conventional studies, in your study that may be copepods and fish.*

Our response: Changed accordingly.

Revised text L 55-58: “Despite extensive monitoring efforts, we still have scarce knowledge on long-term ecosystem responses to climate transitions for many taxonomic groups particularly zooplankton, fish, non-fossilizing algae, and benthic organisms such as tunicates, starfish or macrophytes.”

Reviewer Comment: L32: *This reference needs correction in the reference list – I assume you are referring to the 2018 Special Report?*

Our response: As per reviewer 2 we updated to the latest IPCC report and followed the recommended citation style.

Revised citation: “Intergovernmental Panel on Climate Change (IPCC). The Ocean and Cryosphere in a Changing Climate: Special Report of the Intergovernmental Panel on Climate Change. (Cambridge University Press, 2022). doi:10.1017/9781009157964.”

Reviewer Comment: L49: *replace “on” with “using”*

Our response: Replaced.

Revised text L104-107: “Here, we explore the potential of *sedaDNA* metagenomic shotgun sequencing to reveal shifts in ecosystem composition in response to postglacial sea-ice loss using a sediment core (SO201-2-12KL, 53.993°N, 162.37°E; water depth 2,173 m), which was taken on the western continental slope of Kamchatka about 70 km from the coast of Kamchatka (Fig. 1).”

Reviewer Comment: L50: *“Close to the coast” – can you add the approximate distance to the coast? Around 2,000 m depth does not sound very close, and this would be useful information for your discussion about downslope sediment transport later in the manuscript.*

Our response: We removed “close to the coast” and added the approximate distance.

Revised text L 104-107: “Here, we explore the potential of *sedaDNA* metagenomic shotgun sequencing to reveal shifts in ecosystem composition in response to postglacial sea-ice loss using sediment core SO201-2-12KL, which was taken on the western continental slope of Kamchatka about 70 km from the coast of Kamchatka (Fig. 1).”

Reviewer Comment: L55: *“Our approach includes a broad spectrum of taxonomic and functional groups” – this sentence is unclear; change to “our metagenomic/shotgun approach enables the investigation of”*

Our response: We rephrased this sentence.

Revised text L113-116: “Our approach includes a broad spectrum of taxonomic and functional groups particularly among Eukaryotes, which we consider a major step towards deciphering the consequences of climate transitions for marine ecosystem structure.”

Reviewer Comment: *Fig. 1: Add yellow site marker also in Panels B and C, and in panel A capitalise “March”.*

Our response: We changed the figure accordingly. We left the site markings small, so that they do not obscure the map background too much.

Revised Figure 1:

Results:

Reviewer Comment: *This section needs some focussing. In the current form there are several occasions where parts that should be in the Introduction or Discussion are intermingled with the description of the data and figures/tables, which makes this section difficult to follow. For example, there are discussion points in L77-79, L103-L106, L179, L182-187, L205-207, L213-217; and Introduction points in L123-126, L198-200.*

Our response: We moved parts of the results and material and methods to the introductions. However, we would like to keep the discussion close to the results. To avoid confusion we merged the result and discussion section and added subchapters. For overall conclusions we now use the title “Implications for sea ice decline in the northern Bering Sea”.

Moved and revised text from results/material and methods to introduction L87-103:

“Correlation networks have been used in the interpretation of multidimensional data by means of co-occurrence networks and contributed to our understanding of ecosystems by identifying, for example, habitat preferences in aquatic bacterial communities²⁴ or geographic co-occurrence patterns²⁵. The sparsity of metagenomic datasets resulting from either true absences or insufficient sequencing depth can lead to spurious correlations²⁶. Gaussian copula graphical models (GCGMs) are a novel statistical framework developed to separate environmental effects from intrinsic interactions among taxa²⁷. They are suited to address the compositional structure of sequencing data, non-linear correlations, and to remove spurious correlations via mediator taxa (if a correlation between two taxa arises only because both are correlated to a third taxon) or via similar responses to environmental effects (covariance). Correlation networks have not yet been applied to *sed*aDNA data

before. As sedaDNA samples contain averaged information on decadal to centennial time-scale (here on average 15.5 years during the late glacial and 28.7 years during the Holocene in a centimeter thick sample), correlation in the network should not be interpreted as ecological interactions between linked taxa or actual co-occurrence on a short-time scale. We here assume that positively correlated families show similar responses to environmental changes (i.e., sea ice coverage and SSTs)."

L 69-75:

"Marine sediments are a natural archive of climate history from which sea ice can be reconstructed via proxy records, such as from biomarkers¹⁰ or microfossil remains¹¹. A regional palaeoceanographic framework for past sea ice coverage in the Bering Sea is based on reconstructions using the highly branched isoprenoid alkane biomarker IP₂₅¹² which is produced by diatoms that are bound to a life in sea ice¹⁰ and diatom microfossils¹¹. High values of IP₂₅ suggest seasonal sea-ice and ice-edge processes, while its absence could indicate either permanent sea-ice coverage or dominantly ice-free conditions. Alkenone-based sea surface temperatures (SST_{UK37}), which are produced by haptophytes, and can add complementary information about SSTs of the late summer/early fall season¹³."

L 108-113:

"The paleoenvironmental backbone is based on previous multi-proxy reconstructions of the coring site derived from this core, including diatom microfossils and the biomarker IP₂₅, which show that the coring site was covered by seasonal sea-ice during the Last Glacial Maximum (LGM), Heinrich Stadial 1, and the Younger Dryas, while the phases of the Bølling-Allerød and the Holocene were predominantly sea-ice free^{12,16}."

Reviewer Comment: *Fig. 2 and Fig. 4. Pelagic and Benthic profiles; these are currently part of the sections "Taxonomic assignments to potentially pelagic organisms are dominated by primary producers and fish" and "Benthic sedaDNA sources and network dynamics". While SST and IP25 measurements are shown as part of the figures, a more detailed description of the changes that occurred with varying SST/IP25 or on a temporal scale, i.e.. in relation to the deglacial/sea-ice loss through time, could be included.*

Our response: The temporal changes of the pelagic community is already described in the text. We now included the curve and description of the temporal development of the ratio between pelagic and benthic read counts. And added some text on the benthic community changes.

Revised text L 297-304:

"The ratio between counts assigned to pelagic and benthic families (pelagic:benthic ratio) is highest in samples dated to the late glacial and Younger Dryas compared to the warmer Bølling/Allerød and the Holocene (Fig. 4). The pelagic:benthic ratio shows a weak, negative correlation with SSTs ($\rho = -0.39$, p-value = 0.052), indicating that more reads are assigned to benthic families (and more reads to pelagic families) in phases with warmer SSTs.

In our dataset, Zosteraceae have higher read counts during the Holocene and are significantly, positively correlated with summer/early fall SSTs ($\rho = 0.73$, adjusted p-value = 0.026; Fig. 4b). Laminariaceae (kelp) have higher read counts during the late glacial.

Figure only for review: Correlation of pelagic:benthic ratio with SST and IP25.

Reviewer Comment: L122: *Environmental preference of pelagic taxa inferred by network analysis; this is a lengthy section, consider introducing subheadings to describe the sea-ice and ice-covered networks.*

Our response: We followed this suggestion and split the network analysis and the environmental preference of pelagic taxa into two sections, thereby also accounting for the new network analysis approach.

Revised section titles: We split the sections into first “Co-occurrence network-derived environment associations” (L 183ff) and a second “Environmental preference of pelagic taxa inferred by network analysis” (L 209ff).

Reviewer Comment: L130: *“decadal to centennial”:* *You might have done this already, but have a good overview about sedimentation rates and that ~780 years lie between each depth sampled, based on this could you narrow down the estimate how many years would be covered within your sedaDNA sub-sample?*

Our response: We now give an estimation of accumulated years per 1 cm sub-sample in the introduction section and added the following text.

Revised text L 97-101: “As *sedaDNA* samples contain averaged information on decadal to centennial time-scale depending on sedimentation rates (here according to our age model on average 15.5 years during the late glacial and 28.7 years during the Holocene), correlation in the network should not be interpreted as ecological interactions between linked taxa or actual co-occurrence on a short-time scale.”

Reviewer Comment: L150: *Are you able to investigate these reads to see if they might be one of those two suspected species?*

Our response: We analyzed our data and the database to show whether our approach could distinguish between the two species. Both *Gadus chalcogrammus* and *Gadus macrocephalus* are represented in the database by 9,943 and 5,727 unique minimizers (short substrings used for binning of k-mers) respectively. *Gadus morhua* is however much more represented in the database with 131,648,356 unique minimizers, hence classifications

on species level could be prone to a reference bias. On species level 6 counts were assigned to *Gadus chalcogrammus*, no counts were assigned to *Gadus macrocephalus*, but 159 were assigned to *Boreogadus saida* (Polar cod).

Revised text L 219-224:

“Reads assigned to this family can most likely be attributed to Pacific cod (*Gadus macrocephalus*), walleye pollock (*Gadus chalcogrammus*) or Polar cod (*Boreogadus saida*). While pollock and Pacific cod are generalist predators in the Bering Sea, they are strongly influenced by bottom-up effects through prey composition and availability^{40,46} compared to Polar cod which is a keystone species of the high-Arctic with a strong linkage to sea ice⁴⁷.”

Revised text in Supplementary Information:

“Among Gadidae, our reads were assigned to 7 of the 12 genera represented in the database. All of them except for *Micromesistius* (2 counts assigned) are present in the Bering Sea or the Pacific Arctic. The majority of reads were assigned to the genus *Gadus* (35,832 counts), which is driving the trends on family level. Notable, but still few assignments were found for the genus *Boreogadus* (Polar cod; 163 counts). Today, Polar cod is a keystone species of the high-Arctic with a strong linkage to sea ice⁴, which supports their presence only during the late glacial samples.”

Included Reference:

Kohlbach, D. *et al.* Strong linkage of polar cod (*Boreogadus saida*) to sea ice algae-produced carbon: Evidence from stomach content, fatty acid and stable isotope analyses. *Progress in Oceanography* **152**, 62–74 (2017).

Reviewer Comment: L197: I note that the network analysis for the benthic families is provided with the Supplementary Material, but if not limited in figure numbers it might be worth adding this figure to the main text.

Our response: For pelagic network we present sparnmen and ecoCoupula (suggested by Reviewer3), however, for the benthic taxa much less reads are assigned. As ecoCoupula requires a higher data number to set up reliable models, we refrained from implementing network analyses for benthic taxa at all.

Discussion:

Reviewer Comment: Much of the discussion currently focuses on speculation about potential changes in carbon export resulting from changes in differently sized phytoplankton communities. While this is an important point that is worthy of discussion in this manuscript, I am missing some discussion of the findings of the study about changes in the pelagic and benthic communities with time, correlations with SST, IP₂₅, and in relation to the discussion points that are currently part of the results (see above).

Our response: In accordance with the formatting guide we now use the Section heading “Results and discussion” with additional sub-heading addressing to present and discuss major results of our study. In the extra “discussion” section, which we entitled: “Implications for sea ice decline in the northern Bering Sea” we now discuss how pelagic and benthic communities might change under future warming and sea ice decline.

Revised text: See section “Implications for sea ice decline in the northern Bering Sea” (L 328ff).

Methods:

Reviewer Comment: L334: Add year of extraction.

Our response: Information was added

Revised text L 395-398: “In 2018 we collected and extracted samples from the archive half of a 9.05 m piston corer (SO201-2-12KL, Fig. 1) which was retrieved during RV Sonne cruise SO-201 “KALMAR” in 2009⁴³ and since then stored at 4°C. We applied the chronostratigraphy established by Max et al.⁴³.”

Reviewer Comment: L341: 2-4mL – the mL using makes me wonder if this was a sediment slush, was it diluted? If not I suggest to use cm³ or g as unit.

Our response: We apologize for the confusion. We were subsampling with syringes of which we have cut off the front and extracted all of it. This is why a volume of 2-4mL was taken and extracted.

Revised text L 409-410: “The DNeasy PowerMax Soil Kit (Qiagen, Germany) was used to extract total DNA from an average of 7.5 g sediment.”

Reviewer Comment: L350: change “with twice” to “twice with”

Our response: Changed.

Revised text L 417-420: “We measured the DNA concentration on a Qubit 4.0 fluorometer (Invitrogen, Carlsbad, CA, USA) using the Qubit dsDNA BR Assay Kit (Invitrogen, USA) and depending on the initial DNA concentrations we concentrated a volume of 300 µL or 600 µL using the GeneJET PCR Purification Kit (Thermo Scientific, USA) and eluted twice with 25 µL elution buffer.”

Reviewer Comment: L362: Add after “two”: “blanks”

Our response: Included.

Revised text L 430-433: “Therefore, five of the seven extraction blanks (50 µL per extraction blank were pooled and 5 µL of the pool used for the library preparation) were sequenced together with their corresponding samples in the first run, while the last two blanks were processed later in the second sequencing run.”

Reviewer Comment: L486/487: Delete, this is already in the previous sentence.

Our response: We removed the duplicated sentence.

Reviewer Comment: L487/488: Add a reference.

Our response: We included the following citation: Méheust, M., Stein, R., Fahl, K., Max, L. & Riethdorf, J.-R. High-resolution IP25-based reconstruction of sea-ice variability in the western North Pacific and Bering Sea during the past 18,000 years. *Geo-Marine Letters* **36**, 101–111 (2016).

Reviewer #2 (Remarks to the Author):

Review of MS NCOMMS-22-05048

1. Key results: Your overview of the key messages of the study, in your own words, highlighting what you find significant or notable. Usually, this can be summarized in a short paragraph.

MY COMMENTS:

This study brings the field of ancient environmental DNA (eDNA) forwards by presenting change in functional diversity of a wide range of marine organisms (notably prokaryotic and eukaryotic phytoplankton, macrophytes, fish, some marine mammals, but also potentially less described invertebrates) during the last 20,000 using a novel ancient eDNA approach on marine sediments (shotgun metagenomics). Correlation network analyses and correlations to paleoclimatic variables (SST, IP25) are used to associate taxonomic and functional groups to climate-driven change in sea-ice conditions at the site. The main 'shifts' in ecosystem response to post-glacial sea-ice loss recorded by this novel approach lie in smaller-sized algal primary producers related to different C storage and export from the oceans and major fish community reorganization at higher trophic levels.

2. Validity: Your evaluation of the validity and robustness of the data interpretation and conclusions. If you feel there are flaws that prohibit the manuscript's publication, please describe them in detail.

MY COMMENTS:

Reviewer Comment: 103-106: *While the taphonomic-based explanations provided are plausible causes for the discrepancy in relative abundance, I would think this difference is also likely due to database bias, where a number of groups e.g., invertebrates are relatively poorly represented in NCBI. Perhaps the authors have a better idea of **genomic coverage of their organisms of interest to assess reference bias**, please provide if possible.*

Our response: Database gaps are present for many taxonomic groups. In December 2020, the year we downloaded the database, genomes of 594 fish species were available, which is about 1.85% of estimated fish species on our planet (Randhawa & Pawa 2021). For the Western Bering Sea, we are lucky that the dominant fish (e.g. Salmonidae, Gadidae, Clupeidae) and Copepod (e.g. Calanidae) families are represented by at least one reference genome. Aside from Copepods, other invertebrates such as Echinoderms, Molluscs, and Cnidarians are represented in the database. For example, for nematodes, 154 genomes and over 4 million nucleotide sequences are available.

Revised text L 161-163: "Alternatively, this discrepancy could be explained by differential preservation or degradation due to different skeletal properties or large gaps in the sequence reference database regarding marine macrofauna for both barcoding genes and genomes⁴⁰."

"

Added reference:

40. Hestetun, J.T., Bye-Ingebrigtsen, E., Nilsson, R.H. *et al.* Significant taxon sampling gaps in DNA databases limit the operational use of marine macrofauna metabarcoding. *Mar. Biodivers.* **50**, 70 (2020). <https://doi.org/10.1007/s12526-020-01093-5>

Reviewer Comment: l. 147: *I could not find the read counts associated to copepods (rather %), please add.*

Our response: We included read counts from all families in Supplementary data (file1) and added the following information in the main text.

Revised text L 210ff: "Among algae, this pattern can be observed for Bacillariaceae (0.8% of pelagic counts), Stephanodiscaceae (0.6%), Thalassiosiraceae (1.1%), Tripariaceae (0.3%), and Phaeocystaceae (0.6%), which can be found in sea ice or as a part of

cryopelagic communities⁴⁵. They co-occur with the chlorophytes Mamiellaceae (0.5%) and Bathycoccaceae (1.7%), which contain cold-adapted lineages that prevail in the marginal ice zone or in landfast sea ice^{46,47}. Closely linked to these primary producers are copepods (Calanidae 0.03%, Metridinidae 0.03%), [...]"

Reviewer Comment: I. 174: Warmer conditions seem to favor early-stage fish that, in the case of salmon at least, would occur in freshwater rivers, possibly in quite distant localities, correct? Do we know where the fish populations at the site breed at present? The evidence here is for SST, strictly speaking, which globally would be correlated to land temperatures of the breeding grounds, but **supporting evidence/references** would help make this point clearer.

Our response: Yes, this is correct, spawning grounds for Salmonidae are rivers in Kamchatka and Chukotka. We added text about Salmonidae spawning grounds and land temperature change in the discussion section.

Revised text L 249-254:

"Rivers in Kamchatka and near-by Chukotka are spawning grounds for Salmonidae⁵⁷ and are known as a diversity hotspot; for example, all species of Pacific salmon (*Oncorhynchus*) occur in the area. Summer temperature reconstructions from Chukotka show a similar trend to SST reconstructions from our core⁴³ with lowest values during the pre-15 ka and an about 5°C rise until the early-to-mid Holocene temperature maximum⁵⁸."

Added references:

57. Jones, V. & Solomina, O. The geography of Kamchatka. *Global and Planetary Change* **134**, 3–9 (2015).

58. Andreev, A. A. *et al.* Late Pleistocene to Holocene vegetation and climate changes in northwestern Chukotka (Far East Russia) deduced from lakes Ilirney and Rauchuagytygn pollen records. *Boreas* **50**, 652–670 (2021).

Reviewer Comment: I. 179: Do increased freshwater inputs bring more nutrients overall?

Our response: Yes, although the processes are a bit more complex. We included a sentence and a reference.

Revised text L 259-261: "While freshwater discharge is a source of organic carbon and nutrients that can enhance productivity, the lower salinity of surface waters increases vertical water column stratification⁵⁸."

Included reference: "58. Heikkilä, M., Ribeiro, S., Weckström, K. & Pieńkowski, A. J. Predicting the future of coastal marine ecosystems in the rapidly changing Arctic: the potential of palaeoenvironmental records. *Anthropocene* 100319 (2021) doi:10.1016/j.ancene.2021.100319."

Reviewer Comment: I. 186: Could N-fixation Cyanobacteria also be an advantage? Please elaborate.

Our response: We included some information about the surface-area-to-volume ratio.

Revised text L 263-269: "Under nutrient depleted conditions, cyanobacteria are assumed to have several advantages such as the small surface area-to-volume ratio of their cells and thus lower nutrient requirements⁵⁶, the capability of nitrogen fixation (here limited to

Nostocaceae), and their preference of inorganic nitrogen in the form of ammonium. Ammonium has an energetic advantage compared to nitrate, the preferred form by diatoms, which has to be transformed to ammonium within the cell at an energetic cost⁶⁴.

Reviewer Comment: *Discussion: The evidence presented for a shift in phytoplankton size is quite convincing, but I would like to see some comments on the **change in other functional aspects in primary producers that may offer alternative or complementary explanation on the impact of this shift in composition on the ecosystem e.g., N-fixation by cyanobacteria and role in the biogeochemical cycling.***

Our response: See response above. We included some information about N-fixation by cyanobacteria.

Revised text L 263-267: "Under nutrient-depleted conditions, cyanobacteria are assumed to have several advantages such as the small surface area-to-volume ratio of their cells and thus lower nutrient requirements⁶³, the capability of nitrogen fixation (here limited to Nostocaceae), and their preference for inorganic nitrogen in the form of ammonium. Ammonium has an energetic advantage compared to nitrate, the preferred form by diatoms, which has to be transformed to ammonium within the cell at an energetic cost⁶⁴."

Added reference:

64. Glibert, P. M. *et al.* Pluses and minuses of ammonium and nitrate uptake and assimilation by phytoplankton and implications for productivity and community composition, with emphasis on nitrogen-enriched conditions. *Limnology and Oceanography* **61**, 165–197 (2016).

Reviewer Comment: *I. 203-207: It is potentially interesting that such fewer reads were found from the benthic groups, but it would be best to discuss number of reads assigned from rarefied reads only, not both. Raw reads might differ in different samples based on library success, which could be confounded with e.g., sediment properties during different time periods (e.g., more abundant classified reads in Suppl. Fig. 1, 13-19 ka yrs BP). The difference between benthic and pelagic reads would be more effectively presented by plotting a Pelagic:Benthic ratio along Figs 2 and 4. This part is also somewhat speculative, as a lot of questions remain as of the taphonomy of ancient DNA in sediments.*

Our response: This is an interesting point. We now present the ratio (based on resampled read counts) of pelagic:benthic families and added the ratio to Figures 2 and 4. We also tested this ratio for correlations with SSTs (Spearman's rho = -0.39, p-value = 0.052) and IP25 (Spearman's rho = 0.15, p-value = 0.48) and the added results to the main text.

Revised Figure 2:

Figure 2. Pelagic families in the *sedaDNA* record of core SO201-2-12. Only families represented in at least 3 samples and by at least 1.5% proportion of read counts after resampling are shown. The dendrogram (based on constrained hierarchical clustering) and IP_{25} concentrations from the same core³⁹ and sea-surface temperature (SST_{UK37}) reconstructions of the Northwest Pacific⁴⁰ (blue) and Northeast Pacific⁴¹ (black) are added for comparison. Families which are significantly, positively correlated with IP_{25} or SST_{UK37} are highlighted in blue or red, respectively. The pelagic:benthic ratio (black line) shows the proportion of reads assigned to pelagic families in relation to reads assigned to benthic families. The ratio between reads assigned to phototrophic bacteria and reads assigned to phototrophic protists is given as blue-green line. Black triangles next to the IP_{25} record mark radiocarbon-dated calibrated ages⁴⁰. LGM = Last Glacial Maximum, HS1 = Heinrich Stadial 1, B/A = Bølling/Allerød, YD = Younger Dryas.

Figure for review only: Correlation of pelagic:benthic ratio with SST and IP_{25} .

Revised text L 283-289: “The overall lower number of reads attributed to benthic (263,193 originally, 45,975 after resampling) in comparison to pelagic (476,058 originally, 210,800 after resampling) organisms could suggest that upper-ocean productivity is higher than benthic productivity at the deep continental margin. However, the low assignment rate, particularly among benthic fauna may, to an unknown extent, also relate to the large number of undescribed species and the sequence database gap of deep-sea biota in general⁶⁸. Some typical deep-sea biota were detected such as free-living nematodes (Oxystominidae, 0.1%) albeit only with a few reads (Supplementary data (file 2)).”

L 297-301:

“The ratio between counts assigned to pelagic and benthic families (pelagic:benthic ratio) is highest in samples dated to the late glacial and Younger Dryas compared to the warmer Bølling/Allerød and the Holocene (Figs. 2,4). The pelagic:benthic ratio shows a weak, negative correlation with SSTs ($\rho = -0.39$, p-value = 0.052), indicating that more reads are assigned to benthic families in phases with warmer SSTs.”

Reviewer Comment: *I. 299 How strong is the DNA signal for Copepods (or, as mentioned here, large Copepods)? I could not find the number of reads for the three Families presented in text and Fig. 5. This group is an important part of the foodweb as presented in Fig. 5, but I am not convinced that the strength of the signal for this group supporting the discussion points.*

Our response: We included proportions in the text and added the Copepod families Calanidae and Metridinidae to Figure 2. Calanidae (44 counts after resampling) and Metridinidae (51 counts after resampling) may not be dominant, but Metridinidae are significantly, positively correlated with IP₂₅ ($\rho = 0.622$, adjusted p-value=0.0368) while Calanidae are not significantly, positively correlated with IP₂₅ ($\rho = 0.4698$, adjusted p-value=0.1775), which is provided as Supplementary Table 5 and 6. We included Supplementary Table 6 (currently only provided via PANGAEA link) with read counts for all pelagic and benthic families and the number of edges they have in the network.

Revised text L 216 ff: “Closely linked to these primary producers are copepods (Calanidae 0.03%, Metridinidae 0.03%), [...]”

Revised Figure 2:

Supplementary Table 4. Correlations (Spearman $\rho > 0.4$) and Benjamini-Hochberg adjusted p-values of nodes (family names) with SSTs (late summer/early fall temperature indicator) and IP₂₅ (seasonal sea ice indicator) variables.

Group	Family name	SSTs		IP ₂₅	
		ρ	p-value	ρ	p-value
pelagic	Gadidae	-0.7408	0.0043	0.7198	0.0067
pelagic	Oxytrichidae	-0.7387	0.0046	0.7566	0.0028
pelagic	Bacillariaceae	-0.7205	0.0067	0.686	0.0133
pelagic	Metriclinidae	-0.7082	0.0083	0.6221	0.0368
pelagic	Bathycoccaeae	-0.7008	0.01	0.6913	0.012
pelagic	Phaeocystaceae	-0.6838	0.014	0.5881	0.0572
pelagic	Ulnariaceae	-0.6409	0.0276	0.4631	0.1864
pelagic	Fragilariaceae	-0.6336	0.0315	0.5328	0.1038
pelagic	Ulvaeeae	-0.6229	0.0365	0.6615	0.0208
pelagic	Naviculaceae	-0.6146	0.0414	0.5096	0.1274
pelagic	Suessiaceae	-0.5441	0.0929	0.5985	0.0517
pelagic	Pycnococcaeae	-0.5302	0.1053		
pelagic	Amphipleuraceae	-0.5302	0.1053		
pelagic	Attheyaceae	-0.5292	0.108	0.4781	0.1676
pelagic	Holocentridae	-0.5123	0.1242		
pelagic	Thalassiosiraceae	-0.5123	0.1242		
pelagic	Stephanodiscaceae	-0.4977	0.1429		
pelagic	Chromonadaceae	-0.4849	0.16		
pelagic	Cymatosiraceae	-0.4688	0.1796		
pelagic	Toxariaceae	-0.4528	0.1983		
pelagic	Noelaerhabdaceae	0.4538	0.1967		
pelagic	Selenastreaeae	0.4544	0.1961	-0.5087	0.1289
pelagic	Eucalanidae	0.469	0.1903		
pelagic	Serranidae	0.4608	0.1891		
pelagic	Gloeobacteraceae	0.4654	0.1841	-0.5327	0.1038
pelagic	Merismopediaceae	0.4669	0.1821		
pelagic	Bovichtidae	0.48	0.1655		
pelagic	Clupeidae	0.4815	0.1629	-0.4873	0.1573
pelagic	Fonticulaceae	0.4924	0.1506	-0.4581	0.1914
pelagic	Rhuronectidae	0.5131	0.1238	-0.4742	0.1716
pelagic	Salpingoecidae	0.5205	0.1145		
pelagic	Desmidiaceae	0.5258	0.1086	-0.4684	0.1563
pelagic	Rhinodontidae	0.5269	0.1076		
pelagic	Nephroselmidae	0.5366	0.1012	-0.5652	0.076
pelagic	Chlorobiaceae	0.5577	0.0814	-0.5874	0.058
pelagic	Microcystaceae	0.5777	0.0659	-0.5681	0.0732
pelagic	Sciaenidae	0.5838	0.0605	-0.5127	0.1242
pelagic	Klebsormidiaceae	0.5954	0.0534	-0.6377	0.0293
pelagic	Blenniidae	0.6085	0.0449		
pelagic	Sebastidae	0.6178	0.0398	-0.5859	0.0592
pelagic	Salmonidae	0.6185	0.0393	-0.455	0.1956
pelagic	Chlorellaceae	0.6328	0.0318	-0.512	0.1247
pelagic	Chaetocerotaceae	0.6477	0.0252	-0.7552	0.0028
pelagic	Hyellaceae	0.6483	0.0251	-0.4681	0.1803
pelagic	Syngnathidae	0.6577	0.0218	-0.4796	0.1658
pelagic	Rosellaceae	0.6769	0.0158	-0.5727	0.0699
pelagic	Musculidae	0.7346	0.0047	-0.6936	0.0117
pelagic	Acartoherataceae			-0.594	0.0541
pelagic	Trebouxiaceae			-0.4654	0.1841
pelagic	Chlorococcaeae			-0.4577	0.192
pelagic	Calanidae			0.4698	0.1775
benthic	Bangiaceae	0.6473	0.1668		
benthic	Zosteraeae	0.7319	0.0265	-0.6654	0.097

Reviewer Comment: I. 475: Having control over sea-ice from the same sediment profile is a strong asset of this MS, which would be highlighted further by stating clearly early on in the MS and/or detailed in methods that these records are derived from the same core (if Fig. 2, add also intext). However, the **lack of it for the first and I.475 'last two for SST and the last four for IP25 data' samples** (here there actually seem to be an offset with the data presented in Figs. 2 and 4, where SST and IP25 records appear longer by 1 sample or so) **limit the conclusions, particularly for the early phase.** The SST record available in the Northeast Pacific appears to have incursions into warm conditions at the very base of the record, which could correspond to 'ice-free conditions', or perhaps not well represented by

the averaging of the closest samples (same applies for the top-most sample). Please provide further justification for the extrapolation, or remove samples from statistical analyses with SST and IP₂₅ for these intervals and adjust conclusions accordingly.

Our response: We interpolated the variables SST and IP₂₅ (see figure below) to address this issue with a new method. Our extrapolation at the core base is based on the assumption of stable sea-ice conditions between 20 and 18 ka. This assumption is consistent with several proxy records from several cores across the Bering Sea and the Sea of Okhotsk which all show that winter sea ice coverage was stable during the time between 18 to 20 kyr BP^{11,12,14}. We extrapolated a stable value of IP₂₅ toward the base and are confident that this is the best knowledge based extrapolation method and also superior to deleting the entire samples from data analyses.

Supplementary Figure 11. Interpolated and extrapolated values for the environmental variables IP₂₅ ($\mu\text{g g}^{-1}$ sediment) and SST_{UK37} ($^{\circ}\text{C}$).

Revised text L 70-73:

“A regional palaeoceanographic framework for past sea-ice cover in the Bering Sea is based on reconstructions using the highly branched isoprenoid alkane IP₂₅¹² which is produced by diatoms that are bound to a life in sea ice¹⁰ and diatom microfossils¹¹.”

L 109-113:

“The paleoenvironmental backbone is based on previous multi-proxy reconstructions of the coring site derived from this core, including diatom microfossils and the biomarker IP₂₅, which show that the coring site was covered by seasonal sea-ice during the Last Glacial Maximum (LGM), Heinrich Stadial 1, and the Younger Dryas, while the phases of the Bølling-Allerød and the Holocene were predominantly sea-ice free^{12,16}.”

10. Belt, S. T. & Müller, J. The Arctic sea ice biomarker IP₂₅: a review of current understanding, recommendations for future research and applications in palaeo sea ice reconstructions. *Quaternary Science Reviews* **79**, 9–25 (2013).

11. Matul, A. G. Probable limits of sea ice extent in the northwestern Subarctic Pacific during the last glacial maximum. *Oceanology* **57**, 700–706 (2017).
12. Méheust, M., Stein, R., Fahl, K. & Gersonde, R. Sea-ice variability in the subarctic North Pacific and adjacent Bering Sea during the past 25 ka: new insights from IP25 and Uk'37 proxy records. *Arktos* **4**, 8 (2018).
13. Harada, N., Shin, K. H., Murata, A., Uchida, M. & Nakatani, T. Characteristics of alkenones synthesized by a bloom of *Emiliania huxleyi* in the Bering Sea. *Geochimica et Cosmochimica Acta* **67**, 1507–1519 (2003).
16. Méheust, M., Stein, R., Fahl, K., Max, L. & Riethdorf, J.-R. High-resolution IP25-based reconstruction of sea-ice variability in the western North Pacific and Bering Sea during the past 18,000 years. *Geo-Marine Letters* **36**, 101–111 (2016).

3. Significance: Your view on the potential significance of the conclusions for the field and related fields. If you think that other findings in the published literature compromise the manuscript's significance, please provide relevant references.

MY COMMENTS:

I am not familiar with metabarcoding studies of ancient eDNA from marine sediments, however to my knowledge this could be one of the first applications of shotgun metagenomics in such context, at once providing a paleoecological record of the largest breath of lifeforms over the longest period. The strong record for fish, in particular, is a major advance for the marine-based and sedaDNA field, which is also of potential great significance for evaluating long-term changes in fish stocks and fish ecological and evolutionary dynamics in the future, which remain difficult to quantify in oceans today.

Other papers using shotgun sequencing on marine sediments that I am aware of: Gaffney et al 2020 Geosciences, who presented plant sedaDNA from a tsunami sediment deposits, which did not aim to reconstruct paleoecological dynamics. Eukaryote composition from shotgun metagenomics were also compared to metabarcoding data from Santa Barbara Basin deep-sea sediments over the Holocene, mainly focusing on diatoms (Armbrecht et al 2021 ISME).

Reviewer Comment: *Futhermore, the authors go beyond listing taxa and provide an elegant and straightforward statistical analysis of their molecular data against environmental variables by using correlation networks (please see comments below). Altogether, their novel approach allows them to measure the effects of (long-term) climate change on important ocean processes like primary production and food-web interactions at an unprecedented level of detail and taxonomic breadth. The relevance of this work as 'analogous' to ongoing sea-ice loss trends in subarctic coastal environments that is discussed by the authors in Discussion could be emphasized in the introduction already by describing how the 'seasonal sea-ice' to 'see-ice free' conditions over the period studied have changed and relate to documented ongoing and expected e.g., IPCC trends for relevant regions.*

Our response: We included the information into our introduction.

Revised text L44-46:

“Long-term trends in the Bering Sea show an ongoing decline of sea-ice duration which is projected to decline further due to later freeze-up (by 20 to 30 days) and earlier break-up (by 10-20 days) until the middle of this century³.”

L 65-68:

“Past ecosystem responses to sea-ice loss at the Pleistocene–Holocene boundary in the southern part of the Bering Sea could possibly reveal such long-term rearrangements and serve as an analog for future changes in the Pacific Arctic.”

Added citation:

3. Wang, M., Yang, Q., Overland, J. E. & Stabeno, P. Sea-ice cover timing in the Pacific Arctic: The present and projections to mid-century by selected CMIP5 models. *Deep Sea Research Part II: Topical Studies in Oceanography* **152**, 22–34 (2018).

4. Data and methodology: Your assessment of the validity of the approach, the quality of the data, and the quality of presentation. We ask reviewers to assess all data, including those provided as supplementary information. If any aspect of the data is outside the scope of your expertise, please note this in your report or in the comments to the editor. We may, on a case-by-case basis, ask reviewers to check code provided by the authors (see this Nature editorial for more information).

Reviewers have the right to view the data and code that underlie the work if it would help in the evaluation, even if these have not been provided with the submission (see this Nature editorial). If essential data are not available, please contact the editor to obtain them before submitting the report.

MY COMMENTS:

Reviewer Comment: Outliers: At least three (1.08, 5.6 and 9.84, cal ka yrs[10] [11] [HZ12]) of the samples presented seem to be outliers in the dataset (own branches on dendrogram Figs 2 and 4, very distant from neighbours), and I suspect the low raw, classified pre- and post- read counts exclusion of taxa and resampling may explain this as an artifact, rather than a true signal. This could be showed by presenting e.g., a PCoA of the samples (with and without controls) with a biplot of the main taxa before and after exclusion of taxa in Suppl. I think removing them would change correlations and interpretations slightly (likely strengthen them), and help present the counts from the Figures 2 and 4 on clearer scales. Please provided the data used to plot Supplementary File 1 in table form, and please also provide read counts before and after exclusion of taxa (described on l. 409-424).

Our response: The presentation of this detailed table required a re-processing of the entire data analyses, which was only possible against a newer version of the non-redundant reference database (downloaded April 2021 before it was May 2020) with the Kraken2 classifier 2.0.8-beta instead of 2.0.7-beta. The usage of a newer database and an upgraded version of the classifier Kraken2 changed our data slightly but did not affect any of our conclusions. Therefore, all analyses and plots have been updated and the samples which are mentioned as outliers, do not appear as outliers anymore (Supplementary Figs. 1,2). We now provide all processing steps as Supplementary data including raw read counts, read counts after deduplication, filtering and trimming, classification and resampling of selected families. All text, tables and figures were updated according to the results of the re-processed data. The PCoA based on Bray Curtis dissimilarities shows that the overall compositions of the three samples in question cluster with other samples from the Holocene. Only after taxonomic filtering and resampling their compositions are markedly different. The

PCoA shows that the blanks have a distinct composition which is highly dissimilar from the samples (Supplementary Fig. 2).

Supplementary Figure 1. Number of read counts per sample before taxonomic filtering. The median (324,044 counts) is indicated by the dashed line.

Supplementary Figure 2: PCoA based on Bray Curtis distances (a) of the original dataset classified on family level and (b) after filtering and resampling of pelagic taxa. Data points highlighted in blue are the blanks.

Reviewer Comment: All figure captions and tables titles: specify whether reads presented are rarefied or not.

Our response: Information included.

Revised caption of Fig 3:

Figure 3. Correlation network of pelagic families. Families (nodes) that are positively correlated (Spearman rank correlation coefficients > 0.4, adjusted p-value < 0.1) with each other are connected by links (edges) after resampling. The node size is log-scaled according to taxa abundance, while node colors represent (a) the functional group to which the organisms belong, and (b) whether the node showed a positive trend (Spearman rank correlation coefficients > 0.4, adjusted p-value < 0.2) with the seasonal sea-ice biomarker IP_{25}^{42} (blue) or SSTs⁴⁴ (red). Node labels are given if a family exceeds 53 read counts.”

Reviewer Comment: Fig. 2: Please consider using the same scale for rel. abundances. At present it is difficult to see groups most recorded and transitions. Also, as they are re-sampled and not actual read-counts, I suggest presenting into percentages for ease of comparisons, also to be more consistent with numbers reported in text. Figs 2 and 4: Boxes for the ice-free ecosystem are different, which one is correct (Fig. 4?)? Consider marking the Families (significantly) positively correlated to SST and IP₂₅ in red and blue in Figs 2 and 4 for ease of interpretation. l. 142: Add climatic periods mentioned in text to Figs 2 and 4.

Our response: We combined all comments provided by Reviewer 2 regarding Figures 2 and 4 here and addressed them all during the revision. Please find the revised figures below. We changed it to percentages. On a similar scale a few taxa dominate the plot and make it impossible to see temporal changes for the rest of the families in the plot. Therefore, we are keeping the different scales, but refer to it in the Figure caption. We adjusted the boxes.

Revised Fig. 2:

Figure 2. Pelagic families in the *sedaDNA* record of core SO201-2-12. Only families represented in at least 3 samples and by at least 1.5% proportion of read counts after resampling are shown. The dendrogram (based on constrained hierarchical clustering) and IP₂₅ concentrations from the same core⁴⁰ and sea-surface temperature (SST_{UK37}) reconstructions of the Northwest Pacific⁴¹ (blue) and Northeast Pacific⁴² (black) are added for comparison. Families which are significantly, positively correlated with IP₂₅ or SST_{UK37} are highlighted in blue or red, respectively. The pelagic:benthic ratio (black line) shows the proportion of reads assigned to pelagic families in relation to reads assigned to benthic families. The ratio between reads assigned to phototrophic bacteria and reads assigned to phototrophic protists is given as blue-green line. Black triangles next to the IP₂₅ record mark radiocarbon-dated calibrated ages⁴¹. LGM = Last Glacial Maximum, HS1 = Heinrich Stadial 1, B/A = Bølling/Allerød, YD = Younger Dryas.

Revised Fig. 4:

Figure 4. SedaDNA record of benthic families of core SO201-2-12. Only families represented in at least 3 samples and by at least 1.5% proportion of read counts after resampling are shown. The dendrogram (based on constrained hierarchical clustering) and IP_{25} concentrations from the same core⁴⁰ and sea-surface temperature (SST_{UK37}) reconstructions of the Northwest Pacific⁴¹ (blue) and Northeast Pacific⁴² are added for comparison. Zosteraceae, which are significantly, positively correlated with SST_{UK37} , are highlighted in red. The pelagic:benthic ratio shows the proportion of reads assigned to pelagic families in relation to reads assigned to benthic families. The ratio between reads assigned to phototrophic bacteria and reads assigned to phototrophic protists is given as blue-green line. Black triangles next to the IP_{25} record mark radiocarbon-dated calibrated ages⁴¹. LGM = Last Glacial Maximum, HS1 = Heinrich Stadial 1, B/A = Bølling/Allerød, YD = Younger Dryas.

Reviewer Comment: l. 83: 72.5% of classified read pairs overlapped and were merged. Is the total read number on l.76 including overlapping read pairs (reads from 'the same DNA strand' counted twice?). Please clarify this elsewhere also, it's important that the merged reads are not counted twice.

Our response: Merged reads were not counted twice. To avoid any confusion we now provide the number of read pairs instead of merged or paired reads and adjust the text accordingly.

Revised text L 132-133: "Of those, we were able to classify 13,119,146 read pairs (merged and paired read pairs together)."

Reviewer Comment: l. 190, Fig. 3: Nice figure, well thought out and functional groups easy to distinguish. Please add legend for equivalent % for size of nodes (same applies to other network provided in suppl. materials). Add explanation to caption for labelling of nodes or not (some nodes don't have labels, likely because they have too low %). Some family names are difficult to read and associate to the nodes, consider placing them more off from one another with black lines to the nodes, include fewer names or replace with images of taxa as per Fig 5 where appropriate.

Our response: We added a node size legend and included the threshold for labelling nodes into the caption. After re-analysis with the newer Kraken2 database (see response above), Figure 3 changed slightly.

Revised Figure 3:

Figure 3. Correlation network of pelagic families. Families (nodes) that are positively correlated (Spearman rank correlation coefficients > 0.4, adjusted p-value < 0.1) with each other are connected by links (edges) after resampling. The node size is log-scaled according to taxa abundance, while node colors represent **a** the functional group to which the organisms belong, and **b** whether the node showed a positive trend (Spearman rank correlation coefficients > 0.4, adjusted p-value < 0.2) with the seasonal sea-ice biomarker IP₂₅⁴⁰ (blue) or SSTs⁴² (red). Node labels are given if the family exceeds 53 read counts.

Reviewer Comment: I. 322 The chronology was reported in a 2012 publication; please indicate whether the same, 2009 core was used, and if so, indicate the date or year and procedure for the splitting of the core that was used for the DNA work. Cores stored unopened vs opened for many years prior to DNA sub-sampling may be differently affected by secondary growth of e.g., fungi, yeast and Bacteria that would influence DNA results (as per stated by the authors). If different cores were used, please describe the splitting procedure and how those were correlated for the age-depth model.

Our response: The archive half of the same core was used. We included the underlined information into the manuscript and added a dedicated paragraph “Opening of cores and subsampling procedures to prevent contamination” to the supplement.

Revised text L 395-397:

“In 2018 we collected and extracted samples from the archive half of a 9.05 m piston corer (SO201-2-12KL, Fig. 1) which was retrieved during RV Sonne cruise SO-201 “KALMAR” in 2009⁴³ and since then stored at 4°C.”

L 400-402:

“The sampling was carried out at GEOMAR, Kiel, in a sedimentology laboratory devoid of any molecular biology work. Details about the subsampling procedure and the opening of the cores can be found in the Supplementary Information.”

New paragraph in Supplementary Information: “During the expedition on RV Sonne cruise SO-201 “KALMAR” in 2009, the core liner was cut into 1 m sections on board of the ship, which is common practice. After measuring the magnetic susceptibility (not part of the manuscript) on the round-sections, each 1m-section was split into working and archive halves. The working halves were sampled onboard and completely used up, providing sample material, commonly in 1 cm-slices, for various working groups. Color reflectance measurements (not part of the manuscript) were made on the archive half sections, which were covered with transparent polystyrene foil before measurement. The archive halves were packed into plastic D-tubes and stored at ~4°C, within a few hours of recovery.”

Reviewer Comment: I. 327 As the lab facilities used for sub-sampling are not a dedicated clean facilities, please include the precautions that were undertaken to limit contaminants (air, surfaces, tools) from the wide range of organisms studied at GEOMAR, like fish and phytoplankton.

Our response: We included the following description of sub-sampling procedures and localities in a dedicated chapter in the Supplementary Information.

Revised text in introduction L 400-402:

“The sampling was carried out at GEOMAR, Kiel, in a sedimentology laboratory devoid of any molecular biology work. Details about the subsampling procedure and the opening of the cores can be found in the supplement.”

Added text in supplement: “The sampling was carried out at GEOMAR, Kiel, in a sedimentology laboratory devoid of any molecular biology work. The lab is not used for experiments on modern phytoplankton, fish, or other living organisms. The surface of the table and the core liner were cleaned DNAExitus Plus and rinsed of with MiliQ water. The plastic foil covering the sediment was removed sample-by-sample to limit exposure of the sediments to the surrounding air. Clean, single-use knives (soaked in DNAaway for 10 minutes, rinsed with 70% Ethanol and irradiated for 20 min on each side in a UV-crosslinker) were used to remove ~2mm of the exposed sediment. Then a sterile scalpel blade was used to remove a second layer of sediment (~2 mm). A syringe, of which the front was cut off directly before, was inserted aiming for a sample volume of up to 4 mL. The sample was transferred into a sterile 8 mL tube and stored at -20°C until DNA extraction. During the whole procedure, full body suits, exchangeable arm sleeves, face masks, and two pairs of gloves on top of each other were worn.”

Reviewer Comment: I. 333 The Epp et al 75 is wrong, it is likely the authors meant ‘Epp, L. S., Zimmermann, H. H., & Stoof-Leichsenring, K. R. (2019). Sampling and extraction of ancient DNA from sediments. In B. Shapiro, A. Barlow, P. D. Heintzman, M. Hofreiter, J. L. A. Paijmans, & A. E. R. Soares, (Eds.), *Ancient DNA: Methods and protocols* (pp. 31–44). Springer New York.’ rather than ‘Epp, L. S. A global perspective for biodiversity history with ancient environmental DNA. *Molecular Ecology* 28, 2456–2458 (2019).’

Our response: As we included more information about the subsampling procedure in the Supplementary Information we removed the sentence which included this reference.

Reviewer Comment: I. 345: Please clarify in text whether the same extracts as ref 15 were used (I assume this is the case, as the same extract controls are analysed), or whether new extracts were made from the samples. If the same extracts were used, please clarify that the protocol is exactly as per ref 15 from the first sentence of this paragraph.

Our response: We used the stocks from the same extracts that were kept continuously frozen after the extractions while working aliquots were used for DNA metabarcoding.

Revised text L 410-: “For a previous DNA metabarcoding analysis²⁰ we extracted DNA for 63 samples in 7 batches where each batch contained up to 9 samples and an extraction blank (in total 7 negative controls). From those, we chose the stock solutions of 25 DNA extracts and the corresponding extraction blanks, which had been kept continuously frozen since the extraction for the metagenomic shotgun sequencing.”

Reviewer Comment: I. 376: Were the controls also provided with 10-14 cycles? This may explain why such few reads could be sequenced.

Our response: The extraction and library blanks were treated like the samples and were amplified with 10-14 cycles, we added this information to the main text. The low number of read counts is a result of the low DNA content in our blanks. We additionally added the information about the data processing for samples as well as for negative controls and prepared it in table form as Supplementary data (file 1). Related to another reviewer comment below, we also included a dedicated section (“3. Negative controls”) about read counts of the blanks into the revised Supplementary Information.

Revised text L 445-448: “For samples the subsequent double-indexing was performed by PCR in 10-14 cycles depending on library concentration [...]. Blanks were amplified with the same number of cycles, although initial DNA concentration was 0 ng/μL. ”

New section in supplement:

“Throughout the library preparation process the negative controls were treated in the same way as the samples, also in terms of PCR cycle numbers for the indexing step. We retrieved between 395,384 and 2,494,694 read pairs for the different negative controls of which between 0.01 and 30% were duplicates. After deduplication, trimming and filtering for low complexity, low quality, residual adapters and length only between 0.3 and 2.1% of the original read pairs were left. This suggests, our blanks contain very low levels of DNA and that increasing cycle numbers would predominantly amplify our duplication rate and increase the number of adapters in the sequencing data.”

Reviewer Comment: I.390. A 5% replication rate seems impressively low; is this in a similar range from other sedaDNA studies? Could the authors provide the distribution from Suppl. Fig. 1 for all samples in table form? Apologies if I have missed this.

Our response: We now provide deduplication results from each sample and negative controls in a table form (see Supplementary File 1 “data_processing.xlsx” in the Supplementary Material). To make sure of the duplication rate in our data we re-analyzed the raw data with a different tool (clumpify, BBMap package, <https://www.biostars.org/p/225338/>) than originally used in the manuscript (FastUniq v. 1.1). Both methods remove only exact duplicates from the data and calculate a very similar average duplication rate. FastUniq v. 1.1 calculated a mean duplication rate (including blanks and samples) of about 5.3% and clumpify calculated on average 10.6%. Highest

deviations between both approaches are detected in the duplication rates in the blanks (see Figure below this text, only for response letter), whereas within samples the rates are very similar (FastUniq=4.2%, clumpify=7.2%). The high similarity of duplicate rates in samples found in the two different approaches support the correctness of the duplication rate in our data. This finding is also supported by the publication by Gaia et al. 2019 who compared different deduplication tools and showed a high comparability between the tools.

<https://www.nature.com/articles/s41598-019-48242-w/tables/1>

Unfortunately, duplication rates are usually not distinctly reported from sedDNA studies. We found one publication by Carpenter et al. 2013 on ancient human teeth and hair samples (<https://www.sciencedirect.com/science/article/pii/S000292971300459X>) which reported of a similar duplication rate ranging between 2-20% identified by a deduplication tool in the SAMTools package

(<https://academic.oup.com/bioinformatics/article/25/16/2078/204688?login=true>).

Figure for review only: Figure of % of duplicates detected in raw reads comparing two deduplication approaches (FastUniq and clumpify).

Revised text L 459-466: “We used FastQC v. 0.11.9⁸³ to check the quality of the raw data followed by duplicate removal with FastUniq v. 1.1⁸⁴ which reduced the number of read pairs from 918,186,452 to 876,349,245 (Supplementary data (file 1)). We additionally applied clumpify (BBMap package) to perform deduplication. Both approaches resulted in comparable duplication rates. The average of sample and blank duplication rate for FastUniq is 5.3% and 10.6% for clumpify. The average duplication rate within samples is FastUniq=4.2% and clumpify=7.2 %, when considering blanks only the approaches identified a duplication rate of FastUniq=9.5% and clumpify=22.5 %, which indicates a better performance of clumpify to remove duplicates in the blanks. Afterwards we used Fastp [...]”.

New Supplementary data (file 1): This data file will be uploaded as Supplementary data. Figure caption: “Data processing steps of the bioinformatic pipeline. Read counts are given for all samples and blanks after each step.”

Reviewer Comment: I. 409-424: I think the approach for inclusion/exclusion of taxa is reasonable, however it would be useful to include the read counts that were removed by this ‘exclusion step’ from the initial read count at the classification at Family level step. Pls see additional comment on damage patterns.

Our response: We only excluded taxa on family level and did not remove genera or species read counts from a family. The number of read counts before and after the exclusion step on family level (for merged and unmerged read pairs) are presented as Supplementary data file 1.

Reviewer Comment: I. 425 Though I acknowledge that similar threshold approaches have been used on similar datasets and at least partly agree with the reasoning behind it, keeping 'only families that occurred in at least 3 samples with at least 10 counts' prior to re-sampling (or other form of normalization) increases the risk of removing true positives, rare taxa that may by chance (or because of sediment matrix effects on library building success) occur only in shallower sampled intervals of the core. Could the authors comment on this effect in relation to their data?

Our response: The manuscript builds upon the correlation analysis for which we require a stable signal of trends. Many taxa with single occurrences lead to the problem of zero-inflation to which the correlation analyses are sensitive. Opposed to the analysis of the presence/absence of single taxa, our goal was to decrease the false positives rate for which we had to consider the risk of missing true positives. Thus, for our sample size we determined a stable signal if families occurred in at least 3 samples and with at least 10 read counts prior to resampling.

Revised text L 500-506: “Finally, we kept only families that occurred in at least 3 samples with at least 10 counts, and resampled the dataset 500 times to a pelagic sample effort of 6,593 counts and a benthic sample effort of 1,839 counts per sample to circumvent bias arising from different sequencing depths across the samples¹ to capture the main compositional trend and allow for stable correlation signals. This measure was also effective in reducing zero-inflation of our dataset for the correlation network analysis and the few affected taxa have no significant role within the network (Supplementary Fig. 12).”

Reviewer Comment: Negative controls: It is good practice that the authors included a number of negative controls from the extraction step in the sequencing, however the number of reads from them is quite low (total of ~500 classified reads from all controls combined). It is actually common for controls to yield relatively high levels of reads assigned compared to samples, because human-related taxa are overall more commonly sequenced and thus more represented in reference databases (erroneously or not), and it would be also important to know many raw reads were sequenced for controls (maybe I have missed this information). Please provide the read counts after the different bioinformatic steps for the controls as per the samples.

Our response: We added the information for negative controls and prepared it in table form as Supplementary data (file 1). We also included a dedicated section (“3. Negative controls”) about read counts of the blanks into the revised Supplementary Information (see revised text below).

Revised text in the Supplementary Information: “Throughout the library preparation process the negative controls were treated in the same way as the samples, also in terms of PCR cycle numbers for the indexing step. We retrieved between 395,384 and 2,494,694 read pairs for the different negative controls of which between 0.01 and 30% were duplicates. After deduplication, trimming and filtering for low complexity, low quality, residual adapters and length only between 0.3 and 2.1% of the original read pairs were left. This suggests, our blanks contain very low levels of DNA and that increasing cycle numbers would predominantly amplify our duplication rate and increase the number of adapters in the sequencing data.”

Reviewer Comment: Ideally, one would provide similar sequencing effort to controls as for samples, however such a standard has not yet been established in the field and 'best practice' approaches from previous works have varied, none reaching this ideal (to my

knowledge). Furthermore, it can be challenging to obtain sufficient amounts of reads from controls for sequencing, that would indeed contain very little DNA (increasing the number of indexing PCR amplification cycles usually helps, but also result in high levels of duplication). That being said, **I do not have a reason to dispute** the authenticity of the results provided for the main taxa detected supporting the MS conclusions other than the relatively minor verifications I have pointed out elsewhere, particularly if the damage patterns from at least some of the representants of major groups of organisms can be established.

Our response: Throughout the library preparation process the negative controls were treated in the same way as the samples, also in terms of PCR cycle numbers for the indexing step. After deduplication, quality filtering and trimming of DNA reads (see Supplementary data, Data_processing_steps.xlsx) the blanks contained only between 0.3 and 2.1% of the original read pairs. Comparing this to the samples, we recovered about 56 to 86% of read pairs after the same bioinformatic steps. These results support that the blank controls contain very low levels of DNA and deeper sequencing would most probably result in the sequencing of more adapters and PCR duplicates. With regard to damage pattern analysis, we processed more taxa with the HOPS pipeline and included a dedicated section and additional figures on damage pattern analysis in the Supplement Information, where we also moved part of the lengthy description of the analysis.

Revised text in the Supplementary Information: “Throughout the library preparation process the negative controls were treated in the same way as the samples, also in terms of PCR cycle numbers for the indexing step. We retrieved between 395,384 and 2,494,694 read pairs for the different negative controls of which between 0.01 and 30% were duplicates. After deduplication, trimming and filtering for low complexity, low quality, residual adapters and length only between 0.3 and 2.1% of the original read pairs were left. This suggests, our blanks contain very low levels of DNA and that increasing cycle numbers would predominantly amplify our duplication rate and increase the number of adapters in the sequencing data.

Of the 2,183 read pairs (merged and unmerged) 1,519 read pairs were retrieved on family level after filtering (Supplementary Table 2). The blanks are dominated, as expected, by bacterial- and human-derived sequences (Supplementary Fig. 4). The majority of taxa found in the blanks is however only represented by a single sequence count in total (Supplementary Fig. 5). The composition of the blanks is highly similar as shown in the PCoA plot (Supplementary Fig. 2) and markedly different from the samples.”

Supplementary Figure 4. The composition of all blanks combined shows that reads assigned to Hominidae and bacterial families are dominant in blanks. All taxa with less than five total read counts in blanks were grouped together.

Reviewer Comment: l. 444: Insufficient number of reads assigned were found for the damage pattern analyses, however pelagic and benthic merged and unmerged counts sum up to potentially higher reads counts than merged and unmerged read counts for some samples. Please clarify.

Our response: HOPS can be run either with merged reads or unmerged reads, but not with merged and unmerged reads together. We included the following underlined sentence to improve clarity.

Revised text L 521ff: “HOPS can either run for merged or unmerged data, but it cannot combine both. To reach a sufficient number of read counts for running HOPS, we combined all merged reads (unmerged reads were not considered due to the insufficient number of assigned reads) from all sediment core samples into one fastq-file[...].”

5. Analytical approach: Your assessment of the strength of the analytical approach, including the validity and comprehensiveness of any statistical tests. If any aspect of the analytical approach is outside the scope of your expertise, please note this in your report or in the comments to the editor.

MY COMMENTS:

Reviewer Comment: I'm not very familiar with correlation networks, but I find the use here to be useful in highlighting differences in community assemblages and function under different climatic regimes. I would like for the authors to address or comment on a couple of questions that would relate to the structure of ancient metagenomic datasets could be

addressed in the MS: What is the sensitivity of correlation networks to detect non-linear relationships between taxa, especially over centennial to millennial timescales? Do biogeographical distribution patterns matter on this scale? How is this statistical approach handling the zero-inflated dataset? Trends in the dataset (vs environmental variables) could perhaps be further explored with GAMs (e.g., Simpson, G. L. Modelling palaeoecological time series using generalised additive models. *Frontiers in Ecology and Evolution* 149 (2018).

Our response: Below, we have addressed the three posed questions separately. In summary, Spearman method is a non-linear rank-based correlation, which detects similar (with Spearman non-linear) shifts of abundances over time. Therefore, it is better suited to address correlations in our dataset than, for example the Pearson correlation coefficient. Our additional analyses have shown that (1) zero inflation is not strongly pronounced in our dataset, (2) the affected taxa have no significant role within the network and (3) that zero-inflation doesn't have an effect on the creation of the network (Supplementary Fig. 12).

Reviewer Comment: *What is the sensitivity of correlation networks to detect non-linear relationships between taxa, especially over centennial to millennial timescales?*

New chapter in Supplementary Information: "Spearman method is a non-linear rank-based correlation, which detects monotonic relationships. This approach to infer a network focuses on the generation of a co-occurrence network, where links between taxa don't necessarily mean a direct relationship or even interaction, but rather similar (with Spearman non-linear) shifts of abundances over time. On this time scale each sample represents a local snapshot of a distinct state defined by major environmental drivers like SST or sea ice. Identifying actual relationships between the taxa would require a higher resolution of the time series. We made further efforts to generate networks based on the ecoCopula method which considers environmental covariates as well as mediator taxa. These analyses have shown that the co-occurrence network based on Spearman correlation have a Jaccard Similarity of 0.34 (positive correlations) with networks inferred with ecoCopula. This result shows that the statements of our analysis based on the co-occurrence network remain consistent, considering that the ecoCopula network has fewer edges as well as rewiring without a large impact on the network."

Do biogeographical distribution patterns matter on this scale?

Revised text L 203-207: "Despite removing the environmental effects of IP₂₅ and SSTs for network generation, the green module is mainly composed of families that are positively correlated with IP₂₅ in the Spearman network suggesting a strong co-occurrence pattern of these families based on biogeographic distribution patterns or other environmental factors not included here, such as nutrient availability, salinity, or light conditions."

How is this statistical approach handling the zero-inflated dataset?

Revised text L 500-504: "Finally, we kept only families that occurred in at least 3 samples with at least 10 counts, and resampled the dataset 500 times to a pelagic sample effort of 6,593 counts and a benthic sample effort of 1,839 counts per sample to circumvent bias arising from different sequencing depths across the samples¹ to capture the main compositional trend and allow for stable correlation signals. This measure was also effective

in reducing zero-inflation of our dataset for the correlation network analysis and the few affected taxa have no significant role within the network (Supplementary Fig. 12).”

Revised text in Supplementary Information:

To evaluate the influence of zero inflation on the network, we assessed the distribution of the abundance matrix (Supplementary Fig. 12b), analyzed which role zero-inflated taxa have within the resulting networks (Supplementary Fig. 12a). These analyses have shown that (1) zero inflation is not strongly pronounced in our dataset, (2) the affected taxa have no significant role within the network and (3) that zero-inflation doesn't have an effect on the creation of the network.

Supplementary Figure 12. (a) Spearman correlation network with nodes colored according to the number of zeros in the pelagic *sedDNA* time-series. (b) The histogram shows that the occurrences of zeros in the dataset is not extremely inflated. For better representation of zeros in the histogram, only abundance values from 0 to 50 are shown.

Our response: We appreciate the reviewer's suggestion and compared linear regression (see Figures a and b below) with GAMs (see Figures c and d below) to analyze the trends of the pelagic:benthic ratio with SSTs and IP₂₅. As the GAMs do not add any information, we decided to keep it simple by applying Spearman correlations instead of adding GAMs to our manuscript.

Figures for review only: Correlation of pelagic:benthic ratio with SST (a) and IP25 (b). Generalized additive model (GAM) plots showing the partial effects of SST (c) and IP25 (d) on the pelagic:benthic ratio.

Revised text L 297-301: “The ratio between counts assigned to pelagic and benthic families (pelagic:benthic ratio) is highest in samples dated to the late glacial and Younger Dryas compared to the warmer Bølling/Allerød and the Holocene (Fig. 4). The pelagic:benthic ratio shows a weak, negative trend with SSTs ($\rho = -0.39$, p -value = 0.052), indicating that fewer reads are assigned to benthic families in phases of warmer SSTs.”

Reviewer Comment: To what extent do the correlations between families may be affected by reference bias within closely related taxa, where for instance, only a few members of a family with differential degree of coverage may be available for mapping in NCBI? How could this affect interpretations (e.g., l. 155)?

Our response: Generally, if closely related species of the same family are represented by a reference genome, kraken2 handles the taxonomic assignments well, based on unique minimizers and a LCA approach (Wood et al. 2019). If a family is not represented in the database, a sequence cannot be assigned to this family. If a sequence can equally likely be assigned to two different families, the kraken2 algorithm will place the taxonomic assignment on a higher level. The completeness/incompleteness of the database however does not influence temporal trends. The most expected fish families based on their role in the modern western Bering Sea food web are the Salmonidae, Clupeidae, Gadidae. Our database analysis shows that these are well represented through reference genomes of several

genera (Supplementary Table 2, please find below the summary of the table). We dedicated a new section to the analysis of the database with special regard to fish in the Supplement. In summary, our data suggest that if a family is represented by a genome, kraken2 can manage the taxonomic assignment on this level. Therefore, we are confident that assignments to these families are valid.

Wood, D. E., Lu, J. & Langmead, B. Improved metagenomic analysis with Kraken 2. *Genome Biology* 20, 257 (2019).

Novel chapter in Supplementary Information:

Could correlations be influenced by a reference database bias of closely related taxa?

“To show whether correlations could be influenced by a reference database bias, we performed a test based on the group of fish, which have a sufficiently large number of read counts. The most expected fish families based on their role in the modern western Bering Sea food web are the Salmonidae, Clupeidae, and Gadidae³. Our database analysis shows that these are well represented through reference genomes of several genera (Supplementary Table 2). The analysis shows that for Clupeidae and Gadidae, only genera from the study area determine the signal on family level (Supplementary Fig. 7b,c). Among Clupeidae, 48 genera are represented in the database and 99.3% (2,877 counts) of the reads are assigned to the genus *Clupea*, while the rest (6 counts) are assigned to 5 other genera (Supplementary Fig. 7b). Among Gadidae, our reads were assigned to 8 of the 12 genera represented in the database. All of them except for *Micromesistius* (2 counts assigned) are present in the Bering Sea or the Pacific Arctic. The majority of reads were assigned to the genus *Gadus* (22,411 counts), which is driving the trends on family level. Notable, but few assignments were found for the genus *Boreogadus* (Polar cod; 107 counts). Today, Polar cod is a keystone species of the high-Arctic with a strong linkage to sea ice⁴, which supports their presence only during the late glacial in our samples.

The Bering Sea, and the Kamchatka area in particular, is a diversity hotspot for Salmonidae and provides spawning grounds for the genera *Oncorhynchus*, *Coregonus* and *Salvelinus*, which are all represented in our data on genus level. Of the overall 11 genera represented in the database, counts were mostly assigned to *Oncorhynchus* (2,648 counts), *Salvelinus* (613 counts), *Coregonus* (17,849 counts), and *Salmo* (18,090 counts), while only 6 counts were assigned to *Thymallus* and 2 to *Hucho*, which do not occur in the Bering Sea. The high assignment of reads to the genus *Salmo* could be a result of its high representation in the database or potential gaps in assemblies of other Salmonidae. This suggests that analyzing our data on family level is more appropriate than on lower taxonomic levels.

In summary, our data suggest that if a family is represented by a genome, kraken2 can manage the taxonomic assignment on this level. Therefore, we are confident that assignments to these families are valid.”

Supplementary Table 2 (summary). Read counts for planktic, pelagic, and benthic taxonomic groups before resampling.

Family	Genera in Family (FishBase)	Genera represented in NCBI db	Genome	Assemblies	mitochondrial reference genomes	nucleotide sequences	Unique minimizers	% of Actinopteri minimizers
Salmonidae	11	11	15	50	56	3616430	2005295682	5.21%
Gadidae	11	11	9	15	5	382120	395536200	1.03%
Clupeidae	52	48	6	21	58	258014	59088162	0.15%
Pleuronectidae	26	29	6	12	8	143210	44633306	0.12%

Supplementary Figure 7. Temporal signal of focal fish taxa at family and genus level: (a) Salmonidae, (b) Clupeidae, (c) Gadidae.

Reviewer Comment: Could the author also comment on the potential impact of downcore autocorrelation between samples for correlation networks and overall correlations with [UH15] environmental variables to strengthen the conclusions?

Our response: We assessed temporal autocorrelation for each taxon and can conclude that overall, our data are not prone to temporal autocorrelation. We have provided an overview of the autocorrelation plots for the most important families in the supplement (Supplementary Figs 8 and 9). And added the following sentence in the first paragraph of the statistical analysis section. Correlations with IP_{25} and SSTs (including adjusted p-values) are given in Supplementary Table 4.

Revised text L 530-532: “We tested each of the pelagic and benthic taxa for temporal autocorrelation using *acf* from the stats package and conclude that our trends and correlation networks are not influenced by temporal autocorrelation (Supplementary Figs. 8 and 9).”

Supplementary Figures 8 and 9:

Supplementary Figure 8. Temporal autocorrelation plots of some of the most abundant pelagic taxa among (a-d) fish, (e-f) cyanobacteria, (g-h) diatoms, and (i) chlorophytes suggest that downcore temporal autocorrelation between samples is not affecting the correlation network. The lag 0 autocorrelation is fixed at 1.

Supplementary Figure 9. Temporal autocorrelation plots of some of the most abundant benthic taxa suggest that downcore temporal autocorrelation between samples is negligible. The lag 0 autocorrelation is fixed at 1.

Reviewer Comment: All relative abundance datasets suffer from closed sum effects. To further support one of the main conclusions of the MS about the change between smaller and larger celled primary producers from the pelagic environment, the authors could test the strength of the correlation between **the ratio of read abundances of small:large phytoplankton and their environmental variables.**

Our response: A classification on cell size only is problematic, as we know that pico-sized chlorophytes, like the dominant family Bathycoccaceae, which are in our data also positively correlated with seasonal sea ice data, prefer colder conditions. However, as we are sure about the small cell size of bacteria, we calculated the ratio between reads assigned to phototrophic bacteria, which are classified as pico-sized bacterioplankton, and reads assigned to all remaining phototrophic protists. The ratio (photo.bacteria:photo.protist) is positively correlated with SSTs (Spearman's $\rho = 0.35$, p -value 0.088, see Figure for review only below) with a weak statistical significance which is potentially caused by the few number of samples. This positive relationship supports that pico-sized bacteria in general dominate over phototrophic protists under warming temperature which gives further support for the main conclusions drawn in our manuscript.

Revised text L 341-346: “Phytoplankton groups with large cell sizes such as diatoms are assumed to contribute more to carbon export than or other pico-sized plankton such as phototrophic bacteria⁷⁸, which are presumed to be mostly recycled by the microbial loop in the water column⁷⁹. A shift from larger phototrophic protists to smaller bacterioplankton in the course of a sea-ice loss could therefore alter the amount and quality of food sustaining benthic organisms⁸⁰ and the efficiency of the biological carbon pump⁶³.

Figure for review only: Correlation of photo.bacteria:photo.protist ratio with IP25 (a) and SST (b).

Revised Figure 2

Figure 2. Pelagic families in the *sedaDNA* record of core SO201-2-12. Only families represented in at least 3 samples and by at least 1.5% proportion of read counts after resampling are shown. The dendrogram (based on constrained hierarchical clustering) and IP_{25} concentrations from the same core⁴⁰ and sea-surface temperature (SST_{UK37}) reconstructions of the Northwest Pacific⁴¹ (blue) and Northeast Pacific⁴² (black) are added for comparison. Families which are significantly, positively correlated with IP_{25} or SST_{UK37} are highlighted in blue or red, respectively. The pelagic:benthic ratio (black line) shows the proportion of reads assigned to pelagic families in relation to reads assigned to benthic families. The ratio between reads assigned to phototrophic bacteria and reads assigned to phototrophic protists is given as blue-green line. Black triangles next to the IP_{25} record mark radiocarbon-dated calibrated ages⁴¹. LGM = Last Glacial Maximum, HS1 = Heinrich Stadial 1, B/A = Bølling/Allerød, YD = Younger Dryas.

Revised Figure 4

Figure 4. *SedaDNA* record of benthic families of core SO201-2-12. Only families represented in at least 3 samples and by at least 1.5% proportion of read counts after resampling are shown. The dendrogram (based on constrained hierarchical clustering) and

IP₂₅ concentrations from the same core⁴⁰ and sea-surface temperature (SST_{UK37}) reconstructions of the Northwest Pacific⁴¹ (blue) and Northeast Pacific⁴² (black) are added for comparison. Zosteraceae, which are significantly, positively correlated with SST_{UK37}, are highlighted in red. The pelagic:benthic ratio shows the proportion of reads assigned to pelagic families in relation to reads assigned to benthic families. The ratio between reads assigned to phototrophic bacteria and reads assigned to phototrophic protists is given as blue-green line. Black triangles next to the IP₂₅ record mark radiocarbon-dated calibrated ages⁴¹. LGM = Last Glacial Maximum, HS1 = Heinrich Stadial 1, B/A = Bølling/Allerød, YD = Younger Dryas.

6. Suggested improvements: Your suggestions for additional experiments or data that could help strengthen the work and make it suitable for publication in the journal. Suggestions should be limited to the present scope of the manuscript; that is, they should only include what can be reasonably addressed in a revision and exclude what would significantly change the scope of the work. The editor will assess all the suggestions received and provide additional guidance to the authors.

MY COMMENTS:

I have already provided suggestions in previous sections that would, in my opinion, strengthen the manuscript, and added a few more below.

Reviewer Comment: A major point: **damage patterns** were assessed for diatoms, but overall conclusions would be stronger if the other major groups found were also assessed like fish (codfish, salmon, herring), cyanobacteria and copepods (the latter may have too low counts). Cnidaria, Echinodermata, Mollusca also had potentially sufficiently high number of reads. Where counts are of sufficient abundance, **consider also assessing the damage patterns for some of the main groups that were excluded to show the difference in damage.**

Our response: We now assessed damage patterns for two benthic taxa (*Asteria rubens*, *Pecten maximus*), four eukaryotic algae (*Chaetoceros simplex*, *Thalassiosira pseudonana*, *Fragilariopsis cylindrus*, *Bathycoccus prasinos*), one prokaryotic algae (*Synechococcus*), three pelagic fish (*Oncorhynchus kisutch*, *Salmo trutta* and *Gadus morhua*) and for two taxa that we excluded from the data: 1. freshwater fish *Cyprinus carpio* (excluded because we assume a taxonomic mis-classification) and one dominant bacteria *Marinobacter* (excluded because we did not focus on bacterial composition). To apply damage pattern analysis to non-photosynthetic organisms, we used the nt/nr database instead of the cp database (which was used in the former manuscript version) in the automated HOPS pipeline (Huebler et al. 2019). The pipeline uses MaltExtract function to identify the ratio of C>T (5'end) and G>A (3'end) substitutions for a selection of DNA reads (pre-classified as *ancient*) that have one mismatching lesion in their first 5 bases from either end. Damage pattern profiles of pre-selected *ancient* reads are provided for three time periods 1.08-5.6 cal kyr BP (=dataset1), 6.3-12.6 cal kyr BP (=dataset2) and 13.6-19.9 cal kyr BP (=dataset3) and are placed in the Supplementary information (Supplementary Figure 6a-c). The profiles support an increase of damage pattern with increasing age of the samples. Also excluded taxa show the same trend. As we excluded taxa based on either taxonomic misclassification (like for *Cyprinus*) or

because of non-focus taxa (like *Marinobacter*) we expect ancient DNA damage patterns for such taxa as well.

Revised text in the Supplementary Information: “We confirmed authenticity of pelagic and benthic focal taxa via damage pattern analysis using the automated HOPS pipeline². The pipeline was run in full mode against the NCBI nt database (downloaded 03.11.2020) with options "filter" and "alignment" set to “ancient” and "1", respectively. The required index for the database was built using malt-build with default settings and the newest MEGAN6³ mapping file (megan-nucl-map-Jul2020.db). The malt alignment of only merged reads against the full nt database was run for two benthic taxa (*Asteria rubens*, *Pecten maximus*), four eukaryotic algae (*Chaetoceros simplex*, *Thalassiosira pseudonana*, *Fragilariopsis cylindrus*, *Bathycoccus prasinus*), one prokaryotic algae (*Synechococcus*), three pelagic fish (*Oncorhynchus kisutch*, *Salmo trutta* and *Gadus morhua*) and for two taxa that we excluded from the data: 1. freshwater fish *Cyprinus carpio* (excluded because we assume a taxonomic mis-classification) and one dominant bacteria *Marinobacter* (excluded because we did not focus on non-phototrophic bacterial composition). The pipeline uses MaltExtract function to identify the ratio of C>T (5’end) and G>A (3’end) substitutions for a selection of DNA reads (pre-classified as ancient) that have one mismatching lesion in their first 5 bases from either end. Damage pattern profiles of pre-selected ancient reads are provided for three time periods 1.08-5.6 cal kyr BP (Supplementary Figure 6a), 6.3-12.6 cal kyr BP (Supplementary Figure 6b) and 13.6-19.9 cal kyr BP (Supplementary Figure 6c) and show increasing accumulation of damage over the three binned timeframes for pelagic and benthic families.”

Revised figures:

b

Supplementary Figure 6a-c. Damage profiles of a: dataset1 (1.08-5.6 cal kyr BP), b: dataset2 (6.3-12.6 cal kyr BP) and c: dataset3 (13.6-19.9 cal kyr BP). C>T substitutions in 5' direction and G>A substitutions in 3' are given for each taxon analyzed. Non-C>T substitutions in 5' direction and non-G>A substitutions in 3' direction are given to estimate the noise (gray color). The general color code for benthic taxa (*Asterias rubens* and *Pecten maximus*) is brownish, eukaryotic algae (euk algae) are greenish, pelagic fish are blueish and bacteria are colored purple. Overall, the profiles show increasing accumulation of damage over the three binned timeframes for pelagic and benthic families.

Reviewer Comment: Because it is particularly prone to reference bias because of its high representation in NCBI as a model organism, to further validate the signal of Salmon (vs that of other fish), the authors could consider providing information on **whether reads fall within unique parts of its genome** (vs other fish).

Presenting rarefaction curves for each sample and for both pelagic and benthic groups in supplementary would help assessing sampling completeness at the Family level.

Our response regarding Salmonidae: The strength of kraken2 addresses exactly this point. kraken2 connects short genomic substrings (minimizer, based on k-mers) with the lowest common ancestor (LCA) taxa. When 2 distinct k-mers from distantly related genomes share the same minimizer this can either lead to higher LCA values (in case both genomes are part of the database) or incorrect assignments (if one of the genomes is missing). However, the latter problem is stronger on species and genus level than on higher level

classifications like family level, especially as our focal families are represented by at least one reference genome. We assessed the state of the database and tested the signal of Salmonidae versus that of other fish families (here the most important families of the Western Bering Sea food web: Gadidae, Clupeidae, Pleuronectidae). These families are covered by several reference genomes across several families (a more detailed version of the summary table presented here is available as a chapter in the Supplementary Information, p.12), but almost all genera are represented in the database.

Novel chapter in the Supplementary Information:

“The strength of kraken2 addresses exactly this point. kraken2 connects minimizers (short substrings used for binning of k-mers) with the lowest common ancestor (LCA) taxa. When 2 distinct k-mers from distantly related genomes share the same minimizer, this can either lead to higher LCA values (in case both genomes are part of the database) or incorrect assignments (if one of the genomes is missing). However, the latter problem is stronger on species and genus level than on higher level classifications like family level, especially as our focal families are represented by at least one reference genome. Our method is also less prone to bias by taxa with a high representation in datasets than a BLAST-based approach. BLAST returns the n-best hits, but a model species can dominate the list of hits and a more closely related species with a better hit might be missed if it comes further down in the database and the threshold of hits has been reached.

We assessed the state of the database and tested the signal of Salmonidae versus that of other fish families (here the most important families of the Western Bering Sea food web: Gadidae, Clupeidae, Pleuronectidae). These families are covered by several reference genomes across several genera (Supplementary Table 3). Furthermore, almost all the corresponding genera from our study area are represented in the database.

One important group that is present in our data, but missing in our correlation analyses, due to very few read counts is the family Pleuronectidae, which contains ground fish such as the Pacific halibut (*Hippoglossus stenolepis*; genome available), Arrowtooth flounder (*Atheresthes stomas*; no references in NCBI) and Greenland turbot (*Reinhardtius hippoglossoides*; genome available). The proportion of Pleuronectidae minimizers (0.12%) among Actinopteri is only slightly lower than the proportion of minimizers for Clupeidae (0.15%). Hence other factors than a reference database bias could be responsible for the lower number of read counts assigned to Pleuronectidae, such as lower past abundance compared to Salmonidae and Clupeidae. Salmonidae have many more unique minimizers in the database than the other fish families (5.21% of Actinopteri minimizers).”

Supplementary Table 3. Read counts for planktic, pelagic, and benthic taxonomic groups before resampling. Here only the summary of the table is provided.

Family	Genera in Family (FishBase)	Genera represented in NCBI db	Genome	Assemblies	mitochondrial reference genomes	nucleotide sequences	Unique minimizers	% of Actinopteri minimizers
Salmonidae	11	11	15	50	56	3616430	2005295682	5.21%
Gadidae	11	11	9	15	5	382120	395536200	1.03%
Clupeidae	52	48	6	21	58	258014	59088162	0.15%
Pleuronectidae	26	29	6	12	8	143210	44633306	0.12%

Our response regarding rarefaction curves: We plotted the rarefaction curves and included them with a small description into section “2. Filtering of sequencing data” of the revised supplement.

Added text in the Supplementary Information:

“We checked completeness of pelagic and benthic taxa within the samples on family level with rarefaction curves. The rarefaction curves reflect that many taxa were cautiously filtered out to get stable signals over time (Supplementary Figure 3). For our purposes however, reaching a saturation of the curves is not essential, because we are not investigating presence/absence of taxa, but rather semi-quantitative trends in ecosystem composition.”

Supplementary Figure 3. Rarefaction curves for pelagic and benthic families before resampling. The vertical line indicates the lowest number of read counts for each of the datasets (pelagic = 6,593 counts; benthic = 1,839 counts) to which the two datasets were resampled, while horizontal lines mark the number of families detected for each sample after rarefaction

7. Clarity and context: Your view on the clarity and accessibility of the text, and whether the results have been provided with sufficient context and consideration of previous work. Note that we are not asking for you to comment on language issues such as spelling or grammatical mistakes.

MY COMMENTS:

The manuscript was a nice read, and I have made only a few suggestions that may improve clarity and context for the wider readership.

Reviewer Comment: Intro: Consider providing a bit more context to support the validity of 'climate transition' and the analogous environments compared, by establishing the current, historical and paleo sea-ice conditions, and those expected under future scenarios more clearly (e.g., how many months of sea ice? Extent? spatial patterns?). **How do the 'the phases of the Bølling-Allerød and the Holocene' compare with now/expected scenario in relevant regions?** This relates to my last comment in the significance section.

Our response: We included more information about known sea-ice condition and recent changes in sea-ice decline in the Introduction.

Revised text L 108-113: "The paleoenvironmental backbone is based on previous multi-proxy reconstructions of the coring site derived from this core, including diatom microfossils and the biomarker IP₂₅, which show that the coring site was covered by seasonal sea-ice during the Last Glacial Maximum (LGM), Heinrich Stadial 1, and the Younger Dryas, while the phases of the Bølling-Allerød and the Holocene were predominantly sea-ice free^{12,16}."

L 44-46:

"Long-term trends in the Bering Sea show an ongoing decline of sea-ice duration which is projected to decline further due to later freeze-up (by 20 to 30 days) and earlier break-up (by 10-20 days) until the middle of this century³."

Added citation:

3. Wang, M., Yang, Q., Overland, J. E. & Stabeno, P. Sea-ice cover timing in the Pacific Arctic: The present and projections to mid-century by selected CMIP5 models. *Deep Sea Research Part II: Topical Studies in Oceanography* **152**, 22–34 (2018).

Reviewer Comment: I. 26: This sentence: 'This demonstrates the need for a better understanding of ecosystem changes during the transition from seasonal sea-ice to ice-free conditions due to ongoing climate warming.' I'm actually not certain that 'it' does. Perhaps restructure this part of the intro for better flow.

Our response: We rephrased the sentence.

Revised text L 54-58: "A better understanding of ecosystem changes during the transition from seasonal sea-ice to ice-free conditions due to ongoing climate warming is needed. Despite extensive monitoring efforts, we still have scarce knowledge on long-term ecosystem responses to climate transitions for many taxonomic groups across the food web, particularly zooplankton, fish, non-fossilizing algae, and benthic organisms such as tunicates, starfish, or macrophytes."

Minor comments:

Reviewer Comment: I. 4: 'service' is phrased wrong, should be 'services delivered by' or something along those lines.

Our response: We changed it accordingly.

Revised text L 362-363: "Sea ice is a key factor for the functioning and services provided by polar marine ecosystems."

Reviewer Comment: I. 20. 'timing', do you mean seasonal timing?

Our response: We changed it accordingly.

Revised text L 40-43: "This will likely change the seasonal timing, biomass, and composition of algal blooms, which play a central role in trophic interactions by supporting food webs, including benthic communities as primary producers²."

Reviewer Comment: l. 21: 'play a central role in trophic interactions' could be more specific by adding in what way they play a central role e.g., 'supporting foodwebs as primary producers' or food source etc.

Our response: We added the information.

Revised text L 40-43: "This will likely change the seasonal timing, biomass, and composition of algal blooms, which play a central role in trophic interactions by supporting food webs, including benthic communities as primary producers²."

Reviewer Comment: l. 22: Try to be more specific for the wide readership: 'air temperatures' I assume could be written as e.g., 'high air temperatures' or 'record high air temperatures' if that applies?

Our response: We changed the sentence accordingly.

Revised text L 46-48: "Extreme events, such as the lowest sea-ice extent in the Bering Sea over the last 5,500 years⁴ in 2018 with persistent southerly warm winds, allow for assessments of immediate ecosystem responses to sea-ice loss."

Reviewer Comment: l. 24-26: What were those shifts? Could be more specific here.

Our response: We added the information.

Revised text L 48-53: "Cascading effects through the food web have been recorded, possibly resulting from reduced energy transfer from lower to upper trophic levels⁵: A late spring phytoplankton bloom and a scarcity of large, lipid-rich copepods led to decreasing abundance of young pollock and other forage fish that were later linked with seabird reproductive failures as well as seabird and seal mortality events in 2018 and 2019⁶."

Reviewer Comment: l. 32: Add 'latest' IPCC report

Our response: We adjusted the citation.

"Intergovernmental Panel on Climate Change (IPCC). *The Ocean and Cryosphere in a Changing Climate: Special Report of the Intergovernmental Panel on Climate Change*. (Cambridge University Press, 2022). doi:10.1017/9781009157964."

Reviewer Comment: l. 216: Please add the taxa groups names (not only common names), as per above in text.

Our response: Changed accordingly.

Revised text L 295-298: "Storms can whirl up and redistribute benthic inhabitants^{61,62}, and the very steep Kamchatka slope may foster downslope transport, which might also explain the considerable share of reads assigned to near-shore macrophytes (10% of benthic reads) like seagrass (Zosteraceae, 0.8%), brown (Phaeophyta, 2.9%) and red macroalgae (Rhodophyta 10.9%) in our record.

Reviewer Comment: l. 233: specify whether observations were from paleo or modern datasets.

Our response: It is based on modern observations, but we agree that this sentence is a bit unclear.

Revised text L 314-316: "Important benthic-pelagic relationships are established between seagrass and salmon as well as Pacific herring including foraging opportunities and

spawning grounds^{76,77}, which are reflected in our data by similar temporal co-occurrence patterns.”

8. References: Your view on whether the manuscript references previous literature appropriately.

MY COMMENTS:

References seemed appropriate and up to date.

9. Your expertise: Please indicate any particular part of the manuscript, data or analyses that you feel is outside the scope of your expertise, or that you were unable to assess fully.

MY COMMENTS:

My expertise lies in paleoecology of Arctic freshwater ecosystems and foodwebs, paleoclimatology as well as shotgun metagenomics, and is therefore more limited for marine environments and organisms beyond characteristics that would apply to aquatic ecosystems more broadly, like major ecological functional groups and foodweb interactions.

Alexandra Rouillard

Reviewer #3 (Remarks to the Author):

The manuscript entitled: “Marine ecosystem shifts with deglacial sea-ice loss inferred from ancient DNA shotgun sequencing” suggest that the compositions and structure of a marine ecosystem is different between a period with and without sea ice in the Bearing Sea by looking at fossil record data.

I found the study interesting, but I had concerns about the methodology used especially with regards to the statistical analyses performed in the study. I listed my concerns about the statistical analyses below.

Reviewer Comment: 1. L474-476: “As for both variables the first sample age as well as the last two for SST and the last four for IP_25 data could not be interpolated, we calculated the mean of the first and last three interpolated values and replaced the missing data.”

Why was it not possible to interpolate these samples? Please give detailed explanations. Also, by using “...the mean of the first and last three interpolated values...” as a way to impute the missing data, the imputed missing values could lead to misinterpretation of the results. This is especially true because there are so few samples in the data. To put it differently, for IP_25, 20% of the data (5 out of 25 samples) are imputed. In this context, using an imputation approach that accounts for temporal autocorrelation (e.g. an Autoregressive integrated moving average [ARIMA] model) would be much more appropriate. In addition, the model used for the imputation should be validated using SST and IP_25 data gathered elsewhere to check if the trend makes sense.

Our response: We thank the reviewer for the suggestion. For IP₂₅ the record does not extend to 20 kyr BP, therefore we previously applied an interpolation method which uses a moving average, with the result that the first values could not be retrieved. We interpolated the variables SST and IP₂₅ (Supplementary Table 3, Supplementary Fig. 10) to address this issue with a new method. Our extrapolation at the core base is based on the assumption of stable sea-ice conditions between 20 and 18 ka. This assumption is consistent with several proxies records from several cores across the Bering Sea and the Sea of Okhotsk which all show that winter sea ice coverage was stable during the time between 18 to 20 kyr BP^{11,12,14}. We extrapolated a stable value of IP₂₅ toward the base and are confident that this is the best knowledge based extrapolation method and also superior to deleting the entire samples from data analyses.

Revised text L 70-73 “A regional palaeoceanographic framework for past sea-ice cover in the Bering Sea is based on reconstructions using the highly branched isoprenoid alkane IP₂₅¹² which is produced by diatoms that are bound to a life in sea ice¹⁰ and diatom microfossils¹¹.”

L 108-113:

“The paleoenvironmental backbone is based on previous multi-proxy reconstructions of the coring site derived from this core, including diatom microfossils and the biomarker IP₂₅, which show that the coring site was covered by seasonal sea-ice during the Last Glacial Maximum (LGM), Heinrich Stadial 1, and the Younger Dryas, while the phases of the Bølling-Allerød and the Holocene were predominantly sea-ice free^{11,15}”

Included citations:

10. Belt, S. T. & Müller, J. The Arctic sea ice biomarker IP₂₅: a review of current understanding, recommendations for future research and applications in palaeo sea ice reconstructions. *Quaternary Science Reviews* **79**, 9–25 (2013).
11. Matul, A. G. Probable limits of sea ice extent in the northwestern Subarctic Pacific during the last glacial maximum. *Oceanology* **57**, 700–706 (2017).
12. Méheust, M., Stein, R., Fahl, K. & Gersonde, R. Sea-ice variability in the subarctic North Pacific and adjacent Bering Sea during the past 25 ka: new insights from IP₂₅ and UK'37 proxy records. *Arktos* **4**, 8 (2018).
13. Harada, N., Shin, K. H., Murata, A., Uchida, M. & Nakatani, T. Characteristics of alkenones synthesized by a bloom of *Emiliania huxleyi* in the Bering Sea. *Geochimica et Cosmochimica Acta* **67**, 1507–1519 (2003).
16. Méheust, M., Stein, R., Fahl, K., Max, L. & Riethdorf, J.-R. High-resolution IP₂₅-based reconstruction of sea-ice variability in the western North Pacific and Bering Sea during the past 18,000 years. *Geo-Marine Letters* **36**, 101–111 (2016).

Revised text L 558-562: “SST²⁵ and IP₂₅⁴¹ data were interpolated to our sample ages using R function *approx*⁹⁵. As for the last 2 samples IP₂₅ data could not be extrapolated, we

calculated the mean of the last three interpolated values and replaced the missing data based on regional sea ice reconstructions for the LGM²² (Supplementary Fig. 10).”

Novel chapter in the Supplementary Information: “SST and IP₂₅⁵ data were interpolated to our sample ages using R function *approx*⁶. As for the last 2 samples IP₂₅ data could not be extrapolated, we calculated the mean of the last three interpolated values and replaced the missing data based on our knowledge about past sea ice coverage during the time for the study area from diatom microfossil- and biomarker-derived sea ice reconstructions for the LGM^{7,8}.”

Included citations:

6. R Core Team. *R: A language and environment for statistical computing*. (R Foundation for Statistical Computing, 2018).
7. Maier, E. *et al.* North Pacific freshwater events linked to changes in glacial ocean circulation. *Nature* **559**, 241–245 (2018).
8. Matul, A. G. Probable limits of sea ice extent in the northwestern Subarctic Pacific during the last glacial maximum. *Oceanology* **57**, 700–706 (2017).

Supplementary Table 3. Interpolated values for IP₂₅ (seasonal sea ice indicator) and SSTs (late summer/early fall temperature indicator).

Age (cal kyr BP)	IP ₂₅ (µg g ⁻¹ sediment)	SST _{UK'37} (°C)
1.08	0.002963148	10.62408724
1.75	0.002626213	10.64911873
2.36	0.002154964	11.05654296
2.99	0.001993602	10.63779093
3.66	0.003891224	10.34516472
4.31	0.002803875	9.96946128
5.6	0.004077585	10.34702008
6.26	0.002729124	9.783810255
7.56	0.002782423	10.06475664
8.29	0.003998239	10.46982775
9.84	0.002321933	10.86202396
10.73	0.002810701	10.57618842
11.17	0.006094	10.3278189
11.56	0.004425627	10.04186022
12.13	0.028385708	7.14345425

12.61	0.020262972	7.313923674
13.62	0.004270033	9.318954252
14.03	0.005828588	8.921210969
15.6	0.022091014	6.70318985
15.91	0.018019808	6.59466641
16.5	0.029575612	6.992239568
17.34	0.01472526	5.695324051
18.39	0.018176429	7.306682467
19.3	0.018176429	7.27120261
19.9	0.018176429	7.466771735

Supplementary Figure 10. Temporal evolution of IP25 and SSTs with interpolated and extrapolated values marked by red dots.

Reviewer Comment: 2. Currently, the analysis carried out suppose that each sample is completely independent from the others. In other words, it assumes that there are no temporal autocorrelations in the data. A test of temporal autocorrelation in the taxon data should be carried out to make sure the interpretation is not affected by temporal autocorrelation. If there is temporal autocorrelation in the data, it will need to be accounted for in the analysis and interpretation. If there is no temporal autocorrelation in the taxon data, it would be an interesting addition to the results that is valuable ecologically because it would refine the understanding gain from this system (e.g. a deeper understand of temporal turnover, or temporal beta diversity, could be assessed). There exist a few options to test for temporal autocorrelation for multivariate data, two that come to mind is to use the approach proposed by Legendre and Gauthier (2013) or Dray et al. (2008) (both proposed techniques have been implemented in the adespatial R package). Another approach is to test the

temporal autocorrelation for each taxon independently, this could be done, e.g. with a Moran's I (Moran 1948) or an ARIMA model.

Our response: We assessed temporal autocorrelation for each taxon and can conclude that overall, our data are not prone to temporal autocorrelation. We have provided an overview of the autocorrelation plots for the most important families in the supplement (Supplementary Figs 8 and 9). And added the following sentence in the first paragraph of the statistical analysis section.

Revised text L 533-535: "We tested each of the pelagic and benthic taxa for temporal autocorrelation using *acf* from the stats package and conclude that our trends and correlation networks are not influenced by temporal autocorrelation (Supplementary Figs. 8 and 9)."

Supplementary Figures:

Supplementary Figure 8. Temporal autocorrelation plots of some of the most abundant pelagic taxa among (a-d) fish, (e-f) cyanobacteria, (g-h) diatoms, and (i) chlorophytes suggest that downcore temporal autocorrelation between samples is not affecting the correlation network. The lag 0 autocorrelation is fixed at 1.

Supplementary Figure 9. Temporal autocorrelation plots of some of the most abundant benthic taxa suggest that downcore temporal autocorrelation between samples is negligible. The lag 0 autocorrelation is fixed at 1.

Reviewer Comment: 3. L463-466: “We calculated pairwise Spearman rank correlation coefficients (ρ) using `corr.test` including Benjamini-Hochberg correction for multiple testing from the `psych` package v. 2.0.1291, and those exceeding at least 0.4 (adjusted p -value < 0.05) were used for undirected network generation by the `igraph` package v. 1.2.692 without self-loops and isolated nodes.” Technically, building the network based on Spearman correlation is problematic because it does not explicitly account for the multivariate nature of the data and as such it is likely to lead to biased results and as such wrong interpretations. In this respect, the approach proposed by Popovic et al. (2019) is much better adapted to the question and data of the present study.

Our response: We thank the reviewer for the suggestion. Spearman rank correlations are widely used in microbial ecology (Matchado et al. 2021) and have been shown to better reveal the complexity of species associations compared to compositional analyses of communities and to produce reliable results when different approaches are used for network inference (Gao et al. 2022). We augmented our study by the proposed alternative network inference approach (`ecoCopula`), which controls for mediator taxa and environmental effects. Both the Spearman- and the `ecoCopula`-network have a high overlap of edges (38% of all and 41% of only positive edges are similar), which is significantly different from our randomized null-model (Supplementary Fig. 13b). Therefore, we would like to present the original Spearman-networks in the main manuscript, as the main focus is to analyze co-occurrence dynamics resulting from environmental changes over time. We included additional results from the `ecoCopula` network analysis to the Results and Discussion sections and dedicated a section in the supplement “Gaussian copula graphical model

(ecoCopula network) and comparisons with Spearman network” to describe our approach and findings in more detail.

Gao, C. et al. Co-occurrence networks reveal more complexity than community composition in resistance and resilience of microbial communities. *Nat Commun* 13, 3867 (2022).

Matchado, M. S. et al. Network analysis methods for studying microbial communities: A mini review. *Computational and Structural Biotechnology Journal* 19, 2687–2698 (2021).

Revised text L 195-208: “As Spearman correlations cannot separate environmental effects from intrinsic interactions among taxa, we used ecoCopula to test how mediator taxa and environmental conditions (IP₂₅ and SSTs used as covariates) affect associations between taxa. After accounting for the effects of IP₂₅, SSTs and mediator taxa a high overlap of edges (34% of positive associations) can be found between the two network approaches which is significantly different from the randomized null-model (Supplementary Fig. 13b). The ecoCopula network is composed of 167 nodes and 474 edges, which is in a similar range compared to the edge density of the Spearman network. Two modules are dominated by a specific functional group: the green and red modules by diatoms and the violet module by fish, while the other modules are more mixed (Supplementary Fig. 13a). Despite removing the environmental effects of IP₂₅ and SSTs for network generation, the green module is mainly composed of families that are positively correlated with IP₂₅ in the Spearman network suggesting a strong co-occurrence pattern of these families based on biogeographic distribution patterns or other environmental factors not included here, such as nutrient availability, salinity, or light conditions.”

Revised text L 545-553: “The GCGM network was built using the package ecoCopula²⁷. The *stacked sdm* function was used to build the stacked species regression model with interpolated SSTs and IP₂₅ values as covariates. Then a Gaussian copula graphical lasso was fitted to the co-occurrence data with the function *cgr* and a lambda of 0.51, which was determined empirically (Supplementary Fig. 13). To test the similarity between the Spearman-network and the ecoCopula network, we computed the Jaccard-index over the edges in the networks and compared the results with a randomized null-model. The null-model was generated via double-edge swaps on the generated ecoCopula networks, which was performed ten times for each lambda value (shrinking parameter: with increasing lambda the networks become less dense) between 0.1 and 1.0.”

New citation:

27. Popovic, G. C., Warton, D. I., Thomson, F. J., Hui, F. K. C. & Moles, A. T. Untangling direct species associations from indirect mediator species effects with graphical models. *Methods in Ecology and Evolution* 10, 1571–1583 (2019).

Novel chapter in the Supplementary Information:

“Pelagic networks

Both the Spearman- and the ecoCopula-network have a significant overlap of edges (29% of all and 34% of only positive edges are similar) and are significantly different from the randomized null-model (Supplementary Fig. 13b).

The edge density of the ecoCopula network depends on the shrinking parameter lambda for which an optimal value was determined empirically. Lambda can range from 0 to 1 and we computed ecoCopula networks for each lambda value between 0.1 and 1.0 in 0.1 increments. We show that for increasing lambda values the network becomes less dense (Supplementary Fig. 13b). For each generated ecoCopula network we compared the overlap of edges (all associations, only positive associations, and only negative associations) with the Spearman-network. The highest similarity between the two network approaches was found at a lambda value of 0.51. Furthermore, we could show that the similarity of edges is higher for only positive associations (34%) compared to all (29%) or only negative associations (0.24%). The significance was tested using a null-model which was generated via double-edge swaps on the ecoCopula networks. This was performed ten times for each lambda value between 0.1 and 1.0 (Supplementary Fig. 13b). The network edge density for the ecoCopula network at lambda of 0.51 results in a comparable number of edges to the Spearman network, which is slightly lower due to the removal of associations resulting from correlation with environmental factors via the included covariates IP₂₅ and SST in the stacked species regression model of ecoCopula.”

Supplementary Figure 13. (a) Pelagic ecoCopula network of positive associations for lambda of 0.51 with colors highlighting the different modules. (b) Edge overlap (Jaccard similarity) of ecoCopula networks computed for lambda values between 0.1 and 1.0 with the pelagic Spearman correlation-based network. The number of edges in the Spearman network (positive and negative associations) are indicated by the horizontal dotted line while the number of edges decreases with increasing lambda for the ecoCopula networks. The vertical line indicates the optimal lambda (0.51) at which the edge overlap between the two approaches is highest.

Reviewer Comment:4. L467-468: “We included only positive correlations as negative correlations were rare and did not affect the structure of the network.” **Negative**

correlations should be presented and discussed. They present valuable information that could lead to a better understanding of the studied system.

Our response: We presented negative correlations with environmental variables in Supplementary Tables 5 and 6 and networks with negative correlations in Supplementary Figs. 14 and 15.

Revised text L 553-555: “Co-occurrence networks with negative associations for both Spearman and ecoCopula are presented in Supplementary Figs. 14 and 15, respectively.”

Supplementary Figure 14. Network showing negative associations of the pelagic co-occurrence dataset based on Spearman rank correlation coefficients (> 0.4 , adjusted p-value < 0.1). Negative associations are found between families that are positively correlated with IP25 (blue nodes) and families that are positively correlated with SSTs (red nodes).

Supplementary Figure 15. Network showing negative associations of the pelagic ecoCopula network with colors representing functional groups.

Reference

Dray, S., D. Bauman, F. G. Blanchet, D. Borcard, S. Clappe, G. Guenard, T. Jombart, G. Larocque, P. Legendre, N. Madi, and H. H. Wagner. 2021. adespatial: Multivariate Multiscale Spatial Analysis.

Dray, S., S. Said, and F. Debias. 2008. Spatial ordination of vegetation data using a generalization of Wartenberg's multivariate spatial correlation. *Journal of Vegetation Science* 19:45–56.

Legendre, P., and O. Gauthier. 2014. Statistical methods for temporal and space–time analysis of community composition data. *Proceedings of the Royal Society B: Biological Sciences* 281:20132728.

Moran, P. A. P. 1948. The interpretation of statistical maps. *Journal of the Royal Statistical Society Series B-Statistical Methodology* 10:243–251.

Popovic, G. C., D. I. Warton, F. J. Thomson, F. K. C. Hui, and A. T. Moles. 2019. Untangling direct species associations from indirect mediator species effects with graphical models. *Methods in Ecology and Evolution* 10:1571–1583.

REVIEWER COMMENTS

Reviewer #1 (Remarks to the Author):

Thank you for addressing all my comments.

I only have a few editing suggestions for the revised text sections, these are to improve clarity in a few instances. I am unable to comment on network analyses performed in this study as I am not familiar with these.

Abstract

Add ocean/region (e.g. "Bering Sea", or "Arctic") after Kamchatka, so it clear immediately which polar region (first sentence) this refers to.

Introduction

L. 22: "This will likely change the seasonal timing, biomass, and composition of algal blooms, which play a central role in trophic interactions by supporting food webs, including benthic communities as primary producers²." - Do you mean "and" primary producers? – As the algal blooms are basically the primary producers, you could leave what is after the comma off (or at least the last 3 words).

L25. Has the word decline twice in it, can you rephrase? Eg "reduction or retreat" of sea-ice?

L. 41: well documented

L. 49: sea ice: choose "sea-ice" or "sea ice" throughout text for consistency.

L49ff: "Marine sediments are a natural archive of climate history from which sea ice can be reconstructed via proxy records, such as from biomarkers¹⁰ or microfossil remains¹¹. A regional palaeoceanographic framework for past sea ice coverage in the Bering Sea is based on reconstructions using the highly branched isoprenoid alkane biomarker IP25, ¹², produced by diatoms bound to a life in seasonal sea ice¹⁰, and diatom microfossils¹¹. Alkenone-based sea surface temperatures (SSTUK³⁷),

which are produced by haptophytes, and can add complementary information about SSTs of the late summer/early fall season¹³.”

- I would suggest to simplify this section, break up the long 2nd sentence, and correct the last sentence which currently reads as if haptophytes produce (alkenone-based) sea-surface temperatures. For example: Marine sediments are a natural archive of climate history and can be used to reconstruct sea ice extent via proxies such as biomarkers¹⁰ and/or microfossil remains¹¹. In the Bering Sea, a palaeoceanographic framework for past sea ice coverage has been based on reconstructions using the highly branched isoprenoid alkane biomarker IP25, ¹² (produced by diatoms bound to a life in seasonal sea ice¹⁰) and microfossils (from diatoms)¹¹. Alkenones, which are produced by haptophytes, can add complementary information about SSTs of the late summer/early fall season (SSTUK’37)¹³.”

L. 70: Suggest to add “in xyz organisms” after patterns at the end of sentence.

L. 77: sedaDNA ² sed in italics.

L78 – 83: I’d recommend to generalise this here in the Introduction, and then place the study-specific information in Methods.

L. 90ff “The paleoenvironmental backbone is based on previous multi-proxy reconstructions of the coring site derived from this core, including diatom microfossils and the biomarker IP25, which show that the coring site was covered by seasonal sea-ice during the Last Glacial Maximum (LGM), Heinrich Stadial 1, and the Younger Dryas, while the phases of the Bølling-Allerød and the Holocene were predominantly sea-ice free^{12,16}.” - Make this 2 sentences, replace “this core” with “the same core” for clarity, and rephrase as this reads as if the coring site is derived from the core. For example: “The palaeoenvironmental backbone of this coring site is based on previous multi-proxy reconstructions derived from the same core.

L94: no need to capitalise “Eukaryotes” here

L108: Capitalise Discussion

L11: Shot gun > make 1 word

L112: 918,186,452 paired-end reads of which within samples on average 70.76% passed the quality check – I'm not sure which samples are referred to here with the 70.76%, do you mean you had samples and controls, and 70% of the samples (excl. controls) passed the qc?

L121: mean fragment lengths between 83 and 105 bp – are these means per sample?

L139: Consider adding "low" before proportion of reads, to be more specific

L141: I'm unclear what "this discrepancy" refers to – that less zooplankton than phytoplankton reads were found? Please just add a few words to clarify.

L146 – 148: Very exciting finding!

Fig2: sea-surface temperature (SSTUK'37) reconstructions of the Northwest Pacific⁴¹ (blue) and Northeast Pacific⁴² (black) are added for comparison. - These data lines both look red to me.

L178 – 180: this sentence refers to both network analyses, whereas ll.175-177 and ll.181-188 refer to ecoCopula only, so I'd suggest to move ll.178-180 at the end of this paragraph as a final statement on how both analyses compare.

L191 and L197: use of sea-ice versus sea ice, choose one version throughout the text

L212: Add "sea" in front of ice

L227: add "fish" before spawning

L259: Node labels are given if a family exceeds 53 read counts. – Does 53 read counts correspond to a specific percentage? It seems arbitrary, a brief explanation why this number was chosen would help

L278ff: "The ratio between counts assigned to pelagic and benthic families (pelagic:benthic ratio) is highest in samples dated to the late glacial and Younger Dryas compared to the warmer Bølling/Allerød and the Holocene (Fig. 4). The pelagic:benthic ratio shows a weak, negative correlation with SSTs ($\rho = -0.39$, p -value = 0.052), indicating that more reads are assigned to benthic families (and more reads to pelagic families) in phases with warmer SSTs. – If you do include the bracketed text (only in Response letter, not in main text) then should this be "less" instead of "more" reads to pelagic families in the brackets? It might not apply so just mentioning it in case.

L317: delete “or”

L324: add “latter” before study

L337: add “of fish and mammals” before to 20,000 years

L349: Change The to “This” sedaDNA record

L363: something is missing before breeding success ...” decreased seabird breeding success” maybe?

L393f: “For a previous DNA metabarcoding analysis²⁰ we extracted DNA for

63 samples in 7 batches where each batch contained up to 9 samples and an extraction

blank (in total 7 negative controls). From those, we chose the stock solutions of 25 DNA

extracts and the corresponding extraction blanks, which had been kept continuously frozen

since the extraction for the metagenomic shotgun sequencing.” - The last/new sentence confuses me, do you mean: ...” which had been kept frozen continuously since extraction for the other (metabarcoding) study?”

L397: after “samples.” would be a good spot for the explanation on ages captured by the depth intervals (see above).

L445ff: There’s a sudden change in tense change to past tense for consistency.

L. 513ff: in R?

Supplementary Information:

L32: “The surface of the table and the core liner were cleaned DNAExitus Plus and rinsed of with MiliQ water”. - Add “with” after ‘were cleaned’ and remove “of”.

L37: “A syringe, of which the front was cut off directly before, was inserted aiming for a sample volume of up to 4 mL” - was this syringe sterile? If yes, how was it sterilised (after cutting it)?

L40: “During the whole procedure, full body suits, exchangeable arm sleeves, face masks, and two pairs of gloves on top of each other were worn.” - Assumedly, these upper gloves were changed when contaminated with sediments? Or between sections?

L45: in R?

Supplementary Figure 1.: The older samples (except the oldest 2) seem to have the highest number of read counts, is this correct?

Supp. Fig. 6: “Overall, the profiles show increasing accumulation of damage over the three binned timeframes for pelagic and benthic families” – While a trend can be assumed from the figure, it is not clear where/how this figure shows the increase with time/dataset.

Reviewer #2 (Remarks to the Author):

Manuscript#: NCOMMS-22-05048A

Title: Marine ecosystem shifts with deglacial sea-ice loss inferred from ancient DNA shotgun sequencing

Review of re-submitted MS

I thank the authors for having addressed all of my (extensive) review comments and suggestions as well as that of my fellow reviewers thoroughly, including re-analysis of the dataset. Their improvements to the MS, supplementary materials and tables bring a more robust support to their claims and interpretations and sufficient information to evaluate their work, which I would now recommend for publication.

Reviewer #3 (Remarks to the Author):

This is my second review of the manuscript entitled: "Marine ecosystem shifts with deglacial sea-ice loss inferred from ancient DNA shotgun sequencing"

Generally, I found the work done by the authors to answer my comments relevant and valid. In this respect, after rereading the second version of the manuscript, I have two additional comments:

1. I compared the first and second version of the manuscript and I realized that in the latest version of the manuscript, the way adjusted p-values are reported in the manuscript is almost always as "adjusted p-value < 0.2", while in the previous version either "adjusted p-value < 0.1" or "adjusted p-value < 0.05" was used. Reporting and interpreting results with a p-value threshold of 0.2 suggest that the results may not be as strong. In this respect, please justify and explain the relevance of these results in light of the higher p-value thresholds.

2. L528-530: "...we computed the Jaccard-index over the edges in the networks and compared the results with a randomized null-model."

Why was the Jaccard-index used instead of another index? Please justify.

REVIEWER COMMENTS

Reviewer #1 (Remarks to the Author):

Thank you for addressing all my comments.

I only have a few editing suggestions for the revised text sections, these are to improve clarity in a few instances. I am unable to comment on network analyses performed in this study as I am not familiar with these.

Abstract

Reviewer comment: Add ocean/region (e.g. “Bering Sea”, or “Arctic”) after Kamchatka, so it clear immediately which polar region (first sentence) this refers to.

Our response: Added.

Changes in manuscript:

Here, we used shotgun metagenomics of marine sedimentary ancient DNA off Kamchatka (Western Bering Sea) covering the last ~20,000 years.

Introduction

Reviewer comment: L. 22: “This will likely change the seasonal timing, biomass, and composition of algal blooms, which play a central role in trophic interactions by supporting food webs, including benthic communities as primary producers².” - Do you mean “and” primary producers? – As the algal blooms are basically the primary producers, you could leave what is after the comma off (or at least the last 3 words).

Our response: Changed.

Changes in manuscript: This will likely change the seasonal timing, biomass, and composition of algal blooms, which play a central role in trophic interactions by supporting food webs, including benthic communities and primary producers².

Reviewer comment: L25. Has the word decline twice in it, can you rephrase? Eg “reduction or retreat” of sea-ice?

Our response: Changed.

Changes in manuscript: Long-term trends in the Bering Sea show an ongoing decline of sea-ice duration which is projected to shorten further due to later freeze-up (by 20 to 30 days) and earlier break-up (by 10-20 days) until the middle of this century³.

Reviewer comment: L. 41: well documented

Our response: Changed.

Changes in manuscript: The short-term responses of the Bering Sea ecosystem to warmer and colder regimes have been well-documented^{2,5,8},[...].

Reviewer comment: L. 49: sea ice: choose “sea-ice” or “sea ice” throughout text for consistency.

Our response: We know that this seems inconsistent, but when sea ice is used as a noun or as a characteristic of sea ice it is not hyphenated (e.g. sea ice algae, sea ice-free) whereas when it is used as an adjective (sea-ice decline, sea-ice ecosystem) it is hyphenated. We would like to keep it grammatically correct.

Reviewer comment: L49ff: “Marine sediments are a natural archive of climate history from which sea ice can be reconstructed via proxy records, such as from biomarkers¹⁰ or microfossil remains¹¹. A regional palaeoceanographic framework for past sea ice coverage in the Bering Sea is based on reconstructions using the highly branched isoprenoid alkane biomarker IP₂₅,¹² produced by diatoms bound to a life in seasonal sea ice¹⁰, and diatom microfossils¹¹. Alkenone-based sea surface temperatures (SST_{UK’37}), which are produced by haptophytes, and can add complementary information about SSTs of the late summer/early fall season¹³.”

- I would suggest to **simplify** this section, break up the long 2nd sentence, and correct the last sentence which currently reads as if haptophytes produce (alkenone-based) sea-surface temperatures. For example: Marine sediments are a natural archive of climate history and can be used to reconstruct sea ice extent via proxies such as biomarkers¹⁰ and/or microfossil remains¹¹. In the Bering Sea, a palaeoceanographic framework for past sea ice coverage has been based on reconstructions using the highly branched isoprenoid alkane biomarker IP₂₅,¹² (produced by diatoms bound to a life in seasonal sea ice¹⁰) and microfossils (from diatoms)¹¹. Alkenones, which are produced by haptophytes, can add complementary information about SSTs of the late summer/early fall season (SST_{UK’37})¹³.”

Our response: Thank you for this suggestion. We changed the sentence according to your recommendation.

Changes in manuscript: In the Bering Sea, a palaeoceanographic framework for past sea ice coverage has been based on reconstructions using the highly branched isoprenoid alkane biomarker IP₂₅¹² (produced by diatoms bound to a life in seasonal sea ice¹⁰) and microfossils (from diatoms)¹¹. Alkenones, which are produced by haptophytes, can add complementary information about sea surface temperatures of the late summer/early fall season (SST_{UK’37})¹³.”

Reviewer comment: L. 70: Suggest to add “in xyz organisms” after patterns at the end of sentence.

Our response: Added.

Changes in manuscript: Correlation networks have been used in the interpretation of multidimensional data by means of co-occurrence networks and contributed to our understanding of ecosystems by identifying, for example, habitat preferences in aquatic bacterial communities²⁴ or geographic co-occurrence patterns in European amphibians, reptiles, breeding birds, and mammals²⁵.

Reviewer comment: L. 77: sedaDNA sed in italics.

Our response: Changed.

Changes in manuscript: Correlation networks have not yet been applied to sedaDNA data before.

Reviewer comment: L78 – 83: I'd recommend to generalise this here in the Introduction, and then place the study-specific information in Methods.

Our response: We moved the information to the material and methods section.

Changes in manuscript:

Introduction L87-89: As sedaDNA samples contain averaged information on decadal to centennial time-scales depending on sedimentation rates, correlation in the network should not be interpreted as ecological interactions between linked taxa or actual co-occurrence on a short-time scale.

Sample material L388-390: According to our age model, the one-centimeter thick samples contain on average a contribution of DNA deposition over 15.5 years during the late glacial and over 28.7 years during the Holocene.

Reviewer comment: L. 90ff “The paleoenvironmental backbone is based on previous multi-proxy reconstructions of the coring site derived from this core, including diatom microfossils and the biomarker IP₂₅, which show that the coring site was covered by seasonal sea-ice during the Last Glacial Maximum (LGM), Heinrich Stadial 1, and the Younger Dryas, while the phases of the Bølling-Allerød and the Holocene were predominantly sea-ice free^{12,16}.” -

Make this 2 sentences, replace “this core” with “the same core” for clarity, and rephrase as this reads as if the coring site is derived from the core. For example: “The palaeoenvironmental backbone of this coring site is based on previous multi-proxy reconstructions derived from the same core.

Our response: Changed.

Changes in manuscript: The paleoenvironmental backbone is based on previous multi-proxy reconstructions from the same core, including diatom microfossils and the biomarker IP₂₅. The proxies suggest that the coring site was covered by seasonal sea-ice during the Last Glacial Maximum (LGM), Heinrich Stadial 1, and the Younger Dryas, while the phases of the Bølling-Allerød and the Holocene were predominantly sea-ice free^{12,16}.

Reviewer comment: L94: no need to capitalise “Eukaryotes” here

Our response: Changed.

Changes in manuscript: Our approach includes a broad spectrum of taxonomic and functional groups particularly among eukaryotes, which we consider a major step towards deciphering the consequences of climate transitions for marine ecosystem structure.

Reviewer comment: L108: Capitalise Discussion

Our response: Changed.

Changes in manuscript: Results and Discussion

Reviewer comment: L11: Shot gun > make 1 word

Our response: Changed.

Changes in manuscript: Shotgun sequencing results

Reviewer comment: L112: 918,186,452 paired-end reads of which within samples on average 70.76% passed the quality check – I'm not sure which samples are referred to here with the 70.76%, do you mean you had samples and controls, and 70% of the samples (excl. controls) passed the qc?

Our response: The samples all passed the quality check. We here refer to reads within samples. Fewer reads passed the quality checks of the controls. We hope that our rephrased sentence makes this clearer.

Changes in manuscript: Sequencing (25 samples, 5 negative controls) resulted in 918,186,452 paired-end reads of which within samples on average 70.76% of the reads passed the quality check (Supplementary data file 1).

Reviewer comment: L121: mean fragment lengths between 83 and 105 bp – are these means per sample?

Our response: Yes, we changed among samples to per sample.

Changes in manuscript: This resulted in mean fragment lengths between 83 and 105 bp per sample, which matches the expectation that *sefaDNA* is highly fragmented³⁹.

Reviewer comment: L139: Consider adding “low” before proportion of reads, to be more specific

Our response: Added.

Changes in manuscript: The low proportion of assigned reads likely reflect a strong grazing intensity on zooplankton in the upper water column which leads to a low zooplankton DNA export towards the seafloor.

Reviewer comment: L141: I'm unclear what “this discrepancy” refers to – that less zooplankton than phytoplankton reads were found? Please just add a few words to clarify.

Our response: We rephrased the sentence.

Changes in manuscript: Alternatively, this discrepancy between the proportions of assigned reads to zooplankton (multicellular organisms) and phytoplankton (unicellular organisms) could be explained by differential preservation or degradation due to different skeletal properties or large gaps in the sequence reference database regarding marine macrofauna for both barcoding genes and genomes⁴⁰.

Reviewer comment: L146 – 148: Very exciting finding!

Fig2: sea-surface temperature (SSTUK'37) reconstructions of the Northwest Pacific⁴¹ (blue) and Northeast Pacific⁴² (black) are added for comparison. - These data lines both look red to me.

Our response: Changed in Figs. 3 and 4.

Changes in manuscript: The dendrogram (based on constrained hierarchical clustering) and IP₂₅ concentrations from the same core⁴² and sea-surface temperature (SST_{UK37}) reconstructions of the Northwest Pacific⁴³ (light red) and Northeast Pacific⁴⁴ (dark red) are added for comparison.

Reviewer comment: L178 – 180: this sentence refers to both network analyses, whereas ll.175-177 and ll.181-188 refer to ecoCopula only, so I'd suggest to move ll.178-180 at the end of this paragraph as a final statement on how both analyses compare.

Our response: We moved the sentence to the final part of this paragraph.

Reviewer comment: L191 and L197: use of sea-ice versus sea ice, choose one version throughout the text

Our response: Here, sea ice is used as a noun and is therefore correctly written without a hyphen. Therefore, we do not change it.

Reviewer comment: L212: Add "sea" in front of ice

Our response: Added.

Changes in manuscript: Closely linked to these primary producers are copepods (Calanidae 0.03%, Metridinidae 0.03%), which rely on sea ice algae as a food source due to their high quantity of polyunsaturated fatty acids⁴⁸.

Reviewer comment: L227: add "fish" before spawning

Our response: Added.

Changes in manuscript: Higher read counts assigned to Clupeidae (Pacific herring) and Salmonidae in warm phases, for example the early Holocene (Fig. 2), are consistent with previous studies that have shown that warmer temperatures provide good conditions for fish spawning, quick hatching from eggs, and low mortality rates during early life stages⁵⁴.

Reviewer comment: L259: Node labels are given if a family exceeds 53 read counts. – Does 53 read counts correspond to a specific percentage? It seems arbitrary, a brief explanation why this number was chosen would help

Our response: We log-transformed the node-size and therefore applied a log-scaled threshold of 4 for labeling nodes. If a taxon has at least 54 counts the label is plotted. We add the following sentence.

Changes in manuscript: Node labels are given if a family exceeds the log-scaled threshold of 4.

Reviewer comment: L278ff: "The ratio between counts assigned to pelagic and benthic families (pelagic:benthic ratio) is highest in samples dated to the late glacial and Younger Dryas compared to the warmer Bølling/Allerød and the Holocene (Fig. 4). The pelagic:benthic ratio shows a weak, negative correlation with SSTs ($\rho = -0.39$, p -value = 0.052), indicating that more reads are assigned to benthic families (and more reads to pelagic families) in phases with warmer SSTs. – If you do include

the bracketed text (only in Response letter, not in main text) then should this be “less” instead of “more” reads to pelagic families in the brackets? It might not apply so just mentioning it in case.

Our response: Thank you for pointing that out. It is not a mistake. More reads are assigned to both pelagic and benthic taxa, but proportionally more are assigned to benthic taxa so that the difference between them (the ratio) becomes smaller.

Changes in the manuscript:

During phases with higher SSTs there is a general increase in read counts of pelagic and benthic taxa. This increase is proportionally stronger in the benthic compared to pelagic taxa resulting in a smaller difference between them under warmer SSTs and thus a smaller pelagic:benthic ratio. Overall, the pelagic:benthic ratio shows a weak, negative correlation with SSTs ($\rho = -0.39$, p -value = 0.052), indicating that more reads are assigned to benthic families in phases with warmer SSTs.

Reviewer comment: L317: delete “or”

Our response: Deleted.

Changes in manuscript: Phytoplankton groups with large cell sizes such as diatoms are assumed to contribute more to carbon export than other pico-sized plankton such as phototrophic bacteria⁷⁸, which are presumed to be mostly recycled by the microbial loop in the water column⁷⁹.

Reviewer comment: L324: add “latter” before study

Our response: Added.

Changes in manuscript: The latter study also found that despite the abundance shifts of the two plankton size classes, the chlorophyll a content, which is often used as a proxy for biomass, remained stable.

Reviewer comment: L337: add “of fish and mammals” before to 20,000 years

Our response: Added.

Changes in manuscript: By pushing back the detection limit of fish and mammals to almost 20,000 years, our approach paves the way to assess millennial-scale to glacial–interglacial changes in the distribution and natural dynamics of these key groups in future studies.

Reviewer comment: L349: Change The to “This” sedaDNA record

Our response: Changed.

Changes in manuscript: This sedaDNA record implies that ongoing climate warming on the Bering Sea shelf might lead to a shift towards a pico-sized phytoplankton community composed mainly of cyanobacteria (Fig. 5) where previously, primary productivity was dominated by nano- and microphytoplankton.

Reviewer comment: L363: something is missing before breeding success ...” decreased seabird breeding success” maybe?

Our response: Added.

Changes in manuscript: However, high abundances of pink salmon have been connected with negative effects on other species such as increased seabird mortalities, changes in nesting phenology, and a decrease in breeding success due to their competitive advantage over shared prey^{5,56}.

Reviewer comment: L393f: “For a previous DNA metabarcoding analysis²⁰ we extracted DNA for 63 samples in 7 batches where each batch contained up to 9 samples and an extraction blank (in total 7 negative controls). From those, we chose the stock solutions of 25 DNA extracts and the corresponding extraction blanks, which had been kept continuously frozen since the extraction for the metagenomic shotgun sequencing.” - The last/new sentence confuses me, do you mean: ...” which had been kept frozen continuously since extraction for the other (metabarcoding) study?”

Our response: They were kept frozen since the extraction. We prepared 2 aliquots of the DNA directly after extraction and one was used for metabarcoding while the other was kept frozen and was dedicated to the shotgun sequencing.

Changes in manuscript: For a previous DNA metabarcoding analysis²⁰ we extracted DNA for 63 samples in 7 batches where each batch contained up to 9 samples and an extraction blank (in total 7 negative controls). From those, we chose the stock solutions of 25 DNA extracts and the corresponding extraction blanks, which had been kept continuously frozen since the extraction.

Reviewer comment: L397: after “samples.” would be a good spot for the explanation on ages captured by the depth intervals (see above).

L445ff: There’s a sudden change in tense change to past tense for consistency.

Our response: We would like to keep the information in the introduction.

L445ff: We changed the tense.

Changes in manuscript: The sample choice led to a mean temporal resolution of ~780 years, thereby ensuring that each millennium is represented by at least one and each climatic phase by at least two samples.

Reviewer comment: L. 513ff: in R?

Our response: Yes, we added the following sentence and adjusted all citations.

Changes in manuscript: All statistical analyses have been performed in R v. 4.0.3⁹².

Supplementary Information:

Reviewer comment: L32: “The surface of the table and the core liner were cleaned DNAExitus Plus and rinsed of with MiliQ water”. - Add “with” after ‘were cleaned’ and remove “of”.

Our response: We rephrased the sentence.

Changes in supplement: The surface of the table and the core liner were cleaned DNAExitus Plus which was rinsed off with MiliQ water.

Reviewer comment: L37: “A syringe, of which the front was cut off directly before, was inserted aiming for a sample volume of up to 4 mL” - was this syringe sterile? If yes, how was it sterilised (after cutting it)?

Our response: We used a sterile syringe and our treated (DNA-removal) knives to cut-off the syringe fronts.

Changes in supplement: A sterile syringe, of which the front was cut off with a clean, single-use knife (treated as described above) directly before, was inserted aiming for a sample volume of up to 4 mL.

Reviewer comment: L40: “During the whole procedure, full body suits, exchangeable arm sleeves, face masks, and two pairs of gloves on top of each other were worn.” - Assumedly, these upper gloves were changed when contaminated with sediments? Or between sections?

Our response: Of course. We added the following sentence in Chapter 1. Opening of cores and subsampling procedures to prevent contamination.

Changes in supplement: The gloves were changed between each sample and additionally, when it was contaminated by sediment.

Reviewer comment: L45: in R?

Our response: We added the following sentence.

Changes in supplement: The following analyses were carried out in R.

Reviewer comment: Supplementary Figure 1.: The older samples (except the oldest 2) seem to have the highest number of read counts, is this correct?

Our response: Yes. We added the following sentence.

Changes in supplement: The highest number of taxonomically assigned read counts can be found in samples dated between 18 and 13 cal kyr BP.

Reviewer comment: Supp. Fig. 6: “Overall, the profiles show increasing accumulation of damage over the three binned timeframes for pelagic and benthic families” – While a trend can be assumed from the figure, it is not clear where/how this figure shows the increase with time/dataset.

Our response: The data has been binned into three time intervals and the damage patterns are more pronounced in the bin with the oldest samples. The information about the binned time steps was in the Figure caption. We have prepared an improved Figure, which includes this information as an annotation and which fits on the same page to make the trend better observable.

Changes in supplement:

Caption: Damage profiles of a-b: dataset1 (1.08–5.6 cal kyr BP), c-d: dataset2 (6.3–12.6 cal kyr BP), and e-f: dataset3 (13.6–19.9 cal kyr BP). C>T substitutions in 5' direction and G>A substitutions in 3' and the number of reads used are given for each taxon analyzed. Non-C>T substitutions in 5' direction and non-G>A substitutions in 3' direction are given to estimate the noise (gray to dark gray color). The general color code for benthic taxa (*Asterias rubens* and *Pecten maximus*) is brown shades, eukaryotic algae (euk algae) are green shades, pelagic fish are blue shades, and bacteria are purple.

Overall, the profiles show increasing accumulation of damage over the three binned timeframes for pelagic and benthic families.

Reviewer #2 (Remarks to the Author):

Manuscript#: NCOMMS-22-05048A

Title: Marine ecosystem shifts with deglacial sea-ice loss inferred from ancient DNA shotgun sequencing

Review of re-submitted MS

I thank the authors for having addressed all of my (extensive) review comments and suggestions as well as that of my fellow reviewers thoroughly, including re-analysis of the dataset. Their

improvements to the MS, supplementary materials and tables bring a more robust support to their claims and interpretations and sufficient information to evaluate their work, which I would now recommend for publication.

Reviewer #3 (Remarks to the Author):

This is my second review of the manuscript entitled: “Marine ecosystem shifts with deglacial sea-ice loss inferred from ancient DNA shotgun sequencing”

Generally, I found the work done by the authors to answer my comments relevant and valid. In this respect, after rereading the second version of the manuscript, I have two additional comments:

Reviewer comment: 1. I compared the first and second version of the manuscript and I realized that in the latest version of the manuscript, the way adjusted p-values are reported in the manuscript is almost always as “adjusted p-value < 0.2”, while in the previous version either “adjusted p-value < 0.1” or “adjusted p-value < 0.05” was used. Reporting and interpreting results with a p-value threshold of 0.2 suggest that the results may not be as strong. In this respect, please justify and explain the relevance of these results in light of the higher p-value thresholds.

Our response: We apologize for the loss of clarity after revision of this part. In the last version of the manuscript we used adjusted p-value < 0.1 (not 0.2 as stated by the reviewer) for Spearman network generation and < 0.2 for reporting the taxa-environmental relationships.

We want to confirm that none of our stated results from network analyses are sensitive to the choice of the significance level cutoff (adjusted p-value threshold). This can be visually inferred from comparing networks that were derived from using different p-value thresholds both for network generation and for correlation between taxa and environmental variables. For example, the networks generated with an adjusted p-value cutoff at 0.05 (a-c) and at 0.1 (d-f) are showing very similar patterns including the location of taxa in the mentioned sea-ice and ice-freesub-networks (main result in the text). To indicate that p-value threshold choice does not impact the conclusions in the manuscript we show this figure in the **supplement of the revised manuscript**.

New Supplementary Figure 16: Effect of adjusted p-value threshold ($p < 0.05$ (a-c) and $p < 0.1$ (d-f)) on the network topology. Nodes showing a positive trend (Spearman rank correlations > 0.5 with thresholds of adjusted p-value set to $p < 0.05$ (a,d), $p < 0.1$ (b,e) and $p < 0.2$ (c,f)) with the seasonal sea-ice biomarker IP₂₅ are colored blue or with SST_{UK:37} are colored red.

To improve consistency and clarity in the manuscript, we use in the main revised text consistently > 0.1 as p-value threshold which equals an correlation coefficient (r) of > 0.5 in our analyses i.e. the network is built only from moderately to strongly linked taxa (similar to the last version). Furthermore, we will set the adjusted p-value threshold back to 0.1 for highlighting taxa (i.e. taxa that are moderately to strongly linked with environmental variables (IP₂₅ and SST) (like in the first version of the submitted manuscript and consistent with the taxa-taxa correlation threshold). We consider these p-value thresholds (and corresponding r -values) appropriate for our purpose because the intention of our network analysis is to visualize the marine ecosystem structure not to emphasize specific interactions among single taxa or between single taxa and environmental variables (which would potentially call for more conservative p-values or more data). We assume that a more densely populated network will stimulate new hypotheses for future research.

We are fully aware that careful statistical evaluation of associations in our dataset is necessary and important. Therefore, we have always applied the Benjamini-Hochberg method to correct for false discoveries during our initial manuscript preparations, which is a conservative and standard approach to multiple testing. Yet, we would also like to point out that a larger sample size would have reduced the impact of random error and thus the p-value (These et al. 2016).

Thiese, M. S., Ronna, B. & Ott, U. P value interpretations and considerations. *J Thorac Dis* **8**, E928–E931 (2016).

Changes in manuscript L589: Networks with more and less stringent p-value thresholds are shown in Supplementary Fig. 16.

Adjusted Supplementary Figure 11:

Supplementary Figure 11. Neighbors of IP₂₅- and SST-correlated families. Boxplots showing the number of links to adjacent nodes (neighbors) of IP₂₅- and SST-correlated nodes on the level of functional groups (green) and family (blue). Nodes which are correlated (Spearman rho > 0.4, adjusted p-value < 0.1) with IP₂₅ are significantly connected to more families in comparison to SST-correlated nodes (two sample t-test), but not significantly to more functional groups. The median is represented by the horizontal line inside the box, while lower and upper ends of the box represent the 25th and 75th percentiles, respectively. The upper ends of the whiskers correspond to the smallest and largest values of the 1.5 times interquartile ranges below the 25th and above the 75th percentiles. Outliers are marked as individual circles outside the box.

Reviewer comment: 2. L528-530: "...we computed the Jaccard-index over the edges in the networks and compared the results with a randomized null-model." Why was the Jaccard-index used instead of another index? Please justify.

Our response: Our intention is to show the similarity of the two networks, which means to measure how many of the edges are present in both networks. **We chose the Jaccard similarity as a simple, intuitive metric for binary data and we consider its application for this purpose fully appropriate and will justify this in a revised text version.** Nonetheless we made a test with another appropriate yet more complicated distance measure i.e. Manhattan distance, which yields the same conclusion.

New text in the supplement (Chapter “Gaussian copula graphical model (ecoCopula network) and comparisons with Spearman network”): We chose the Jaccard similarity as a simple, intuitive metric for binary data to measure how many edges are present in both networks. Following Tantardini et al.¹⁰, the simplest way of evaluating differences between two adjacency matrices is to compute differences directly, for which several metrics can be used. Since (1) we are not including the edge weights and only need a binary metric (an edge is either present or absent) and (2) shared presences of edges are more informative than shared absences of edges (asymmetric similarity), the Jaccard similarity¹¹, which divides the intersection of edges by the union of edges, is therefore appropriate. Shared absences might not reflect the similarity at all¹², therefore other indices, such as euclidean distance were omitted.

10 Tantardini, M., Ieva, F., Tajoli, L. & Piccardi, C. Comparing methods for comparing networks. *Sci Rep* **9**, 17557 (2019).

11 Jaccard, P. Distribution de la flore alpine dans le Bassin des Dranses et dans quelques regions voisines. *Bull Soc Vaudoise Sci Nat* 241–272 (1901).

12 Brusco, M., Cradit, J. D. & Steinley, D. A comparison of 71 binary similarity coefficients: The effect of base rates. *PLOS ONE* **16**, e0247751 (2021).

REVIEWERS' COMMENTS

Reviewer #3 (Remarks to the Author):

This is my third review of the manuscript entitled: "Marine ecosystem shifts with deglacial sea-ice loss inferred from ancient DNA shotgun sequencing"

I am generally happy with the work done by the authors to answer my comments. In this respect, I do not have any additional comments on the manuscript.